# REVISIT VISUAL PROMPT TUNING: THE EXPRESSIVENESS OF PROMPT EXPERTS

**Minh Le**[*3†], **Anh Nguyen**[*1], **Huy Nguyen**[2], **Chau Nguyen**[4†], **Anh Tran**[1], **Nhat Ho**[2]

[1] Qualcomm AI Research[‡]    [2] The University of Texas at Austin
[3] Trivita AI    [4] Hanoi National University of Education

## ABSTRACT

Visual Prompt Tuning (VPT) has proven effective for parameter-efficient adaptation of pre-trained vision models to downstream tasks by inserting task-specific learnable prompt tokens. Despite its empirical success, a comprehensive theoretical understanding of VPT remains an active area of research. Building on the recently established connection between Mixture of Experts (MoE) and prompt-based methods, wherein each attention head can be conceptualized as a composition of multiple MoE models, we reinterpret VPT as the introduction of new *prompt experts* into these MoE structures. We identify a key limitation in existing VPT frameworks: the *restricted functional expressiveness* of prompt experts, which remain static and thus limited in their adaptability. To address this, we propose **Visual Adaptive Prompt Tuning (VAPT)**, a novel method that endows prompt experts with enhanced expressiveness while preserving parameter efficiency. Empirical evaluations on VTAB-1K and FGVC demonstrate that VAPT achieves *substantial performance improvements*, surpassing fully fine-tuned baselines by **7.34%** and **1.04%**, respectively. Moreover, VAPT consistently outperforms VPT while *requiring fewer additional parameters*. Furthermore, our theoretical analysis indicates that VAPT achieves optimal sample efficiency. Collectively, these results underscore the theoretical grounding and empirical advantages of our approach. Our code is publicly available at https://github.com/Minhchuyentoancbn/VAPT.

## 1 INTRODUCTION

Foundational vision models, pre-trained on large-scale datasets (Dosovitskiy, 2020; Radford et al., 2021; Kirillov et al., 2023), have demonstrated remarkable success and robust generalization across a wide range of computer vision tasks. As a result, fine-tuning these models for specific downstream tasks has become a widely adopted paradigm (Iofinova et al., 2022). However, fully fine-tuning large foundational models can be computationally prohibitive, leading to growing interest in parameter-efficient fine-tuning (PEFT) techniques (Cai et al., 2020; Zhang et al., 2020; Hu et al., 2021), which update only a small subset of parameters. Among these methods, Visual Prompt Tuning (VPT) (Jia et al., 2022) has emerged as a simple yet powerful approach that appends learnable *prompt* tokens to the input, which serve as task-specific instructions to guide the pre-trained transformer model. Despite VPT's empirical effectiveness, its theoretical underpinnings remain an active area of research (Petrov et al., 2023; Oymak et al., 2023; Wang et al., 2024).

Recently, Le et al. (2024) established a formal connection between attention mechanisms (Vaswani, 2017), prompt-based methods, and Mixture of Experts (MoE) models (Jacobs et al., 1991; Shazeer et al., 2017), yielding new insights into the design and optimization of prompting strategies. Their analysis demonstrates that *each attention head* in a transformer can be equivalently interpreted as *a composition of multiple MoE models* stacked together. Within this framework, VPT corresponds to fine-tuning these implicit, pre-trained MoE structures by introducing new, learnable *prompt experts*. These prompt experts collaborate with the pre-trained experts to facilitate effective task adaptation.

---

[*] Equal contribution    [†] Work done while at Qualcomm
[‡] Qualcomm AI Research is an initiative of Qualcomm Technologies, Inc.

This connection opens avenues for deeper theoretical investigations and the development of advanced strategies for prompt-based learning (Le et al., 2025b;a; Diep et al., 2025).

Building on this MoE interpretation, we identify a key limitation in the standard formulation of VPT. While the pre-trained experts within attention heads are linear functions of the input features, the newly introduced *prompt experts are modeled as constant, input-invariant vectors*. Given that effective adaptation relies on collaboration between these prompt and pre-trained experts, we hypothesize that this functional disparity, specifically, the restricted expressiveness of static prompts, may constrain VPT's adaptation efficacy. This design also deviates from standard MoE practices, where experts are typically designed to be input-adaptive. However, increasing the expressiveness of prompt experts raises concerns about parameter and computational overhead. This leads us to the following question:

> *(Q) Can we improve model performance by increasing the expressiveness of prompt experts while maintaining parameter efficiency?*

To answer this question, we propose **Visual Adaptive Prompt Tuning (VAPT)**, a novel approach that incorporates input-dependent prompt experts while *maintaining the parameter efficiency* characteristic of VPT. The VAPT design comprises two main components: *token-wise projectors* and a shared *feature projector*, which leverage global information from input features to generate adaptive prompt tokens. A key advantage of VAPT is its efficiency: it achieves enhanced expressiveness with minimal computational overhead, adding only **0.6%** FLOPs relative to VPT and requiring fewer trainable parameters. Additionally, the token-wise projectors and channel-wise convolutions result in a structurally simple formulation of the prompt experts. This simplicity enables a rigorous theoretical analysis (see Section 5), a contribution largely absent in the prior prompt-tuning literature. Our analysis demonstrates that VAPT attains optimal sample efficiency for prompt estimation. Empirical results strongly corroborate these theoretical findings. For example, in a low-data regime on the Stanford Dogs dataset (Khosla et al., 2011) (using only **1%** of training data), VAPT achieves **60.1%** accuracy, a stark contrast to VPT's **3.6%**. Moreover, on standard benchmarks VTAB-1K (Zhai et al., 2019) and FGVC (Jia et al., 2022), VAPT significantly improves performance over fully fine-tuned baselines by **7.34%** and **1.04%**, respectively. Crucially, *VAPT consistently outperforms VPT* across benchmarks despite *utilizing fewer parameters*, underscoring its theoretical and empirical strengths.

**Contributions. 1.** From MoE perspective, we identify a key limitation in the formulation of VPT: its prompt experts are input-invariant, thereby constraining their functional expressiveness. **2.** We introduce **VAPT**, a novel formulation that injects input-adaptive prompt experts while preserving the parameter-efficiency of VPT. **3.** VAPT's simple formulation enables a theoretical analysis to validate its effectiveness, an aspect largely missing in prior work. Our theoretical analysis demonstrates that VAPT attains optimal sample efficiency for prompt estimation, providing a rigorous foundation for its practical value. **4.** Extensive experiments show that VAPT consistently surpasses VPT with fewer trainable parameters, validating its effectiveness and efficiency both theoretically and empirically.

**Notation.** For $n \in \mathbb{N}$, let $[n] = \{1, 2, \ldots, n\}$. For a set $S$, $|S|$ denotes its cardinality. Given a vector $u = (u_1, u_2, \ldots, u_d) \in \mathbb{R}^d$ and $\alpha = (\alpha_1, \alpha_2, \ldots, \alpha_d) \in \mathbb{N}^d$, define $u^\alpha = u_1^{\alpha_1} u_2^{\alpha_2} \ldots u_d^{\alpha_d}$, $|u| = u_1 + u_2 + \ldots + u_d$ and $\alpha! = \alpha_1! \alpha_2! \ldots \alpha_d!$. Let $\|u\|$ denote the Euclidean norm of $u$. For positive sequences $(a_n)_{n \geq 1}$ and $(b_n)_{n \geq 1}$, we write $a_n = \mathcal{O}(b_n)$ or $a_n \lesssim b_n$ if $a_n \leq C b_n$ for all $n \in \mathbb{N}$ for some $C > 0$. The notation $a_n = \mathcal{O}_P(b_n)$ indicates $a_n/b_n$ is stochastically bounded.

## 2 BACKGROUND

In this section, we first review Visual Prompt Tuning in Section 2.1 and then the Mixture of Experts model in Section 2.2. Additional related work is presented in Appendix C.

### 2.1 VISUAL PROMPT TUNING

Vision Transformer (ViT) (Dosovitskiy, 2020) has proven to be a powerful backbone architecture for visual recognition. A ViT contains $L$ blocks, each comprising a *multi-head self-attention* (MSA) layer followed by a *feed-forward network* (FFN). For clarity, we consider the $l$-th ViT block. Let $\tilde{\boldsymbol{X}}^{(l)} = \left[ \boldsymbol{x}_1^{(l)}, \ldots, \boldsymbol{x}_N^{(l)} \right]^\top \in \mathbb{R}^{N \times d}$ be the input tokens, where $N$ is the sequence length, and $d$ is the embedding dimension. The MSA layer processes $\boldsymbol{X}^Q = \boldsymbol{X}^K = \boldsymbol{X}^V = \tilde{\boldsymbol{X}}^{(l)} \in \mathbb{R}^{N \times d}$ as queries,

keys, and values, producing:

$$\text{MSA}(\tilde{\boldsymbol{X}}^{(l)}) = \text{Concat}(\boldsymbol{h}_1, ..., \boldsymbol{h}_M)W^O \in \mathbb{R}^{N \times d},$$

$$\boldsymbol{h}_m = \text{Attention}(\boldsymbol{X}^Q W_m^Q, \boldsymbol{X}^K W_m^K, \boldsymbol{X}^V W_m^V) \in \mathbb{R}^{N \times d_v}, \tag{1}$$

for $m \in [M]$, where $M$ is the number of attention heads, $W_m^Q \in \mathbb{R}^{d \times d_k}$, $W_m^K \in \mathbb{R}^{d \times d_k}$, $W_m^V \in \mathbb{R}^{d \times d_v}$, and $W^O \in \mathbb{R}^{Md_v \times d}$ are projection matrices with $d_k = d_v = \frac{d}{M}$.

Visual Prompt Tuning (VPT) (Jia et al., 2022) adapts a pre-trained ViT by appending $N_p$ learnable *prompt* parameters $\boldsymbol{P}^{(l)} = \left[\boldsymbol{p}_1^{(l)}, \ldots, \boldsymbol{p}_{N_p}^{(l)}\right]^\top \in \mathbb{R}^{N_p \times d}$, (where $N_p$ is the prompt length), to the input of each ViT block, thereby modifying the MSA layer's output as follows:

$$\text{MSA}(\tilde{\boldsymbol{X}}^{(l)}, \boldsymbol{P}^{(l)}) = \text{Concat}(\tilde{\boldsymbol{h}}_1, ..., \tilde{\boldsymbol{h}}_M)W^O,$$

$$\tilde{\boldsymbol{h}}_m = \text{Attention}\left(\begin{bmatrix}\boldsymbol{X}^Q \\ \boldsymbol{P}^{(l)}\end{bmatrix}W_m^Q, \begin{bmatrix}\boldsymbol{X}^K \\ \boldsymbol{P}^{(l)}\end{bmatrix}W_m^K, \begin{bmatrix}\boldsymbol{X}^V \\ \boldsymbol{P}^{(l)}\end{bmatrix}W_m^V\right)$$

$$= \left[\tilde{\boldsymbol{h}}_{m,1}, \ldots, \tilde{\boldsymbol{h}}_{m,N+N_p}\right]^\top \in \mathbb{R}^{(N+N_p) \times d_v}. \tag{2}$$

During training, *only the prompts $\boldsymbol{P}^{(l)}$ and the classification head are updated*, while all ViT weights, including $W_m^Q$, $W_m^K$, $W_m^V$, and $W_O$, remain frozen. Within the VPT framework, MSA output tokens corresponding to the input prompt locations are discarded and replaced by the prompts of the next layer $\boldsymbol{P}^{(l+1)}$, before these tokens enter the subsequent block. Consequently, in Equation (2), $\tilde{\boldsymbol{h}}_{m,N+1}, \ldots, \tilde{\boldsymbol{h}}_{m,N+N_p}$ are not utilized downstream and their computation can therefore be bypassed.

## 2.2 MIXTURE OF EXPERTS

Mixture of Experts (MoE) is a class of statistical machine learning frameworks that combines multiple models, known as experts, to produce more expressive and accurate predictions (Jacobs et al., 1991; Jordan & Jacobs, 1994). An MoE model typically consists of $N'$ *expert functions*, $f_i : \mathbb{R}^d \to \mathbb{R}^{d_v}$ for $i \in [N']$, along with a *gating function*, $G : \mathbb{R}^d \to \mathbb{R}^{N'}$ which assigns weights to experts based on learned *score functions*, $s_i : \mathbb{R}^d \to \mathbb{R}$. For a given input $\boldsymbol{h} \in \mathbb{R}^d$, the MoE model generates output as:

$$\mathbf{y} = \sum_{j=1}^{N'} G(\boldsymbol{h})_j \cdot f_j(\boldsymbol{h}) = \sum_{j=1}^{N'} \frac{\exp\left(s_j(\boldsymbol{h})\right)}{\sum_{\ell=1}^{N'} \exp\left(s_\ell(\boldsymbol{h})\right)} \cdot f_j(\boldsymbol{h}),$$

where $G(\boldsymbol{h}) = \text{softmax}(s_1(\boldsymbol{h}), \ldots, s_{N'}(\boldsymbol{h}))$. Recent studies reveal connections between the attention mechanism, prompt-based methods, and MoE frameworks (Le et al., 2024; 2025b), presenting new promising opportunities to investigate prompt-based techniques through MoE lens.

## 3 MOTIVATION

**Mixture of Experts meets Visual Prompt Tuning.** We begin by discussing the connection between VPT and MoE. Following the notation from Section 2.1 and Equation (2), let $\boldsymbol{X}^{(l)} = \left[\boldsymbol{x}_1^{(l)^\top}, \ldots, \boldsymbol{x}_N^{(l)^\top}\right]^\top \in \mathbb{R}^{Nd}$ be the concatenation of all $N$ input tokens to the $l$-th MSA layer. For notational simplicity, the layer superscript $(l)$ is omitted in the subsequent derivations. As shown in Le et al. (2024), each output vector $\tilde{\boldsymbol{h}}_{m,i}$ within the $m$-th attention head can be *equivalently* expressed as the output of an MoE model with input $\boldsymbol{X}$. Specifically, we define the set of experts as:

$$f_j(\boldsymbol{X}) = W_m^{V^\top} E_j \boldsymbol{X} = W_m^{V^\top} \boldsymbol{x}_j, \tag{3}$$

$$f_{N+j'}(\boldsymbol{X}) = W_m^{V^\top} \boldsymbol{p}_{j'}, \tag{4}$$

where $j \in [N]$ and $j' \in [N_p]$. The corresponding score functions are defined as:

$$s_{i,j}(\boldsymbol{X}) = \frac{\boldsymbol{X}^\top E_i^\top W_m^Q W_m^{K^\top} E_j \boldsymbol{X}}{\sqrt{d_v}} = \frac{\boldsymbol{x}_i^\top W_m^Q W_m^{K^\top} \boldsymbol{x}_j}{\sqrt{d_v}}, \tag{5}$$

$$s_{i,N+j'}(\boldsymbol{X}) = \frac{\boldsymbol{X}^\top E_i^\top W_m^Q W_m^{K^\top} \boldsymbol{p}_{j'}}{\sqrt{d_v}} = \frac{\boldsymbol{x}_i^\top W_m^Q W_m^{K^\top} \boldsymbol{p}_{j'}}{\sqrt{d_v}}, \tag{6}$$

for $i \in [N]$. Here, $E_j \in \mathbb{R}^{d \times Nd}$ is a two-dimensional matrix such that $E_j \boldsymbol{X} = \boldsymbol{x}_j$. Finally, the output of the $m$-th attention head for the $i$-th token can be expressed as follows:

$$\tilde{\boldsymbol{h}}_{m,i} = \sum_{j=1}^{N} \frac{\exp(s_{i,j}(\boldsymbol{X}))}{\sum_{k=1}^{N+N_p} \exp(s_{i,k}(\boldsymbol{X}))} f_j(\boldsymbol{X}) + \sum_{j'=1}^{N_p} \frac{\exp(s_{i,N+j'}(\boldsymbol{X}))}{\sum_{k=1}^{N+N_p} \exp(s_{i,k}(\boldsymbol{X}))} f_{N+j'}(\boldsymbol{X}). \quad (7)$$

These formulations reveal that *each attention head* in ViT implicitly encodes *multiple MoE models*, $\tilde{\boldsymbol{h}}_{m,i}$ for $i \in [N]$. This contrasts with conventional MoE layers (Shazeer et al., 2017), where experts and their gating functions typically operate on individual token embeddings $\boldsymbol{x}_i$. In our formulation, the expert networks and their score functions process the entire input sequence $\boldsymbol{X}$. Furthermore, these initial experts $f_1, \ldots, f_N$ and score functions are inherently part of the pre-trained ViT, with their parameters embedded within the model weights (see Equations (3) and (5)), thereby constituting *pre-trained experts*. Meanwhile, the new *prompt experts* $f_{N+1}, \ldots, f_{N+N_p}$, whose learnable parameters are contained within prompts (see Equations (4) and (6)), are introduced to efficiently adapt the model to downstream tasks. Consequently, VPT can be viewed as an efficient method for fine-tuning these implicit MoE models by *adding new prompt experts*. These added experts effectively act as downstream task experts, enabling specialization for new tasks without retraining the entire model.

**Restricted Functional Expressiveness of Prompt Experts.** Equation (4) reveals a key limitation of prompt experts $f_{N+1}, \ldots, f_{N+N_p}$. Although they are formally functions of $\boldsymbol{X}$, they are represented by fixed prompt vectors $\boldsymbol{p}_1, \ldots, \boldsymbol{p}_{N_p}$ that *remain constant regardless of the input*. While their associated score functions $s_{i,N+j'}$ in Equation (6) are input-dependent linear functions, the functional form realized by the prompt expert itself is static. *This departs from typical MoE usage in the literature*, where experts are adaptive functions of the input. Furthermore, it contrasts with the pre-trained experts $f_1, \ldots, f_N$, which are linear functions of $\boldsymbol{X}$ (see Equation (3)), making them comparatively more expressive. We hypothesize that this limited flexibility may constrain the effectiveness of VPT as a fine-tuning strategy. Supporting this, prior work Petrov et al. (2023) demonstrates that prompt tuning can only add a bias term to the output of an attention block, thereby restricting representational capacity. This observation naturally motivates the central question: *Can the performance of visual prompt tuning be improved by making prompt experts more expressive?* Addressing this, we propose a novel prompt formulation to enhance their expressiveness in Section 4.

**Balancing Expressivity and Efficiency.** Despite the limitations noted above, one of the main advantages of the current prompt design in VPT is its simplicity. As indicated by Equation (4), each prompt expert requires only $d$ parameters, making it highly *parameter-efficient*. One might consider a naive linear design for a prompt expert $f_{N+j'} : \mathbb{R}^{Nd} \to \mathbb{R}^{d_v}$, where $f_{N+j'}(\boldsymbol{X}) = W_{j'}^\top \boldsymbol{X}$. However, this approach would necessitate up to $Nd \times d_v$ parameters, drastically increasing storage and computational overhead compared to the current approach, especially given the high dimensionality $Nd$ of the input $\boldsymbol{X}$. This illustrates the core design challenge: *enhancing prompt expert expressiveness without sacrificing the parameter efficiency that makes VPT attractive*. Our objective is therefore to devise a prompt mechanism that effectively balances these competing requirements.

## 4 VAPT: Visual Adaptive Prompt Tuning

To investigate adaptive prompting, this section presents our proposed method, VAPT, which integrates two key modules: token-wise projectors and a feature projector. These components are detailed in Section 4.1 and Section 4.2, respectively. Figure 1 illustrates the overall VAPT architecture.

### 4.1 Aggregating Global Information

To rigorously design the prompt experts, we first re-examine the formulation of the input and pre-trained experts. The input is defined as $\boldsymbol{X} = \left[ \boldsymbol{x}_1^\top, \ldots, \boldsymbol{x}_N^\top \right]^\top$, formed by concatenating $N$ feature tokens. These tokens can be spatially organized into the input *feature map* $\boldsymbol{X}_{\text{img}} \in \mathbb{R}^{H \times W \times d}$, where $H$ and $W$ are its height and width, respectively, and $N = H \times W$. Given that each token $x_j \in \mathbb{R}^d$ corresponds to a small patch of this feature map, Equation (3) implies that each pre-trained expert $f_j$ process their respective patches $\boldsymbol{x}_j$ independently. Consequently, these experts are inherently limited to capturing only *local information* within the feature map.

**Token-wise Projectors.** As prompt experts are designed to collaborate with pre-trained experts for adaptation to downstream tasks, it is desirable for them to specialize in information complementary

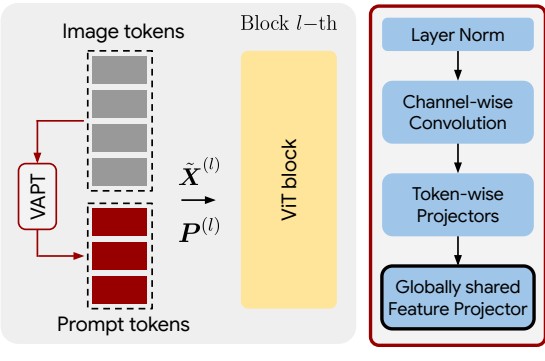

Figure 1: Overview of VAPT. Unlike VPT, where prompt tokens remain constant irrespective of the input, VAPT dynamically generates prompt tokens $\boldsymbol{P}^{(l)}$ conditioned on the input $\tilde{\boldsymbol{X}}^{(l)}$ via a VAPT block. This endows prompt experts with more adaptive and expressive functional formulation.

to that captured by pre-trained experts. To this end, our prompt experts are designed to extract *global information* from the feature map using *token-wise projectors*, defined as follows:

$$G_{j'}(\boldsymbol{X}) = (\sum_{k=1}^{N} \alpha_{j',k} E_k)\boldsymbol{X} = \sum_{k=1}^{N} \alpha_{j',k}\boldsymbol{x}_k \in \mathbb{R}^d, \tag{8}$$

for $j' \in [N_p]$, where $\alpha_{j',k}$ are learnable scalars for $k \in [N]$. These projectors aggregate tokens across spatial locations, thereby facilitating global information exchange within the feature map.

**Channel-wise Convolution.** Token-wise projectors aggregate features through weighted averaging of feature tokens. While computationally efficient, this approach may not fully capture important *spatial relationships*. For instance, the adjacency between $\boldsymbol{x}_1$ and $\boldsymbol{x}_2$, which correspond to neighboring patches in the feature map, can be overlooked by token-wise projectors that treat tokens independently of their spatial origin. To address this, we introduce a *channel-wise convolution* layer, applied to the input feature map before token-wise projectors. Let $F = [w_{i,j}]_{i,j=1}^{K}$ represent a kernel of size $K$. Unlike standard convolution kernels which utilize distinct weights for each input channel, $F$ reuses the *same $K \times K$ weights across all $d$ channels*. This design significantly reduces the parameter count by a factor of $d$, without sacrificing its ability to model local spatial interactions. We show that this channel-wise convolution not only saves parameters but also improves performance in Appendix E.6. Formally, the channel-wise convolution is applied to $\boldsymbol{X}_{\text{img}}$ as:

$$\boldsymbol{X}_{\text{conv}} = F * \boldsymbol{X}_{\text{img}} \in \mathbb{R}^{H' \times W' \times d}, \tag{9}$$

where $*$ denotes the 2D convolution operation, and $H' = H - K + 1$ and $W' = W - K + 1$ are the height and width of the output feature map, respectively. By convolving neighboring patches, this operation explicitly encodes local spatial relationships into the feature map $\boldsymbol{X}_{\text{conv}}$. We employ a single channel-wise convolution layer within each ViT block. The resulting feature map $\boldsymbol{X}_{\text{conv}}$ is subsequently flattened to $[\boldsymbol{x}_1^{\text{conv}}, \ldots, \boldsymbol{x}_{H' \cdot W'}^{\text{conv}}]^\top \in \mathbb{R}^{H' \cdot W' \times d}$ before being processed by the token-wise projectors. The final aggregated features can be expressed as:

$$G_{j'}(\boldsymbol{X}_{\text{conv}}) = \sum_{k=1}^{H' \cdot W'} \alpha_{j',k}\boldsymbol{x}_k^{\text{conv}} = W_{j'}\boldsymbol{X} \in \mathbb{R}^d \tag{10}$$

for $j' \in [N_p]$. Crucially, since both channel-wise convolution and token-wise projection are linear operations, the aggregated features also constitute a linear function of $\boldsymbol{X}$, with the transformation weights $W_{j'} \in \mathbb{R}^{d \times \tilde{N}d}$. This aggregated information is then leveraged to construct our adaptive prompts in the next section.

## 4.2 ADAPTIVE PROMPT

Each prompt initially aggregates its corresponding global feature token $G_{j'}(\boldsymbol{X}_{\text{conv}})$ through a token-wise projector. Subsequently, to generate the final adaptive prompts, we introduce a *feature projector* implemented as a small MLP $g : \mathbb{R}^d \to \mathbb{R}^d$, defined as follows:

$$g(\boldsymbol{x}) = W^{(2)}\sigma(W^{(1)}\boldsymbol{x}), \tag{11}$$

where $W^{(1)} \in \mathbb{R}^{r \times d}$, $W^{(2)} \in \mathbb{R}^{d \times r}$, $r \ll d$, and $\sigma(\cdot)$ is a non-linear activation function (ReLU in this work). This feature projector is applied to the aggregated features to produce the final adaptive prompt tokens. Formally, the adaptive prompts at each block are given by:

$$\boldsymbol{P}_{j'}(\boldsymbol{X}) = g(G_{j'}(\boldsymbol{X}_{\text{conv}})) = W^{(2)}\sigma(W^{(1)}W_{j'}\boldsymbol{X}) = W^{(2)}\sigma(W_{j'}^{(1)}\boldsymbol{X}) \in \mathbb{R}^d, \qquad (12)$$

for $j' \in [N_p]$, where $W_{j'}^{(1)} = W^{(1)}W_{j'}$. Our proposed adaptive prompts refine the prompt experts and their associated score functions within the MSA layers as follows:

$$f_{N+j'}(\boldsymbol{X}) = W_m^{V\top}\boldsymbol{P}_{j'}(\boldsymbol{X}), \qquad (13)$$

$$s_{i,N+j'}(\boldsymbol{X}) = \frac{\boldsymbol{X}^\top E_i^\top W_m^Q W_m^{K\top}\boldsymbol{P}_{j'}(\boldsymbol{X})}{\sqrt{d_v}}, \qquad (14)$$

where $i \in [N]$ and $j' \in [N_p]$. Equation (13) demonstrates that the prompt experts become adaptive to the input $\boldsymbol{X}$, enhancing their expressiveness and addressing the limited flexibility discussed in Section 3. The statistical advantages of this formulation are further analyzed in Section 5.

**Efficiency Considerations.** Following VPT, we insert adaptive prompts into every ViT block. While the feature projector is a lightweight MLP, distinct projector for each block would incur substantial parameter and memory overhead. To mitigate this, we employ a single, *shared feature projector* $g(\cdot)$ *reused across all ViT blocks*. This significantly reduces the overall computational cost and enhances efficiency, while preserving prompt expert expressiveness relative to VPT. Furthermore, an independent LayerNorm (Lei Ba et al., 2016) is incorporated, we incorporate an independent LayerNorm (Lei Ba et al., 2016) before the token-wise projectors in each block. Regarding parameter complexity, VPT introduces approximately $L \times N_p \times d$ parameters across $L$ ViT blocks. In contrast, VAPT's learnable parameters include token-wise projectors, a channel-wise convolution layer per block, and a shared feature projector. The total parameter count is given by:

$$\underbrace{L \times N_p \times H' \times W'}_{\text{token-wise projectors}} + \underbrace{L \times K^2}_{\text{convolution}} + \underbrace{2 \times r \times d}_{\text{feature projector}}, \qquad (15)$$

where $H' \times W' < N$, with $K, r$ as small constants. Notably, for typical ViT configurations (*e.g.,* ViT-B/16, $N = 196$ and $d = 768$), $N$ is considerably smaller than $d$. Consequently, VAPT can achieve *greater parameter efficiency* than VPT, as empirically shown in Table 1 and Table 2. Moreover, VAPT introduces only *marginal computational overhead*, up to **0.6%** relative to VPT (see Appendix E.7).

**Novelty.** A central contribution of VAPT is its ability to achieve both flexibility and effectiveness while *preserving computational efficiency*. A common assumption is that increasing functional expressiveness necessarily incurs substantial computational or parameter overhead. As outlined in the above analysis, a naive implementation of a linear prompt expert would require a prohibitively large number of parameters due to the high dimensionality ($Nd$) of the input $\boldsymbol{X}$. In contrast, VAPT strikes a favorable balance between parameter efficiency and the expressivity of prompt experts. Despite the absence of a principled framework for designing adaptive prompts, VAPT's architectural choices of token-wise projectors and channel-wise convolutions yield a straightforward implementation and a compact, interpretable prompt-expert formulation. This architectural simplicity is particularly important, as it enables the rigorous theoretical analysis in Section 5, where we show that VAPT achieves an optimal sample-efficiency rate. Prior work often relies on more complex architectures or heuristic mechanisms to adapt prompts across inputs (Wang et al., 2022; Huang et al., 2023; Kim et al., 2023). While these designs can perform well empirically, their functional complexity typically precludes meaningful theoretical understanding. By contrast, VAPT is deliberately designed to retain a simple yet expressive functional form, leading to the clean expert formulations in Equations (13) and (14). These formulations support a thorough theoretical analysis without sacrificing empirical performance, addressing a key gap in the literature: VAPT not only performs well in practice but also enjoys provable robustness and generalization guarantees. Importantly, our approach leverages the *existing* MoE structure implicit in attention heads rather than imposing an additional, external MoE module, as is common in prior work (Zeng et al., 2025). Conventional MoE-based methods explicitly insert new routing components and expert modules into the Transformer architecture, introducing substantial architectural modifications. In contrast, VAPT avoids such intrusive changes to the backbone while still admitting a formulation that is amenable to formal analysis. This perspective allows us to design VAPT as a simple and practical mechanism that fully benefits from MoE theory to rigorously characterize its behavior. For a more detailed comparison with related methods, please refer to Appendix C.

## 5 STATISTICAL ADVANTAGES OF VAPT

In this section, we explore the theoretical advantages of VAPT through its MoE connection, as detailed in Equation (7). This perspective provides a rigorous framework for analyzing the convergence properties of prompt estimation in MoE models (Le et al., 2024; 2025b). Recalling from Section 3 that the MoE models $\tilde{\boldsymbol{h}}_{m,1}, \ldots, \tilde{\boldsymbol{h}}_{m,N}$ in each attention head share a common structure of experts and score functions. Furthermore, as noted in Section 2.1, omitting $\tilde{\boldsymbol{h}}_{m,N+1}, \ldots, \tilde{\boldsymbol{h}}_{m,N+N_p}$ does not affect ViT block output. Thus, to simplify our analysis while maintaining rigor, we focus on the first head (*i.e.,* $m = 1$), and specifically the first row of its attention matrix (*i.e.,* $i = 1$) in Equation (7). Within this simplified setting, we consider a regression framework for MoE models as follows.

**Problem Setup.** Assume that the i.i.d. samples of size $n$: $(\boldsymbol{X}_1, Y_1), (\boldsymbol{X}_2, Y_2), \ldots, (\boldsymbol{X}_n, Y_n) \in \mathbb{R}^d \times \mathbb{R}^{d'}$ are generated from the following regression model:

$$Y_i = f_{G_*}(\boldsymbol{X}_i) + \varepsilon_i, \quad i = 1, 2, \ldots, n, \tag{16}$$

where $\varepsilon_1, \varepsilon_2, \ldots, \varepsilon_n$ are independent Gaussian noise variables with $\mathbb{E}[\varepsilon_i | \boldsymbol{X}_i] = 0$ and $\text{Var}(\varepsilon_i | \boldsymbol{X}_i) = \nu^2 I_{d'}$ for $i \in [n]$. Furthermore, $\boldsymbol{X}_1, \boldsymbol{X}_2, \ldots, \boldsymbol{X}_n$ are assumed to be i.i.d. samples from a distribution $\mu$. The ground-truth regression function $f_{G_*}(\cdot)$ is an MoE model of the form

$$f_{G_*}(\boldsymbol{X}) := \sum_{j=1}^{N} \frac{\exp(\boldsymbol{X}^\top A_j^0 \boldsymbol{X} + a_j^0)}{D_{f,G_*}(\boldsymbol{X})} \cdot h(\boldsymbol{X}, \eta_j^0)$$

$$+ \sum_{j'=1}^{L} \frac{\exp((BW_{*,2}\sigma(W_{*,1j'}\boldsymbol{X}))^\top \boldsymbol{X} + b_{*,j'})}{D_f(\boldsymbol{X})} \cdot CW_{*,2}\sigma(W_{*,1j'}\boldsymbol{X}), \tag{17}$$

where $D_{f,G_*}(\boldsymbol{X}) = \sum_{k=1}^{N} \exp(\boldsymbol{X}^\top A_k^0 \boldsymbol{X} + a_k^0) + \sum_{j'=1}^{L} \exp((BW_{*,2}\sigma(W_{*,1j'}\boldsymbol{X}))^\top \boldsymbol{X} + b_{*,j'})$. Here, $G_* = \sum_{j'=1}^{L} \exp(b_{*,j'})\delta_{(W_{*,1j'}, W_{*,2})}$ denotes the true *mixing measure*, which is a weighted sum of Dirac measures $\delta$ associated with the unknown parameters $(b_{*,j'}, W_{*,1j'}, W_{*,2})_{j'=1}^{L}$ in the parameter space $\Theta \subset \mathbb{R} \times \mathbb{R}^{r \times d} \times \mathbb{R}^{d \times r}$. The matrix $A_j^0$, the expert parameter $\eta_j^0$, and the bias parameter $a_j^0$ are known for $j \in [N]$. Finally, the matrices $B \in \mathbb{R}^{d \times d}$ and $C \in \mathbb{R}^{d' \times d}$ are given; these play the role of pre-trained projection matrices in the context of prompt tuning.

**Least-Squares Estimator:** To estimate the unknown prompt parameters or, equivalently, the ground-truth mixing measure $G_*$, we use the least-squares method (van de Geer, 2000). In particular, we consider the estimator defined as follows:

$$\widehat{G}_n := arg\,min_{G \in \mathcal{G}_{L'}(\Theta)} \sum_{i=1}^{n} \left\| Y_i - f_G(\boldsymbol{X}_i) \right\|^2, \tag{18}$$

where $\mathcal{G}_{L'}(\Theta) := \{G = \sum_{i=1}^{\ell} \exp(b_i)\delta_{(W_{1,i}, W_2)} : \ell \in [L'], (b_i, W_{1,i}, W_2) \in \Theta\}$ is the set of all mixing measures with at most $L'$ atoms. Since the true number of experts $L$ is generally unknown, we assume the number of fitted experts $L'$ is sufficiently large, *i.e.,* $L' > L$. To analyze prompt estimation convergence, it is essential to define a suitable loss function on the prompt parameters. In this work, we propose the *Voronoi loss function*, derived from the concept of Voronoi cells (Manole & Ho, 2022).

**Voronoi Loss.** Given a mixing measure $G \in \mathcal{G}_{L'}(\Theta)$, we consider a Voronoi cell set $\{\mathcal{V}_j \equiv \mathcal{V}_j(G), j \in [L]\}$ generated by the atoms of $G_*$, where

$$\mathcal{V}_j := \{i \in [L'] : \|Z_i - Z_{*,j}\| \leq \|Z_i - Z_{*,\ell}\|, \forall \ell \neq j\},$$

where $Z_i := (W_{1,i}, W_2)$. The Voronoi loss tailored to the setting in Equation (17) is defined as:

$$\mathcal{D}_1(G, G_*) := \sum_{j'=1}^{L} \left| \sum_{i \in \mathcal{V}_{j'}} \exp(b_i) - \exp(b_{*,j'}) \right| + \sum_{j' \in [L]:|\mathcal{V}_{j'}|=1} \sum_{i \in \mathcal{V}_{j'}} \exp(b_i)(\|\Delta W_{1ij'}\| + \|\Delta W_2\|)$$

$$+ \sum_{j' \in [L]:|\mathcal{V}_{j'}|>1} \sum_{i \in \mathcal{V}_{j'}} \exp(b_i)(\|\Delta W_{1ij'}\|^2 + \|\Delta W_2\|^2),$$

Table 1: **Overall Comparison for ViT-B/16 Supervised Pre-trained on ImageNet-21K**. Following Jia et al. (2022), we report average accuracy (3 runs) on FGVC and VTAB-1K, "Number of Wins" [·] compared to full fine-tuning, and "Number of Wins over VPT" {·}. "Tuned/Total" denotes the average percentage of parameters tuned (24 tasks), "Scope" specifies the tuning scope, and "Additional parameters" indicates if parameters beyond pre-trained backbone/head are introduced. Per-task results are in Appendix E.1.

| ViT-B/16 (Dosovitskiy, 2020) (85.8M) | Tuned/ Total (%) | Scope Input | Scope Backbone | Extra params | FGVC [5] | VTAB-1K (Zhai et al., 2019) [19] *Natural* [7] | *Specialized* [4] | *Structured* [8] | Mean Total |
|---|---|---|---|---|---|---|---|---|---|
| Full (Iofinova et al., 2022) | 100.00 | | ✓ | | 88.54 | 75.88 | 83.36 | 47.64 | 65.57 |
| Linear (Iofinova et al., 2022) | 0.08 | | | | 79.32 [0] | 68.93 [1] | 77.16 [1] | 26.84 [0] | 52.94 |
| Partial-1 (Yosinski et al., 2014) | 8.34 | | | | 82.63 [0] | 69.44 [2] | 78.53 [0] | 34.17 [0] | 56.52 |
| MLP-3 (Chen et al., 2020) | 1.44 | | | ✓ | 79.80 [0] | 67.80 [2] | 72.83 [0] | 30.62 [0] | 53.21 |
| Sidetune (Zhang et al., 2020) | 10.08 | | ✓ | ✓ | 78.35 [0] | 58.21 [0] | 68.12 [0] | 23.41 [0] | 45.65 |
| Bias (Rebuffi et al., 2017) | 0.80 | | ✓ | | 88.41 [3] | 73.30 [3] | 78.25 [0] | 44.09 [2] | 62.05 |
| Adapter (Cai et al., 2020) | 1.02 | | ✓ | ✓ | 85.46 [1] | 70.67 [4] | 77.80 [0] | 33.09 [0] | 62.41 |
| LoRA (Hu et al., 2021) | 0.73 | | ✓ | ✓ | 89.46 [3] | 78.26 [5] | 83.78 [2] | 56.20 [7] | 72.25 |
| VPT-Shallow (Jia et al., 2022) | 0.16 | ✓ | | ✓ | 84.62 [1] | 76.81 [4] | 79.66 [0] | 46.98 [4] | 64.85 |
| VPT-Deep (Jia et al., 2022) | 0.73 | ✓ | | ✓ | 89.11 [4] | 78.48 [6] | 82.43 [2] | 54.98 [8] | 69.43 |
| E2VPT (Han et al., 2023) | 0.39 | ✓ | ✓ | ✓ | 89.22 [4] | 80.01 [6] | 84.43 [3] | 57.39 [8] | 71.42 |
| SA2VP (Pei et al., 2024) | 0.65 | ✓ | | ✓ | 90.08 [4] | 80.97 [6] | 85.73 [4] | 60.80 [8] | 73.47 |
| ViaPT (Xiao et al., 2025) | 0.66 | ✓ | | ✓ | 91.40 [4] | 82.62 [6] | 85.22 [2] | 61.25 [8] | 73.70 |
| VFPT (Zeng et al., 2024) | 0.66 | ✓ | | ✓ | 89.24 [4] | 81.35 [6] | 84.93 [4] | 60.19 [8] | 73.20 |
| **VAPT (Ours)** | **0.36** | ✓ | | ✓ | **89.58 [4] {4}** | **81.43 [6] {7}** | **85.13 [4] {4}** | **59.34 [8] {8}** | **72.91** |

where we denote $\Delta W_{1ij'} := W_{1i} - W_{*,1j'}$ for any $i, j'$, and $\Delta W_2 := W_2 - W_{*,2}$.

Equipped with this loss function, we wrap up the setting in Equation (17) by providing the convergence rate of prompt estimation in Theorem 1. For that purpose, it is necessary to make essential assumptions on the activation function $\sigma$. However, due to the space limitations, we defer these assumptions to the proof of Theorem 1 in Appendix A.1.

*Theorem* 1. Let $\widehat{G}_n$ be the least-squares estimator defined in Equation (18) and assume that the activation function $\sigma$ satisfies the Assumptions (A.1)-(A.3) specified in Appendix A.1, we obtain that

$$\mathcal{D}_1(\widehat{G}_n, G_*) = \mathcal{O}_P([\log(n)/n]^{\frac{1}{2}}).$$

**Implications for Prompt Estimation.** Given the formulation of the Voronoi loss function $\mathcal{D}_1$, Theorem 1 indicates that the estimation rates for the true parameters $(W_{*,1j}, W_{*,2})$ for indices $j$ such that $|\mathcal{V}_j| = 1$ are of parametric order $\mathcal{O}_P([\log(n)/n]^{\frac{1}{2}})$. Due to the Lipschitz property of the activation function $\sigma$, this directly leads to an estimation rate of $\mathcal{O}_P([\log(n)/n]^{\frac{1}{2}})$ for the true prompt $\boldsymbol{P}_j^*(\boldsymbol{X}) = W_{*,2}\sigma(W_{*,1j}\boldsymbol{X})$. On the other hand, for true parameters $(W_{*,1j}, W_{*,2})$ where $|\mathcal{V}_j| > 1$, their estimation rates are of the order $\mathcal{O}_P([\log(n)/n]^{1/4})$, which yields an estimation rate of $\mathcal{O}_P([\log(n)/n]^{1/4})$ for true prompt $\boldsymbol{P}_j^*(\boldsymbol{X}) = W_{*,2}\sigma(W_{*,1j}\boldsymbol{X})$. Finally, all these rates are optimal, up to the logarithmic factor, demonstrating the statistical benefits of visual adaptive prompt tuning for the non-linear setting of the activation function $\sigma$.

**Linear Activation Setting.** For completeness, we also show that the visual adaptive prompt tuning achieves optimal sample efficiency when the activation function $\sigma(\cdot)$ is linear identity in Appendix B.

## 6 EXPERIMENT

In this section, we compare VAPT with VPT and other widely used PEFT methods. We also examine the robustness of VAPT under different pre-training objectives and present our findings on its sample efficiency. For additional results, including ablation studies, semantic segmentation results, statistical significance tests, computational cost, and interpretive visualizations, please refer to Appendix E.

### 6.1 EXPERIMENTAL SETUP

**Datasets.** We evaluate VAPT on two benchmarks: FGVC and VTAB-1K (Zhai et al., 2019). FGVC includes five fine-grained classification datasets: CUB (Wah et al., 2011), Oxford Flowers (Nilsback & Zisserman, 2008), Stanford Cars (Gebru et al., 2017), Stanford Dogs (Khosla et al., 2011), and NABirds (Van Horn et al., 2015) that require distinguishing between visually similar classes. VTAB-1K comprises 19 datasets, each with 1,000 training examples, grouped into: *Natural* (images captured by standard cameras), *Specialized* (images collected via specialized equipment), and *Structured* (tasks requiring structural understanding, such as 3D depth prediction). Beyond classification, we also assess performance on semantic segmentation using ADE20K (Zhou et al., 2019).

Table 2: **Comparison of Different Pre-training Objectives.** We consider MAE (He et al., 2022) and MoCo v3 (Chen et al., 2021) using ViT-B/16. We report test accuracy on VTAB-1K, the "Number of Wins" [·] relative to full fine-tuning and the "Number of Wins over VPT" {·}. "Tuned/Total" denotes the average percentage of parameters tuned. Per-task results are in Appendix E.2.

| Pre-trained objectives | | MAE (He et al., 2022) | | | | MoCo v3 (Chen et al., 2021) | | |
|---|---|---|---|---|---|---|---|---|
| Params & Data
Methods | Tuned/
Total (%) | VTAB-1K (Zhai et al., 2019) [19]
*Natural* [7] | *Specialized* [4] | *Structured* [8] | Tuned/
Total (%) | VTAB-1K (Zhai et al., 2019) [19]
*Natural* [7] | *Specialized* [4] | *Structured* [8] |
| Full (Iofinova et al., 2022) | 100.00 | 59.31 | 79.68 | 53.82 | 100.00 | 71.95 | 84.72 | 51.98 |
| Linear (Iofinova et al., 2022) | 0.04 | 18.87 [0] | 53.72 [0] | 23.70 [0] | 0.04 | 67.46 [4] | 81.08 [0] | 30.33 [0] |
| Partial-1 (Yosinski et al., 2014) | 8.30 | 58.44 [5] | 78.28 [1] | 47.64 [1] | 8.30 | 72.31 [5] | 84.58 [2] | 47.89 [1] |
| Bias (Rebuffi et al., 2017) | 0.16 | 54.55 [1] | 75.68 [1] | 47.70 [0] | 0.16 | 72.89 [3] | 81.14 [0] | 53.43 [4] |
| Adapter (Cai et al., 2020) | 0.87 | 54.90 [3] | 75.19 [1] | 38.98 [0] | 1.12 | 74.19 [4] | 82.66 [1] | 47.69 [2] |
| VPT-Shallow (Jia et al., 2022) | 0.05 | 39.96 [1] | 69.65 [0] | 27.50 [0] | 0.06 | 67.34 [3] | 82.26 [0] | 37.55 [0] |
| VPT-Deep (Jia et al., 2022) | 0.31 | 36.02 [0] | 60.61 [1] | 26.57 [0] | 0.22 | 70.27 [4] | 83.04 [0] | 42.38 [0] |
| GateVPT (Yoo et al., 2023) | 0.05 | 47.61 [2] | 76.86 [1] | 36.80 [1] | 0.06 | 74.84 [4] | 83.38 [1] | 49.10 [3] |
| E2VPT (Han et al., 2023) | 0.07 | 59.52 [4] | 77.80 [1] | 44.65 [3] | 0.13 | 76.47 [4] | 87.28 [2] | 54.91 [6] |
| ViaPT (Xiao et al., 2025) | 0.36 | 54.26 [-] | 78.01 [-] | 37.52 [-] | 0.30 | 79.12 [-] | 86.81 [-] | 60.05 [-] |
| VFPT (Zeng et al., 2024) | 0.38 | 53.59 [6] | 77.75 [1] | 36.15 [1] | 0.22 | 77.47 [5] | 85.76 [3] | 58.74 [6] |
| **VAPT (Ours)** | **0.28** | **59.23 [5] {7}** | **80.73 [2] {3}** | **47.24 [2] {7}** | **0.27** | **77.69 [6] {7}** | **83.95 [2] {3}** | **60.74 [7] {8}** |

**Baselines.** In line with prior work (Jia et al., 2022; Han et al., 2023), we compare VAPT against commonly used PEFT methods. All classification experiments use standard Vision Transformer (ViT) (Dosovitskiy, 2020) that are supervised pre-trained on ImageNet-21K (Deng et al., 2009). For semantic segmentation, we employ SETR-PUP (Zheng et al., 2021), which uses ViT as the backbone encoder. Additionally, we evaluate VAPT with backbones pre-trained using two self-supervised learning: MAE (He et al., 2022) and MoCo v3 (Chen et al., 2021).

**Training.** We perform a grid search on `val` set of each task to determine the optimal learning rate, weight decay, kernel size $K$, and projector dimension $r$. We schedule the learning rate using a cosine decay schedule and train all models for 100 epochs. Following Jia et al. (2022), we use batch sizes of 64 and 128. Additional implementation details can be found in Appendix D.

## 6.2 EMPIRICAL RESULTS

**Overall Comparison.** Table 1 compares VAPT against full fine-tuning and other prominent PEFT methods on VTAB-1K and FGVC. Full denotes full fine-tuning, which updates all model parameters, while methods such as Linear, Partial-1 (top layer), and MLP-3 (3 MLP layers) modify only a subset of parameters. Sidetune (Zhang et al., 2020), Bias (Rebuffi et al., 2017), Adapter (Cai et al., 2020), and LoRA (Hu et al., 2021) introduce trainable modules for adaptation. Concurrent visual prompt tuning approaches, VPT (Jia et al., 2022), E2VPT (Han et al., 2023), SA2VP (Pei et al., 2024), ViaPT (Xiao et al., 2025) and VFPT (Zeng et al., 2024), are also included for comparison (see Appendix C for further details). Our results indicate that VAPT surpasses full fine-tuning in **22 out of 24** tasks. Specifically, VAPT achieves a notable **1.04%** accuracy increase on FGVC and a substantial **11.70%** improvement on VTAB-1K *Structured*. Across the entire VTAB-1K benchmark, VAPT demonstrates an average gain of **7.34%** over full fine-tuning, while updating merely **0.36%** of the backbone parameters. These findings highlight VAPT's significant *effectiveness* and *efficiency* as an innovative PEFT method. Additionally, VAPT achieves state-of-the-art performance compared to other PEFT approaches. Among prompt tuning methods, VAPT consistently outperforms VPT in **23 out of 24 tasks** and attains competitive performance with VFPT, a recent state-of-the-art approach, despite *utilizing nearly 50% fewer parameters*. We attribute these improvements to VAPT's design, which enhances the expressiveness of prompt experts. These results underscore VAPT's potential as *a powerful tool for improving performance with significantly reduced parameter overhead*.

**Different Pre-training Methods.** We investigate VAPT's performance when initialized with backbones pre-trained using different self-supervised learning (SSL) objectives, specifically MAE (He et al., 2022) and MoCo v3 (Chen et al., 2021), with detailed results in Table 2. Previous studies (Jia et al., 2022; Yoo et al., 2023) have indicated that VPT can exhibit suboptimal performance with SSL pre-trained backbones. In contrast, VAPT achieves significant performance improvements by effectively leveraging the rich information present in the input features. For example, VAPT demonstrates a remarkable **23.21%** accuracy improvement under MAE on VTAB-1K *Natural* tasks and an **18.36%** improvement with MoCo v3 on VTAB-1K *Structured*. Furthermore, when compared to other PEFT methods, VAPT consistently outperforms them, achieving the **highest** "Number of

Wins" relative to full fine-tuning, with **9 out of 19** tasks under MAE and **15 out of 19** tasks under MoCo v3. In addition, VAPT outperforms state-of-the-art approaches such as VFPT under MAE and remains competitive under MoCo v3. These results underscore the *generality* and *robustness* of our method across different pre-training objectives, supported by both theoretical analysis (Section 5) and empirical evidence.

**Sample Efficiency.** To empirically validate the theoretical sample efficiency from Section 5, we conducted experiments on the Stanford Dogs dataset (Khosla et al., 2011). Following d'Ascoli et al. (2021), we subsampled each class by fraction $f = \{0.01, 0.1, 0.3, 0.5, 1.0\}$ and scaled the number of training epochs by $1/f$, thereby ensuring a constant total number of image presentations to the model. The results, presented in Figure 2 and Table 3, demonstrate that VAPT consistently outperforms VPT across all evaluated training set sizes. Notably, with only 1% of the data, VPT achieves an accuracy of **3.6%**, whereas VAPT attains a substantially higher accuracy of **60.1%**. Furthermore, VAPT requires only 30% of the data to match the performance of VPT trained on the full 100% dataset, signifying an approximate $3\times$ reduction in data requirements. This underscores the superior sample efficiency of our approach.

It is important to note that the primary objective of this work is not to establish a new state-of-the-art on every dataset. As outlined in the abstract and introduction, our focus is on scholarly value rather than purely incremental methodological gains: we aim to provide the community with clear insights into the benefits of increasing the functional expressiveness of prompt experts within visual prompt tuning frameworks.

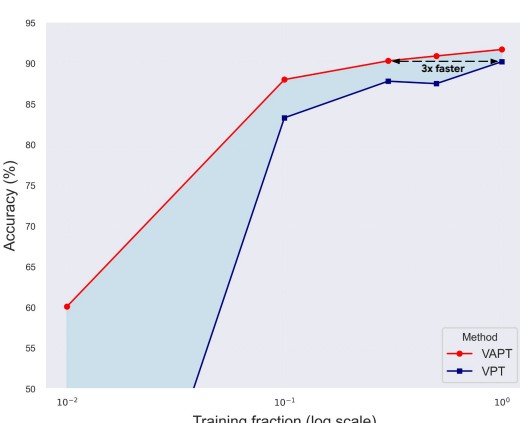

Figure 2: Comparison of VAPT and VPT across varying fractions of training data on Stanford Dogs.

Table 3: Classification accuracy on Stanford Dogs (Khosla et al., 2011) using varying fractions of training data. **Bold** indicates the best performance.

| Training fraction | VPT | VAPT | Gap |
|---|---|---|---|
| 1% | 3.6 | **60.1** | **56.5%** |
| 10% | 83.3 | **88.0** | 4.7% |
| 30% | 87.8 | **90.3** | 2.5% |
| 50% | 87.5 | **90.9** | 3.4% |
| 100% | 90.2 | **91.7** | 1.5% |

VAPT serves as a concrete instantiation of this idea. Its adaptive formulation illustrates the potential of adaptive prompt experts to deliver *improved performance*, *stronger parameter efficiency*, and *enhanced sample efficiency* in both theory and practice.

Within this perspective, the most relevant comparison is between VAPT's adaptive prompt-expert formulation and VPT, where prompt experts are constant functions. Our experiments consistently show that VAPT outperforms VPT on the majority of benchmarks while using a more compact set of trainable parameters. Moreover, our empirical findings quantify the statistical advantages of VAPT over VPT, and these observations are rigorously supported by our theoretical analysis in Section 5 and Appendix A. Together, these results reinforce our central claim that increasing the expressiveness of prompt experts is a principled and effective direction for advancing visual prompt tuning.

## 7 CONCLUSION

In this paper, we highlight the limited functional expressiveness of prompt experts in existing VPT formulation and introduce VAPT, a novel adaptive prompt design. Our approach achieves superior performance compared to VPT while using fewer parameters and demonstrates optimal sample efficiency both theoretically and empirically. Our theoretical analysis is grounded in interpreting self-attention as a composition of multiple MoE models. Given that self-attention is central to most Transformer architectures, we believe our analysis can naturally extend to other Transformer variants. While our experiments were conducted on Vision Transformers, the potential demonstrated by adaptive prompts suggests that future work could explore alternative designs across diverse Transformer architectures and broaden their applications to a wider range of tasks. Furthermore, future research could investigate the potential synergies between VAPT and VPT, for instance, by exploring their integration.

## REPRODUCIBILITY STATEMENT

In order to facilitate the reproduction of our empirical results, we provide detailed descriptions of the experimental setup in Section 6.1 and Appendix D. All datasets used in this study are publicly available, enabling full replication of our experiments.

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

# Supplement to "Revisit Visual Prompt Tuning: The Expressiveness of Prompt Experts"

In this supplementary material, we provide detailed proofs of the main results in Appendix A and additional theoretical findings in Appendix B. A discussion of related work is presented in Appendix C. Implementation details of our experiments are described in Appendix D, while additional experimental results are included in Appendix E. Specifically, these include detailed per-task results corresponding to the main experiments in Section 6 (Appendices E.1, E.2), statistical significance tests (Appendix E.3), an evaluation of VAPT performance across different backbone scales (Appendix E.4), and semantic segmentation results (Appendix E.5). Finally, ablation studies, computational cost analyses, adversarial robustness, and interpretive visualizations are presented in Appendix E.6, Appendix E.7, Appendix E.8, and Appendix E.9, respectively.

## A PROOFS

In this appendix, we provide proofs for key theoretical results on prompt estimation presented in the main text.

### A.1 PROOF OF THEOREM 1

*Proposition* 1. Given the least-squares estimator $\widehat{G}_n$ defined in Equation (18), under the $L_2(\mu)$ norm, the estimator $f_{\widehat{G}_n}(\cdot)$ converges to the true model $f_{G_*}(\cdot)$ at a parametric rate with respect to the sample size. That is,

$$\|f_{\widehat{G}_n} - f_{G_*}\|_{L_2(\mu)} = \mathcal{O}_P([\log(n)/n]^{\frac{1}{2}}). \tag{19}$$

The proof of Proposition 1 is provided in Appendix A.2. Building on the convergence rate established therein, we now aim to demonstrate the following inequality:

$$\inf_{G \in \mathcal{G}_{L'}(\Theta)} \|f_G - f_{G_*}\|_{L_2(\mu)}/\mathcal{D}_1(G, G_*) > 0. \tag{20}$$

We divide the proof of the above inequality into local and global parts presented in Appendices A.1.1 and A.1.2, respectively. Before presenting the proof details, let us introduce some essential assumptions on the activation function $\sigma$.

**Assumptions.** We impose the following assumptions on the activation function $\sigma$:

*(A.1) (Identifiability)* If there exist parameters $(W_1, W_2)$ and $(W_1', W_2')$ such that $W_2\sigma(W_1\boldsymbol{X}) = W_2'\sigma(W_1'\boldsymbol{X})$ holds for almost surely $\boldsymbol{X}$, then $(W_1, W_2) = (W_1', W_2')$.

*(A.2) (Uniform Lipschitz)* Let $F(\boldsymbol{X}; W_1, W_2) := \exp((BW_2\sigma(W_1\boldsymbol{X}))^\top \boldsymbol{X})CW_2\sigma(W_1\boldsymbol{X})$. Then, for any $\tau \in \{1, 2\}$, we have

$$\sum_{|\alpha|=\tau} \left|\left(\frac{\partial^{|\alpha|}F}{\partial W_1^{\alpha_1}\partial W_2^{\alpha_2}}(\boldsymbol{X}; W_1, W_2) - \frac{\partial^{|\alpha|}F}{\partial W_1^{\alpha_1}\partial W_2^{\alpha_2}}(\boldsymbol{X}; W_1', W_2')\right)\gamma^\alpha\right| \leq C\|(W_1, W_2) - (W_1', W_2')\|^\zeta\|\gamma\|^\tau,$$

for any vector $\gamma \in \mathbb{R}^{2dr}$ and for some positive constants $\zeta$ and $C$ that are independent of $\boldsymbol{X}$ and $(W_1, W_2), (W_1', W_2')$. Here, $\alpha = (\alpha_1, \alpha_2) \in \mathbb{N}^{r \times d} \times \mathbb{N}^{d \times r}$.

*(A.3) (Strong identifiability)* The function $\sigma$ is twice differentiable almost surely. For any natural number $\ell$ and distinct parameters $\{W_{1,j} : j \in [\ell]\}$, the functions in the set

$$\Big\{\sigma(W_{1,j}\boldsymbol{X}), \sigma^2(W_{1,j}\boldsymbol{X})\boldsymbol{X}^{(u)}, \sigma^{(1)}(W_{1,j}\boldsymbol{X})\boldsymbol{X}^{(u)}, \sigma^{(1)}(W_{1,j}\boldsymbol{X})\sigma(W_{1,j}\boldsymbol{X})\boldsymbol{X}^{(u)}\boldsymbol{X}^{(v)},$$

$$[\sigma^{(1)}(W_{1,j}\boldsymbol{X})]^2\sigma(W_{1,j}\boldsymbol{X})\boldsymbol{X}^{(u)}\boldsymbol{X}^{(v)}\boldsymbol{X}^{(w)}, \sigma^{(1)}(W_{1,j}\boldsymbol{X})\sigma^2(W_{1,j}\boldsymbol{X})\boldsymbol{X}^{(u)}\boldsymbol{X}^{(v)}\boldsymbol{X}^{(w)},$$

$$[\sigma^{(1)}(W_{1,j}\boldsymbol{X})]^2\boldsymbol{X}^{(u)}\boldsymbol{X}^{(v)}\boldsymbol{X}^{(w)}, \sigma^{(2)}(W_{1,j}\boldsymbol{X})\boldsymbol{X}^{(u)}, \sigma^{(2)}(W_{1,j}\boldsymbol{X})\boldsymbol{X}^{(u)}\boldsymbol{X}^{(v)},$$

$$\sigma^{(2)}(W_{1,j}\boldsymbol{X})\boldsymbol{X}^{(u)}\boldsymbol{X}^{(v)}\boldsymbol{X}^{(w)} : j \in [\ell], u, v, w \in [d]\Big\}$$

are linearly independent for almost surely $\boldsymbol{X}$. Here, $\sigma^{(1)}$ and $\sigma^{(2)}$ denote the first and second derivatives of $\sigma$, respectively, which are applied element-wise to the matrices $W_{1,j}\boldsymbol{X}$.

### A.1.1 LOCAL PART

The local part of the inequality (20) corresponds to the following inequality:

$$\lim_{\varepsilon \to 0} \inf_{G \in \mathcal{G}_{L'}(\Theta):\mathcal{D}_1(G,G_*) \leq \varepsilon} \frac{\|f_G - f_{G_*}\|_{L_2(\mu)}}{\mathcal{D}_1(G,G_*)} > 0.$$

We assume, for the sake of contradiction, that the above inequality does not hold. Then, we can find a sequence of measures $G_n := \sum_{j'=1}^{L'} \exp(b_{n,j'}) \delta_{(W_{n,1j'}, W_{n,2})}$ in $\mathcal{G}_{L'}(\Theta)$ such that

$$\begin{cases} \mathcal{D}_{1n} := \mathcal{D}_1(G_n, G_*) \to 0, \\ \|f_{G_n} - f_{G_*}\|_{L_2(\mu)}/\mathcal{D}_{1n} \to 0. \end{cases}$$

For the sake of the presentation, $\mathcal{V}_j^n := \mathcal{V}_j(G_n)$ is denoted as a Voronoi cell of $G_n$ generated by the $j$-th components of the true measure $G_*$. Given that the ensuing arguments are asymptotic, without loss of generality we assume that those Voronoi cells do not depend on the sample size, *i.e.*, we have $\mathcal{V}_j = \mathcal{V}_j^n$ for all $n$ and $1 \leq j \leq L$. Hence, the Voronoi loss $\mathcal{D}_{1n}$ can be rewritten as follows:

$$\mathcal{D}_{1n} := \sum_{j'=1}^{L} \Big| \sum_{i \in \mathcal{V}_{j'}} \exp(b_{n,i}) - \exp(b_{*,j'}) \Big| + \sum_{j' \in [L]:|\mathcal{V}_{j'}|=1} \sum_{i \in \mathcal{V}_{j'}} \exp(b_{n,i})(\|\Delta W_{n,1ij'}\| + \|\Delta W_{n,2}\|)$$

$$+ \sum_{j' \in [L]:|\mathcal{V}_{j'}|>1} \sum_{i \in \mathcal{V}_{j'}} \exp(b_{n,i})(\|\Delta W_{n,1ij'}\|^2 + \|\Delta W_{n,2}\|^2)$$

where we define $\Delta W_{n,1ij'} = W_{n,1i} - W_{*,1j'}$ and $\Delta W_{n,2} = W_{n,2} - W_{*,2}$ for all $i \in \mathcal{V}_{j'}$.

From the hypothesis, since $\mathcal{D}_{1n} \to 0$ as $n \to \infty$, we have $\sum_{i \in \mathcal{V}_j} \exp(b_{n,i}) \to \exp(b_{*,j})$, $W_{n,1i} \to W_{*,1j'}$, and $W_{n,2} \to W_{*,2}$ for any $i \in \mathcal{V}_j, j \in [L]$. Throughout this proof, for simplicity of argument we assume without loss of generality that $B = I_d$, $C = I_d$, and $r = 1$. We note that our techniques can be generalized to the general case of these given matrices. Now, we divide the proof of the local part into the following three main substeps:

**Step 1 - Taylor expansion.** We first define the following function:

$$Q_n(\boldsymbol{X}) := \Big[ \sum_{j=1}^{N} \exp(\boldsymbol{X}^\top A_j^0 \boldsymbol{X} + a_j^0) + \sum_{j'=1}^{L} \exp((W_{*,2}\sigma(W_{*,1j'}\boldsymbol{X}))^\top \boldsymbol{X} + b_{*,j'}) \Big] \cdot [f_{G_n}(\boldsymbol{X}) - f_{G_*}(\boldsymbol{X})].$$

Then, we can decompose the function $Q_n(\boldsymbol{X})$ as follows:

$$Q_n(\boldsymbol{X}) = \sum_{j=1}^{L} \sum_{i \in \mathcal{V}_j} \exp(b_{n,i}) \Big[ \exp((W_{n,2}\sigma(W_{n,1i}\boldsymbol{X}))^\top \boldsymbol{X}) W_{n,2}\sigma(W_{n,1i}\boldsymbol{X})$$

$$- \exp((W_{*,2}\sigma(W_{*,1j}\boldsymbol{X}))^\top \boldsymbol{X}) W_{*,2}\sigma(W_{*,1j}\boldsymbol{X}) \Big]$$

$$- \sum_{j=1}^{L} \sum_{i \in \mathcal{V}_j} \exp(b_{n,i}) \Big[ \exp((W_{n,2}\sigma(W_{n,1i}\boldsymbol{X}))^\top \boldsymbol{X}) - \exp((W_{*,2}\sigma(W_{*,1j}\boldsymbol{X}))^\top \boldsymbol{X}) \Big] f_{G_n}(\boldsymbol{X})$$

$$+ \sum_{j=1}^{L} \Big( \sum_{i \in \mathcal{V}_j} \exp(b_{n,i}) - \exp(b_{*,j}) \Big) \exp((W_{*,2}\sigma(W_{*,1j}\boldsymbol{X}))^\top \boldsymbol{X}) \Big[ W_{*,2}\sigma(W_{*,1j}\boldsymbol{X}) - f_{G_n}(\boldsymbol{X}) \Big]$$

$$:= A_n(\boldsymbol{X}) - B_n(\boldsymbol{X}) + C_n(\boldsymbol{X}). \tag{21}$$

**Decomposition of the function $A_n(\boldsymbol{X})$.** To streamline the argument, we define the following functions: $\bar{E}(\boldsymbol{X}; W_1, W_2) := \exp((W_2\sigma(W_1\boldsymbol{X}))^\top \boldsymbol{X})$ and $\bar{H}(\boldsymbol{X}; W_1, W_2) = W_2\sigma(W_1\boldsymbol{X})$. Furthermore, the product of these functions is defined as $\bar{F}(\boldsymbol{X}; W_1, W_2) = \bar{E}(\boldsymbol{X}; W_1, W_2)\bar{H}(\boldsymbol{X}; W_1, W_2)$. To account for the difference in the number of elements among Voronoi cells, we further decompose

the function $A_n(\boldsymbol{X})$ as follows:

$$
\begin{aligned}
A_n(\boldsymbol{X}) = & \sum_{j:|\mathcal{V}_j|=1} \sum_{i \in \mathcal{V}_j} \exp(b_{n,i}) \Big[ \bar{F}(\boldsymbol{X}; W_{n,1i}, W_{n,2}) - \bar{F}(\boldsymbol{X}; W_{*,1j}, W_{*,2}) \Big] \\
& + \sum_{j:|\mathcal{V}_j|>1} \sum_{i \in \mathcal{V}_j} \exp(b_{n,i}) \Big[ \bar{F}(\boldsymbol{X}; W_{n,1i}, W_{n,2}) - \bar{F}(\boldsymbol{X}; W_{*,1j}, W_{*,2}) \Big] \\
:= & A_{n,1}(\boldsymbol{X}) + A_{n,2}(\boldsymbol{X})
\end{aligned}
$$

An application of the first-order Taylor expansion leads to:

$$
\begin{aligned}
\bar{E}(\boldsymbol{X}; W_{n,1i}, W_{n,2}) = & \bar{E}(\boldsymbol{X}; W_{*,1j}, W_{*,2}) + \sum_{|\alpha|=1} (\Delta W_{n,1ij})^{\alpha_1} (\Delta W_{n,2})^{\alpha_2} \frac{\partial^{|\alpha_1|+|\alpha_2|} \bar{E}}{\partial W_1^{\alpha_1} \partial W_2^{\alpha_2}}(\boldsymbol{X}; W_{*,1j}, W_{*,2}) \\
& + \bar{R}_{ij,1}(\boldsymbol{X}), \\
\bar{H}(\boldsymbol{X}; W_{n,1i}, W_{n,2}) = & \bar{H}(\boldsymbol{X}; W_{*,1j}, W_{*,2}) + \sum_{|\alpha|=1} (\Delta W_{n,1ij})^{\alpha_1} (\Delta W_{n,2})^{\alpha_2} \frac{\partial^{|\alpha_1|+|\alpha_2|} \bar{H}}{\partial W_1^{\alpha_1} \partial W_2^{\alpha_2}}(\boldsymbol{X}; W_{*,1j}, W_{*,2}) \\
& + \bar{R}_{ij,2}(\boldsymbol{X}).
\end{aligned}
$$

Here, the indices $i, j$ in the above equations satisfy $|\mathcal{V}_j| = 1$, *i.e.,* Voronoi cells with exactly one element and $i \in \mathcal{V}_j$. Furthermore, the functions $\bar{R}_{ij,1}(\boldsymbol{X})$ and $\bar{R}_{ij,2}(\boldsymbol{X})$ in these expressions represent the Taylor remainders from the expansions of the functions $\bar{E}$ and $\bar{H}$. Combining these results leads to:

$$
\begin{aligned}
A_{n,1}(\boldsymbol{X}) = & \sum_{j:|\mathcal{V}_j|=1} \sum_{i \in \mathcal{V}_j} \frac{\exp(b_{n,i})}{\alpha!} \sum_{|\alpha|=1} \Bigg\{ (\Delta W_{n,1ij})^{\alpha_1} (\Delta W_{n,2})^{\alpha_2} \frac{\partial^{|\alpha_1|+|\alpha_2|} \bar{E}}{\partial W_1^{\alpha_1} \partial W_2^{\alpha_2}}(\boldsymbol{X}; W_{*,1j}, W_{*,2}) \bar{H}(\boldsymbol{X}; W_{*,1j}, W_{*,2}) \\
& + (\Delta W_{n,1ij})^{\alpha_1} (\Delta W_{n,2})^{\alpha_2} \frac{\partial^{|\alpha_1|+|\alpha_2|} \bar{H}}{\partial W_1^{\alpha_1} \partial W_2^{\alpha_2}}(\boldsymbol{X}; W_{*,1j}, W_{*,2}) \bar{E}(\boldsymbol{X}; W_{*,1j}, W_{*,2}) \Bigg\} + \tilde{R}_{n,1}(\boldsymbol{X}) \\
= & \sum_{j:|\mathcal{V}_j|=1} \sum_{|\alpha|=1} \Bigg\{ \bar{M}_{n,j,\alpha_1,\alpha_2} \frac{\partial^{|\alpha_1|+|\alpha_2|} \bar{E}}{\partial W_1^{\alpha_1} \partial W_2^{\alpha_2}}(\boldsymbol{X}; W_{*,1j}, W_{*,2}) \bar{H}(\boldsymbol{X}; W_{*,1j}, W_{*,2}) \\
& + \bar{M}_{n,j,\alpha_1,\alpha_2} \frac{\partial^{|\alpha_1|+|\alpha_2|} \bar{H}}{\partial W_1^{\alpha_1} \partial W_2^{\alpha_2}}(\boldsymbol{X}; W_{*,1j}, W_{*,2}) \bar{E}(\boldsymbol{X}; W_{*,1j}, W_{*,2}) \Bigg\} + \tilde{R}_{n,1}(\boldsymbol{X})
\end{aligned}
$$

where $\alpha = (\alpha_1, \alpha_2)$. Furthermore, due to the uniform smoothness of the functions $\bar{E}$ and $\bar{H}$, the function $\tilde{R}_{n,1}(\boldsymbol{X})$ in the above equation satisfies $\tilde{R}_{n,1}(\boldsymbol{X})/\mathcal{D}_{1n} \to 0$ when $n$ approaches infinity. Finally, the terms $M_{n,j,\alpha_1,\alpha_2}$ in this equation admit the following forms:

$$
\bar{M}_{n,j,\alpha_1,\alpha_2} = \sum_{i \in \mathcal{V}_j} \frac{\exp(b_{n,i})}{\alpha!} (\Delta W_{n,1ij})^{\alpha_1} (\Delta W_{n,2})^{\alpha_2},
$$

for any $|\alpha| = 1$.

We now proceed to decompose the function $A_{n,2}(\boldsymbol{X})$. Unlike the function $A_{n,1}(\boldsymbol{X})$ for which we utilized only a first-order Taylor expansion, the analysis of $A_{n,2}(\boldsymbol{X})$ must account for Voronoi cells potentially containing more than one element. Consequently, we employ second-order Taylor expansions of the functions $\bar{E}$ and $\bar{H}$. In particular, an application of the second-order Taylor expansion leads to:

$$
\begin{aligned}
A_{n,2}(\boldsymbol{X}) = & \sum_{j:|\mathcal{V}_j|>1} \sum_{1 \le |\alpha| \le 2} \Bigg\{ \bar{M}_{n,j,\alpha_1,\alpha_2} \frac{\partial^{|\alpha_1|+|\alpha_2|} \bar{E}}{\partial W_1^{\alpha_1} \partial W_2^{\alpha_2}}(\boldsymbol{X}; W_{*,1j}, W_{*,2}) \bar{H}(\boldsymbol{X}; W_{*,1j}, W_{*,2}) \\
& + \bar{M}_{n,j,\alpha_1,\alpha_2} \frac{\partial^{|\alpha_1|+|\alpha_2|} \bar{H}}{\partial W_1^{\alpha_1} \partial W_2^{\alpha_2}}(\boldsymbol{X}; W_{*,1j}, W_{*,2}) \bar{E}(\boldsymbol{X}; W_{*,1j}, W_{*,2}) \Bigg\} \\
& + \sum_{|\alpha|=1, |\beta|=1} \bar{M}_{n,j,\alpha,\beta} \frac{\partial^{|\alpha_1|+|\alpha_2|} \bar{E}}{\partial W_1^{\alpha_1} \partial W_2^{\alpha_2}}(\boldsymbol{X}; W_{*,1j}, W_{*,2}) \frac{\partial^{|\beta_1|+|\beta_2|} \bar{H}}{\partial W_1^{\beta_1} \partial W_2^{\beta_2}}(\boldsymbol{X}; W_{*,1j}, W_{*,2}) + \tilde{R}_{n,2}(\boldsymbol{X})
\end{aligned}
$$

where $\alpha = (\alpha_1, \alpha_2)$, $\beta = (\beta_1, \beta_2)$. Furthermore, due to the uniform smoothness of the functions $\bar{E}$ and $\bar{H}$, the function $\tilde{R}_{n,2}(\boldsymbol{X})$, which represents the combination of Taylor remainders from the second-order Taylor expansion, satisfies $\bar{R}_{n,2}(\boldsymbol{X})/\mathcal{D}_{1n} \to 0$ as $n$ goes to infinity. Additionally, the coefficients $\bar{M}_{n,j,\alpha_1,\alpha_2}$ and $\bar{M}_{n,j,\alpha_1,\alpha_2,\beta_1,\beta_2}$ in the above formulation take the following forms:

$$\bar{M}_{n,j,\alpha_1,\alpha_2} = \sum_{i \in \mathcal{V}_j} \frac{\exp(b_{n,i})}{\alpha!}(\Delta W_{n,1ij})^{\alpha_1}(\Delta W_{n,2})^{\alpha_2},$$

for any multi-index $\alpha$ such that $|\alpha| = 2$ and

$$\bar{M}_{n,j,\alpha_1,\alpha_2,\beta_1,\beta_2} = \sum_{i \in \mathcal{V}_j} \frac{\exp(b_{n,i})}{\alpha!\beta!}(\Delta W_{n,1ij})^{\alpha_1+\beta_1}(\Delta W_{n,2})^{\alpha_2+\beta_2},$$

for any coefficients $\alpha$ and $\beta$ such that $|\alpha| = |\beta| = 1$. Given the expressions for the functions $\bar{E}(\boldsymbol{X}; W_1, W_2)$ and $\bar{H}(\boldsymbol{X}; W_1, W_2)$, their partial derivatives take the following forms:

$$\frac{\partial \bar{E}}{\partial W_1^{(u)}}(\boldsymbol{X}; W_1, W_2) = \exp((W_2\sigma(W_1\boldsymbol{X}))^\top \boldsymbol{X})\boldsymbol{X}^\top W_2\sigma^{(1)}(W_1\boldsymbol{X})\boldsymbol{X}^{(u)},$$

$$\frac{\partial \bar{E}}{\partial W_2^{(v)}}(\boldsymbol{X}; W_1, W_2) = \exp((W_2\sigma(W_1\boldsymbol{X}))^\top \boldsymbol{X})\boldsymbol{X}^{(v)}\sigma(W_1\boldsymbol{X}),$$

$$\frac{\partial^2 \bar{E}}{\partial W_1^{(u)}\partial W_1^{(v)}}(\boldsymbol{X}; W_1, W_2) = \exp((W_2\sigma(W_1\boldsymbol{X}))^\top \boldsymbol{X})[\boldsymbol{X}^\top W_2\sigma^{(1)}(W_1\boldsymbol{X})]^2\boldsymbol{X}^{(u)}\boldsymbol{X}^{(v)}$$
$$+ \exp((W_2\sigma(W_1\boldsymbol{X}))^\top \boldsymbol{X})\boldsymbol{X}^\top W_2\sigma^{(2)}(W_1\boldsymbol{X})\boldsymbol{X}^{(u)}\boldsymbol{X}^{(v)},$$

$$\frac{\partial^2 \bar{E}}{\partial W_2^{(u)}\partial W_2^{(v)}}(\boldsymbol{X}; W_1, W_2) = \exp((W_2\sigma(W_1\boldsymbol{X}))^\top \boldsymbol{X})\boldsymbol{X}^{(u)}\boldsymbol{X}^{(v)}\sigma^2(W_1\boldsymbol{X}),$$

$$\frac{\partial^2 \bar{E}}{\partial W_1^{(u)}\partial W_2^{(v)}}(\boldsymbol{X}; W_1, W_2) = \exp((W_2\sigma(W_1\boldsymbol{X}))^\top \boldsymbol{X})\boldsymbol{X}^\top W_2\sigma^{(1)}(W_1\boldsymbol{X})\boldsymbol{X}^{(u)}\boldsymbol{X}^{(v)}\sigma(W_1\boldsymbol{X})$$
$$+ \exp((W_2\sigma(W_1\boldsymbol{X}))^\top \boldsymbol{X})\sigma^{(1)}(W_1\boldsymbol{X})\boldsymbol{X}^{(u)}\boldsymbol{X}^{(v)},$$

$$\frac{\partial \bar{H}}{\partial W_1^{(u)}}(\boldsymbol{X}; W_1, W_2) = W_2\sigma^{(1)}(W_1\boldsymbol{X})\boldsymbol{X}^{(u)},$$

$$\frac{\partial \bar{H}}{\partial W_2}(\boldsymbol{X}; W_1, W_2) = \sigma(W_1\boldsymbol{X})I_d,$$

$$\frac{\partial^2 \bar{H}}{\partial W_1^{(u)}\partial W_1^{(v)}}(\boldsymbol{X}; W_1, W_2) = W_2\sigma^{(2)}(W_1\boldsymbol{X})\boldsymbol{X}^{(u)}\boldsymbol{X}^{(v)},$$

$$\frac{\partial^2 \bar{H}}{\partial W_1^{(u)}\partial W_2}(\boldsymbol{X}; W_1, W_2) = \sigma^{(2)}(W_1\boldsymbol{X})\boldsymbol{X}^{(u)}I_d,$$

$$\frac{\partial^2 \bar{H}}{\partial W_2^{(u)}\partial W_2^{(v)}}(\boldsymbol{X}; W_1, W_2) = \boldsymbol{0}.$$

Putting the above formulations together, the functions $A_{n,1}(\boldsymbol{X})$ and $A_{n,2}(\boldsymbol{X})$ can be rewritten as follows:

$$
\begin{aligned}
A_{n,1}(\boldsymbol{X}) = & \sum_{j:|\mathcal{V}_j|=1} \exp((W_{*,2}\sigma(W_{*,1j}\boldsymbol{X}))^\top \boldsymbol{X}) \big[ \sigma^{(1)}(W_{*,1j}\boldsymbol{X})\sigma(W_{*,1j}\boldsymbol{X})\boldsymbol{X}^\top L_{n,1,j}\boldsymbol{X}W_{*,2} \\
& + \sigma^2(W_{*,1j}\boldsymbol{X})L_{n,2,j}^\top \boldsymbol{X}W_{*,2} + \sigma^{(1)}(W_{*,1j}\boldsymbol{X})L_{n,3,j}^\top \boldsymbol{X}W_{*,2} + \sigma(W_{*,1j}\boldsymbol{X})L_{n,2,j}^\top \big] + \bar{R}_{n,1}(\boldsymbol{X}), \\
A_{n,2}(\boldsymbol{X}) = & \sum_{j:|\mathcal{V}_j|>1} \exp((W_{*,2}\sigma(W_{*,1j}\boldsymbol{X}))^\top \boldsymbol{X}) \big[ \sigma^{(1)}(W_{*,1j}\boldsymbol{X})\sigma(W_{*,1j}\boldsymbol{X})\boldsymbol{X}^\top L_{n,1,j}\boldsymbol{X}W_{*,2} \\
& + \sigma^2(W_{*,1j}\boldsymbol{X})L_{n,2,j}^\top \boldsymbol{X}W_{*,2} + \sigma^{(1)}(W_{*,1j}\boldsymbol{X})L_{n,3,j}^\top \boldsymbol{X}W_{*,2} + \sigma(W_{*,1j}\boldsymbol{X})L_{n,2,j}^\top \\
& + \big( [\sigma^{(1)}(W_{*,1j}\boldsymbol{X})]^2 \sigma(W_{*,1j}\boldsymbol{X}) + \sigma^{(2)}(W_{*,1j}\boldsymbol{X})\sigma(W_{*,1j}\boldsymbol{X}) \big)\boldsymbol{X}^\top L_{n,4,j}\boldsymbol{X}\boldsymbol{X}^\top W_{*,2}W_{*,2} \\
& + \sigma^2(W_{*,1j}\boldsymbol{X})\sigma(W_{*,1j}\boldsymbol{X})\boldsymbol{X}^\top L_{n,5,j}\boldsymbol{X}W_{*,2} + \sigma^{(2)}(W_{*,1j}\boldsymbol{X})\boldsymbol{X}^\top L_{n,4,j}\boldsymbol{X}W_{*,2} \\
& + \big( \sigma^{(1)}(W_{*,1j}\boldsymbol{X})[\sigma(W_{*,1j}\boldsymbol{X})]^2 \boldsymbol{X}^\top W_{*,2} + \sigma^{(1)}(W_{*,1j}\boldsymbol{X})\sigma(W_{*,1j}\boldsymbol{X}) \big)\boldsymbol{X}^\top L_{n,6,j}\boldsymbol{X}W_{*,2} \\
& + \sigma^{(2)}(W_{*,1j}\boldsymbol{X})\boldsymbol{X}^\top L_{n,6,j} + [\sigma^{(1)}(W_{*,1j}\boldsymbol{X})]^2 \boldsymbol{X}^\top L_{n,4,j}\boldsymbol{X}\boldsymbol{X}^\top W_{*,2}W_{*,2} \\
& + \sigma^{(1)}(W_{*,1j}\boldsymbol{X})\sigma(W_{*,1j}\boldsymbol{X})\boldsymbol{X}^\top W_{*,2}\boldsymbol{X}^\top L_{n,6,j} + \sigma^{(1)}(W_{*,1j}\boldsymbol{X})\sigma(W_{*,1j}\boldsymbol{X})\boldsymbol{X}^\top L_{n,6,j}\boldsymbol{X}W_{*,2} \\
& + [\sigma(W_{*,1j}\boldsymbol{X})]^2 \boldsymbol{X}^\top L_{n,5,j} \big] + \bar{R}_{n,2}(\boldsymbol{X}),
\end{aligned}
$$

where the formulations of $L_{n,1,j}, L_{n,2,j}, \dots, L_{n,6,j}$ in these equations are given by:

$$
\begin{aligned}
L_{1,n} & := M_{n,j,e_{1:d},\mathbf{0}_d}W_{*,2}^\top, \quad M_{n,j,e_{1:d},\mathbf{0}_d} := (M_{n,j,e_u,\mathbf{0}_d})_{u=1}^d \\
L_{n,2,j} & := (M_{n,j,\mathbf{0}_d,e_u})_{u=1}^d, \\
L_{n,3,j} & := M_{n,j,e_{1:d},\mathbf{0}_d}, \\
L_{n,4,j} & := (M_{n,j,e_u+e_v,\mathbf{0}_d})_{u,v=1}^d, \\
L_{n,5,j} & := (M_{n,j,\mathbf{0}_d,e_u+e_v})_{u,v=1}^d, \\
L_{n,6,j} & := (M_{n,j,e_u,e_v})_{u,v=1}^d,
\end{aligned}
$$

Here, for each $1 \le u \le d$, $e_u$ denotes the standard basis vector in $\mathbb{R}^d$ with 1 in the $u$-th position and 0 in all other positions.

**Decomposition of the function $B_n(\boldsymbol{X})$.** Similar to our decomposition of the function $A_n(\boldsymbol{X})$, in decomposing the function $B_n(\boldsymbol{X})$, we also distinguish between Voronoi cells containing exactly one element and those containing more than one element. Therefore, we obtain:

$$
\begin{aligned}
B_n(\boldsymbol{X}) = & \sum_{j:|\mathcal{V}_j|=1} \sum_{i \in \mathcal{V}_j} \exp(b_{n,i}) \Big[ \bar{E}(\boldsymbol{X}; W_{n,1i}, W_{n,2}) - \bar{E}(\boldsymbol{X}; W_{*,1j}, W_{*,2}) \Big] f_{G_n}(\boldsymbol{X}) \\
& + \sum_{j:|\mathcal{V}_j|>1} \sum_{i \in \mathcal{V}_j} \exp(b_{n,i}) \Big[ \bar{E}(\boldsymbol{X}; W_{n,1i}, W_{n,2}) - \bar{E}(\boldsymbol{X}; W_{*,1j}, W_{*,2}) \Big] f_{G_n}(\boldsymbol{X}) \\
:= & B_{n,1}(\boldsymbol{X}) + B_{n,2}(\boldsymbol{X}).
\end{aligned}
$$

For Voronoi cells with exactly one element, we utilize the first-order Taylor expansion, while for those with more than one element, the second-order Taylor expansion is employed. This strategy leads to the following representations:

$$
B_{n,1}(\boldsymbol{X}) = \sum_{j:|\mathcal{V}_j|=1} \sum_{|\alpha|=1} \bar{M}_{n,j,\alpha_1,\alpha_2} \frac{\partial^{|\alpha_1|+|\alpha_2|}\bar{E}}{\partial W_1^{\alpha_1}\partial W_2^{\alpha_2}}(\boldsymbol{X}; W_{*,1j}, W_{*,2}) f_{G_n}(\boldsymbol{X}) + \bar{R}_{n,3}(\boldsymbol{X}),
$$

$$
B_{n,2}(\boldsymbol{X}) = \sum_{j:|\mathcal{V}_j|=1} \sum_{1 \le |\alpha| \le 2} \bar{M}_{n,j,\alpha_1,\alpha_2} \frac{\partial^{|\alpha_1|+|\alpha_2|}\bar{E}}{\partial W_1^{\alpha_1}\partial W_2^{\alpha_2}}(\boldsymbol{X}; W_{*,1j}, W_{*,2}) f_{G_n}(\boldsymbol{X}) + \bar{R}_{n,4}(\boldsymbol{X}).
$$

In these expressions, the functions $\bar{R}_{n,3}(\boldsymbol{X})$ and $\bar{R}_{n,4}(\boldsymbol{X})$ correspond to the Taylor remainders. Due to the uniform smoothness of the functions $\bar{E}$, we obtain $R_{n,3}(\boldsymbol{X})/\mathcal{D}_{1n} \to 0$ and $R_{n,4}(\boldsymbol{X})/\mathcal{D}_{1n} \to 0$

as $n$ approaches infinity. By computing the closed-form expressions for the partial derivatives of the function $\bar{E}$, both functions $B_{n,1}(\boldsymbol{X})$ and $B_{n,2}(\boldsymbol{X})$ can be rewritten as follows:

$$
B_{n,1}(\boldsymbol{X}) = \sum_{j:|\mathcal{V}_j|=1} \exp((W_{*,2}\sigma(W_{*,1j}\boldsymbol{X}))^\top \boldsymbol{X}) \big[\sigma^{(1)}(W_{*,1j}\boldsymbol{X})\boldsymbol{X}^\top L_{n,1,j}\boldsymbol{X} + \sigma(W_{*,1j}\boldsymbol{X})L_{n,2,j}^\top \boldsymbol{X}\big] f_{G_n}(\boldsymbol{X})
$$
$$
+ \bar{R}_{n,3}(\boldsymbol{X}),
$$
$$
B_{n,2}(\boldsymbol{X}) = \sum_{j:|\mathcal{V}_j|>1} \exp((W_{*,2}\sigma(W_{*,1j}\boldsymbol{X}))^\top \boldsymbol{X}) \big[\sigma^{(1)}(W_{*,1j}\boldsymbol{X})\boldsymbol{X}^\top L_{n,1,j}\boldsymbol{X} + \sigma(W_{*,1j}\boldsymbol{X})L_{n,2,j}^\top \boldsymbol{X}
$$
$$
+ \big([\sigma^{(1)}(W_{*,1j}\boldsymbol{X})]^2 + \sigma^{(2)}(W_{*,1j}\boldsymbol{X})\big)\boldsymbol{X}^\top L_{n,4,j}\boldsymbol{X}\boldsymbol{X}^\top W_{*,2} + \sigma^2(W_{*,1j}\boldsymbol{X})\boldsymbol{X}^\top L_{n,5,j}\boldsymbol{X}
$$
$$
+ \big(\sigma^{(1)}(W_{*,1j}\boldsymbol{X})\sigma(W_{*,1j}\boldsymbol{X})\boldsymbol{X}^\top W_{*,2} + \sigma^{(1)}(W_{*,1j}\boldsymbol{X})\big)\boldsymbol{X}^\top L_{n,6,j}\boldsymbol{X}\big] f_{G_n}(\boldsymbol{X}) + \bar{R}_{n,4}(\boldsymbol{X}),
$$

Combining these results, the function $Q_n(\boldsymbol{X})$ can be rewritten as follows:

$$
Q_n(\boldsymbol{X}) = \sum_{j:|\mathcal{V}_j|=1} \exp((W_{*,2}\sigma(W_{*,1j}\boldsymbol{X}))^\top \boldsymbol{X}) \big[\sigma^{(1)}(W_{*,1j}\boldsymbol{X})\sigma(W_{*,1j}\boldsymbol{X})\boldsymbol{X}^\top L_{n,1,j}\boldsymbol{X}W_{*,2}
$$
$$
+ \sigma^2(W_{*,1j}\boldsymbol{X})L_{n,2,j}^\top \boldsymbol{X}W_{*,2} + \sigma^{(1)}(W_{*,1j}\boldsymbol{X})L_{n,3,j}^\top \boldsymbol{X}W_{*,2} + \sigma(W_{*,1j}\boldsymbol{X})L_{n,2,j}^\top\big]
$$
$$
+ \sum_{j:|\mathcal{V}_j|>1} \exp((W_{*,2}\sigma(W_{*,1j}\boldsymbol{X}))^\top \boldsymbol{X}) \big[\sigma^{(1)}(W_{*,1j}\boldsymbol{X})\sigma(W_{*,1j}\boldsymbol{X})\boldsymbol{X}^\top L_{n,1,j}\boldsymbol{X}W_{*,2}
$$
$$
+ \sigma^2(W_{*,1j}\boldsymbol{X})L_{n,2,j}^\top \boldsymbol{X}W_{*,2} + \sigma^{(1)}(W_{*,1j}\boldsymbol{X})L_{n,3,j}^\top \boldsymbol{X}W_{*,2} + \sigma(W_{*,1j}\boldsymbol{X})L_{n,2,j}^\top
$$
$$
+ \big([\sigma^{(1)}(W_{*,1j}\boldsymbol{X})]^2\sigma(W_{*,1j}\boldsymbol{X}) + \sigma^{(2)}(W_{*,1j}\boldsymbol{X})\sigma(W_{*,1j}\boldsymbol{X})\big)\boldsymbol{X}^\top L_{n,4,j}\boldsymbol{X}\boldsymbol{X}^\top W_{*,2}W_{*,2}
$$
$$
+ \sigma^2(W_{*,1j}\boldsymbol{X})\sigma(W_{*,1j}\boldsymbol{X})\boldsymbol{X}^\top L_{n,5,j}\boldsymbol{X}W_{*,2} + \sigma^{(2)}(W_{*,1j}\boldsymbol{X})\boldsymbol{X}^\top L_{n,4,j}\boldsymbol{X}W_{*,2}
$$
$$
+ \big(\sigma^{(1)}(W_{*,1j}\boldsymbol{X})[\sigma(W_{*,1j}\boldsymbol{X})]^2\boldsymbol{X}^\top W_{*,2} + \sigma^{(1)}(W_{*,1j}\boldsymbol{X})\sigma(W_{*,1j}\boldsymbol{X})\big)\boldsymbol{X}^\top L_{n,6,j}\boldsymbol{X}W_{*,2}
$$
$$
+ \sigma^{(2)}(W_{*,1j}\boldsymbol{X})\boldsymbol{X}^\top L_{n,6,j} + [\sigma^{(1)}(W_{*,1j}\boldsymbol{X})]^2\boldsymbol{X}^\top L_{n,4,j}\boldsymbol{X}\boldsymbol{X}^\top W_{*,2}W_{*,2}
$$
$$
+ \sigma^{(1)}(W_{*,1j}\boldsymbol{X})\sigma(W_{*,1j}\boldsymbol{X})\boldsymbol{X}^\top W_{*,2}\boldsymbol{X}^\top L_{n,6,j} + \sigma^{(1)}(W_{*,1j}\boldsymbol{X})\sigma(W_{*,1j}\boldsymbol{X})\boldsymbol{X}^\top L_{n,6,j}\boldsymbol{X}W_{*,2}
$$
$$
+ [\sigma(W_{*,1j}\boldsymbol{X})]^2\boldsymbol{X}^\top L_{n,5,j}\big]
$$

$$
- \sum_{j:|\mathcal{V}_j|=1} \exp((W_{*,2}\sigma(W_{*,1j}\boldsymbol{X}))^\top \boldsymbol{X}) \big[\sigma^{(1)}(W_{*,1j}\boldsymbol{X})\boldsymbol{X}^\top L_{n,1,j}\boldsymbol{X} + \sigma(W_{*,1j}\boldsymbol{X})L_{n,2,j}^\top \boldsymbol{X}\big] f_{G_n}(\boldsymbol{X})
$$
$$
- \sum_{j:|\mathcal{V}_j|>1} \exp((W_{*,2}\sigma(W_{*,1j}\boldsymbol{X}))^\top \boldsymbol{X}) \big[\sigma^{(1)}(W_{*,1j}\boldsymbol{X})\boldsymbol{X}^\top L_{n,1,j}\boldsymbol{X} + \sigma(W_{*,1j}\boldsymbol{X})L_{n,2,j}^\top \boldsymbol{X}
$$
$$
+ \big([\sigma^{(1)}(W_{*,1j}\boldsymbol{X})]^2 + \sigma^{(2)}(W_{*,1j}\boldsymbol{X})\big)\boldsymbol{X}^\top L_{n,4,j}\boldsymbol{X}\boldsymbol{X}^\top W_{*,2} + \sigma^2(W_{*,1j}\boldsymbol{X})\boldsymbol{X}^\top L_{n,5,j}\boldsymbol{X}
$$
$$
+ \big(\sigma^{(1)}(W_{*,1j}\boldsymbol{X})\sigma(W_{*,1j}\boldsymbol{X})\boldsymbol{X}^\top W_{*,2} + \sigma^{(1)}(W_{*,1j}\boldsymbol{X})\big)\boldsymbol{X}^\top L_{n,6,j}\boldsymbol{X}\big] f_{G_n}(\boldsymbol{X})
$$
$$
- \sum_{j=1}^{L} \bar{M}_{n,j,\boldsymbol{0}_d,\boldsymbol{0}_d} \exp((W_{*,2}\sigma(W_{*,1j}\boldsymbol{X}))^\top \boldsymbol{X}) f_{G_n}(\boldsymbol{X})
$$
$$
+ \sum_{j=1}^{L} \bar{M}_{n,j,\boldsymbol{0}_d,\boldsymbol{0}_d} \exp((W_{*,2}\sigma(W_{*,1j}\boldsymbol{X}))^\top \boldsymbol{X})\sigma(W_{*,1j}\boldsymbol{X})W_{*,2}
$$
$$
+ \tilde{R}_{n,1}(\boldsymbol{X}) + \tilde{R}_{n,2}(\boldsymbol{X}) - \bar{R}_{n,3}(\boldsymbol{X}) - \bar{R}_{n,4}(\boldsymbol{X}) \tag{22}
$$

where the coefficient $\bar{M}_{n,j,\boldsymbol{0}_d,\boldsymbol{0}_d} := \sum_{i\in\mathcal{V}_j} \exp(b_{n,i}) - \exp(b_{*,j})$ for any $j \in [L]$.

**Step 2 - Non-vanishing coefficients.** An important insight from Equation (22) is that the ratio $Q_n(\boldsymbol{X})/\mathcal{D}_{1n}$ can be expressed as a linear combination of the following independent functions:

$$E(\boldsymbol{X}; W_{*,1j}, W_{*,2})\sigma(W_{*,1j}\boldsymbol{X})W_{*,2}, \ E(\boldsymbol{X}; W_{*,1j}, W_{*,2})\sigma^{(1)}(W_{*,1j}\boldsymbol{X})\sigma(W_{*,1j}\boldsymbol{X})\boldsymbol{X}^{(u)}\boldsymbol{X}^{(v)}W_{*,2},$$

$$E(\boldsymbol{X}; W_{*,1j}, W_{*,2})\sigma^2(W_{*,1j}\boldsymbol{X})\boldsymbol{X}^{(u)}W_{*,2}, \ E(\boldsymbol{X}; W_{*,1j}, W_{*,2})\sigma^{(1)}(W_{*,1j}\boldsymbol{X})\boldsymbol{X}^{(u)}W_{*,2},$$

$$E(\boldsymbol{X}; W_{*,1j}, W_{*,2})\sigma(W_{*,1j}\boldsymbol{X})e_u, \ E(\boldsymbol{X}; W_{*,1j}, W_{*,2})[\sigma^{(1)}(W_{*,1j}\boldsymbol{X})]^2\sigma(W_{*,1j}\boldsymbol{X})\boldsymbol{X}^{(u)}\boldsymbol{X}^{(v)}\boldsymbol{X}^{(w)}W_{*,2},$$

$$E(\boldsymbol{X}; W_{*,1j}, W_{*,2})\sigma^{(2)}(W_{*,1j}\boldsymbol{X})\sigma(W_{*,1j}\boldsymbol{X})\boldsymbol{X}^{(u)}\boldsymbol{X}^{(v)}\boldsymbol{X}^{(w)}W_{*,2}, \ E(\boldsymbol{X}; W_{*,1j}, W_{*,2})\sigma^{(2)}(W_{*,1j}\boldsymbol{X})\boldsymbol{X}^{(u)}\boldsymbol{X}^{(v)}W_{*,2},$$

$$E(\boldsymbol{X}; W_{*,1j}, W_{*,2})\sigma^2(W_{*,1j}\boldsymbol{X})\sigma(W_{*,1j}\boldsymbol{X})\boldsymbol{X}^{(u)}\boldsymbol{X}^{(v)}W_{*,2},$$

$$E(\boldsymbol{X}; W_{*,1j}, W_{*,2})\sigma^{(1)}(W_{*,1j}\boldsymbol{X})[\sigma(W_{*,1j}\boldsymbol{X})]^2\boldsymbol{X}^{(u)}\boldsymbol{X}^{(v)}\boldsymbol{X}^{(w)}W_{*,2}, \ E(\boldsymbol{X}; W_{*,1j}, W_{*,2})\sigma^{(2)}(W_{*,1j}\boldsymbol{X})\boldsymbol{X}^{(u)}e_u,$$

$$E(\boldsymbol{X}; W_{*,1j}, W_{*,2})[\sigma^{(1)}(W_{*,1j}\boldsymbol{X})]^2\boldsymbol{X}^{(u)}\boldsymbol{X}^{(v)}\boldsymbol{X}^{(w)}W_{*,2}, \ E(\boldsymbol{X}; W_{*,1j}, W_{*,2})\sigma^{(1)}(W_{*,1j}\boldsymbol{X})\sigma(W_{*,1j}\boldsymbol{X})\boldsymbol{X}^{(u)}\boldsymbol{X}^{(v)}e_v,$$

$$E(\boldsymbol{X}; W_{*,1j}, W_{*,2})[\sigma(W_{*,1j}\boldsymbol{X})]^2\boldsymbol{X}^{(u)}e_u,$$

$$E(\boldsymbol{X}; W_{*,1j}, W_{*,2})\sigma^{(1)}(W_{*,1j}\boldsymbol{X})\boldsymbol{X}^{(u)}\boldsymbol{X}^{(v)}f_{G_n}(\boldsymbol{X}), \ E(\boldsymbol{X}; W_{*,1j}, W_{*,2})\sigma(W_{*,1j}\boldsymbol{X})\boldsymbol{X}^{(u)}f_{G_n}(\boldsymbol{X}),$$

$$E(\boldsymbol{X}; W_{*,1j}, W_{*,2})[\sigma^{(1)}(W_{*,1j}\boldsymbol{X})]^2\boldsymbol{X}^{(u)}\boldsymbol{X}^{(v)}\boldsymbol{X}^{(w)}f_{G_n}(\boldsymbol{X}), \ E(\boldsymbol{X}; W_{*,1j}, W_{*,2})\sigma^{(2)}(W_{*,1j}\boldsymbol{X})\boldsymbol{X}^{(u)}\boldsymbol{X}^{(v)}\boldsymbol{X}^{(w)}f_{G_n}(\boldsymbol{X}),$$

$$E(\boldsymbol{X}; W_{*,1j}, W_{*,2})\sigma^2(W_{*,1j}\boldsymbol{X})\boldsymbol{X}^{(u)}\boldsymbol{X}^{(v)}f_{G_n}(\boldsymbol{X}), \ E(\boldsymbol{X}; W_{*,1j}, W_{*,2})\sigma^{(1)}(W_{*,1j}\boldsymbol{X})\boldsymbol{X}^{(u)}\boldsymbol{X}^{(v)}f_{G_n}(\boldsymbol{X}),$$

$$E(\boldsymbol{X}; W_{*,1j}, W_{*,2})\sigma^{(1)}(W_{*,1j}\boldsymbol{X})\sigma(W_{*,1j}\boldsymbol{X})\boldsymbol{X}^{(u)}\boldsymbol{X}^{(v)}\boldsymbol{X}^{(w)}f_{G_n}(\boldsymbol{X}), \ E(\boldsymbol{X}; W_{*,1j}, W_{*,2})f_{G_n}(\boldsymbol{X}),$$

for any $1 \leq j \leq L$ and $1 \leq u, v, w \leq d$.

We now demonstrate that not all of the coefficients of these linearly independent functions converge to 0 as $n$ goes to infinity. To prove this by contradiction, assume that all these coefficients do converge to 0 as $n$ goes to $\infty$. From the representation of the ratio $Q_n(\boldsymbol{X})/\mathcal{D}_{1n}$ in terms of these linearly independent functions, it follows that these ratios $L_{n,1,j}^{(u)}/\mathcal{D}_{1n}$, $L_{n,2,j}^{(u)}/\mathcal{D}_{1n}$, $L_{n,3,j}^{(u)}/\mathcal{D}_{1n}$, $L_{n,4,j}^{(uv)}/\mathcal{D}_{1n}$, $L_{n,5,j}^{(uv)}/\mathcal{D}_{1n}$, $L_{n,6,j}^{(uv)}/\mathcal{D}_{1n}$, and $M_{n,j,\mathbf{0}_d,\mathbf{0}_d}/\mathcal{D}_{1n}$ all converge to 0 as $n$ approaches infinity for all indices $u, v \in [d]$ and $j \in [L]$.

By first considering the vanishing of the ratio $\bar{M}_{n,j,\mathbf{0}_d,\mathbf{0}_d}/\mathcal{D}_{1n}$ to 0 for any $j \in [L]$ and then taking the absolute value of this ratio, we find that

$$\frac{|M_{n,j,\mathbf{0}_d,\mathbf{0}_d}|}{\mathcal{D}_{1n}} = \frac{|\sum_{i \in \mathcal{V}_j} \exp(b_{n,i}) - \exp(b_{*,j})|}{\mathcal{D}_{1n}} \to 0,$$

By varying the index $j$ from 1 to $L$ and summing the corresponding limits, we find that

$$\frac{\sum_{j=1}^{L} |\sum_{i \in \mathcal{V}_j} \exp(b_{n,i}) - \exp(b_{*,j})|}{\mathcal{D}_{1n}} \to 0. \tag{23}$$

Our strategy is now to consider separately the terms corresponding to Voronoi cells with exactly one element and those with more than one element. In particular, for Voronoi cells with exactly one element, *i.e.,* for indices $j \in [L]$ such that $|\mathcal{V}_j| = 1$, as the ratio $L_{n,3,j}^{(u)}/\mathcal{D}_{1n}$ approaches 0, we have

$$\frac{\sum_{i \in \mathcal{V}_j} \exp(b_{n,i})\|\Delta W_{n,1ij}\|_1}{\mathcal{D}_{1n}} = \frac{\sum_{u=1}^{d} |L_{n,3,j}^{(u)}|}{\mathcal{D}_{1n}} \to 0.$$

Similarly, the condition $L_{n,2,j}/\mathcal{D}_{1n} \to 0$ implies that $\dfrac{\sum_{i \in \mathcal{V}_j} \exp(b_{n,i})\|\Delta W_{n,2ij}\|_1}{\mathcal{D}_{1n}} \to 0$. Combining these results, we obtain

$$\frac{\sum_{j:|\mathcal{V}_j|=1} \sum_{i \in \mathcal{V}_j} \exp(b_{n,i})(\|\Delta W_{n,1ij}\|_1 + \|\Delta W_{n,2ij}\|_1)}{\mathcal{D}_{1n}} \to 0.$$

Due to the equivalence of the $\ell_1$ and $\ell_2$ norms, we deduce that

$$\frac{\sum_{j:|\mathcal{V}_j|=1} \sum_{i \in \mathcal{V}_j} \exp(b_{n,i})(\|\Delta W_{n,1ij}\| + \|\Delta W_{n,2ij}\|)}{\mathcal{D}_{1n}} \to 0. \tag{24}$$

We now turn to Voronoi cells with more than one element, namely, indices $j \in [L]$ such that $|\mathcal{V}_j| > 1$. As the ratios $L_{n,4,j}^{(uu)}/\mathcal{D}_{1n} \to 0$, we find that

$$\frac{\sum_{u=1}^d L_{n,4,j}^{(uu)}}{\mathcal{D}_{1n}} = \frac{\sum_{i \in \mathcal{V}_j} \exp(b_{n,i}) \|\Delta W_{n,1ij}\|^2}{\mathcal{D}_{1n}} \to 0.$$

Likewise, as $L_{n,5,j}^{(uu)}/\mathcal{D}_{1n} \to 0$, we arrive at $\dfrac{\sum_{i \in \mathcal{V}_j} \exp(b_{n,i}) \|\Delta W_{n,2ij}\|^2}{\mathcal{D}_{1n}} \to 0$. These results together imply that

$$\frac{\sum_{j:|\mathcal{V}_j|>1} \sum_{i \in \mathcal{V}_j} \exp(b_{n,i})(\|\Delta W_{n,1ij}\|^2 + \|\Delta W_{n,2ij}\|^2)}{\mathcal{D}_{1n}} \to 0. \tag{25}$$

Combining the results of Equations (23), (24), and (25), we achieve that

$$\frac{\mathcal{D}_{1n}}{\mathcal{D}_{1n}} = 1 \to 0, \text{ as } n \to \infty,$$

which is a contradiction. Consequently, at least one of the coefficients of the linearly independent functions in the expression for the ratio $Q_n(\boldsymbol{X})/\mathcal{D}_{1n}$ does not converge to 0 as $n$ approaches infinity.

**Step 3 - Application of Fatou's lemma.** In this step, we divide all coefficients of the linearly independent terms in the expression of the ratio $Q_n(\boldsymbol{X})/\mathcal{D}_{1n}$, namely, $L_{n,1,j}^{(u)}/\mathcal{D}_{1n}$, $L_{n,2,j}^{(u)}/\mathcal{D}_{1n}$, $L_{n,3,j}^{(u)}/\mathcal{D}_{1n}$, $L_{n,4,j}^{(uv)}/\mathcal{D}_{1n}$, $L_{n,5,j}^{(uv)}/\mathcal{D}_{1n}$, $L_{n,6,j}^{(uv)}/\mathcal{D}_{1n}$, and $M_{n,j,\mathbf{0}_d,\mathbf{0}_d}/\mathcal{D}_{1n}$ for all $u, v \in [d]$, by the maximum of their absolute values. Specifically, we denote by $m_n$ the maximum of the absolute values of these coefficients. Since not all of these coefficients approach 0, it follows that $1/m_n$ does not approach infinity as $n \to \infty$.

From the hypothesis, we have $\|f_{G_n} - f_{G_*}\|_{L_2(\mu)}/\mathcal{D}_{1n} \to 0$ as $n \to \infty$. As $1/m_n \not\to \infty$, it follows that $\|f_{G_n} - f_{G_*}\|_{L_2(\mu)}/(m_n \mathcal{D}_{1n}) \to 0$. An application of Fatou's lemma yields

$$\lim_{n\to\infty} \frac{\|f_{G_n} - f_{G_*}\|_{L_2(\mu)}}{m_n \mathcal{D}_{1n}} \geq \int \liminf_{n\to\infty} \frac{|f_{G_n}(\boldsymbol{X}) - f_{G_*}(\boldsymbol{X})|}{m_n \mathcal{D}_{1n}} d\mu(\boldsymbol{X}).$$

Combining these results, we obtain $\liminf_{n\to\infty} \dfrac{|f_{G_n}(\boldsymbol{X}) - f_{G_*}(\boldsymbol{X})|}{m_n \mathcal{D}_{2n}} = 0$ for almost surely $\boldsymbol{X}$. To simplify the presentation, we define the following limits:

$$\frac{L_{n,\tau,j}}{m_n \mathcal{D}_{1n}} \to \lambda_{\tau,j}, \qquad \frac{M_{n,j,\mathbf{0}_d,\mathbf{0}_d}}{\mathcal{D}_{1n}} \to \lambda_{0,j},$$

for any $1 \leq \tau \leq 6$ and $1 \leq j \leq L$. By the definition of $m_n$, at least one coefficient in the sets $\{\lambda_{0,j}, \lambda_{1,j}, \lambda_{2,j}, \lambda_{3,j}\}_{j:|\mathcal{V}_j|=1}$, $\{\lambda_{0,j}, \lambda_{1,j}, \lambda_{2,j}, \lambda_{3,j}, \lambda_{4,j}, \lambda_{5,j}, \lambda_{6,j}\}_{j:|\mathcal{V}_j|>1}$ must be non-zero. The condition $\liminf_{n\to\infty} \dfrac{|f_{G_n}(\boldsymbol{X}) - f_{G_*}(\boldsymbol{X})|}{m_n \mathcal{D}_{1n}} = 0$, or equivalently, $\liminf_{n\to\infty} \dfrac{|Q_n(\boldsymbol{X})|}{m_n \mathcal{D}_{1n}} = 0$ implies that

$$\sum_{j:|\mathcal{V}_j|=1} \exp((W_{*,2}\sigma(W_{*,1j}\boldsymbol{X}))^\top \boldsymbol{X}) \big[ \sigma^{(1)}(W_{*,1j}\boldsymbol{X})\sigma(W_{*,1j}\boldsymbol{X})\boldsymbol{X}^\top \lambda_{1,j} \boldsymbol{X} W_{*,2}$$

$$+ \sigma^2(W_{*,1j}\boldsymbol{X})\lambda_{2,j}^\top \boldsymbol{X} W_{*,2} + \sigma^{(1)}(W_{*,1j}\boldsymbol{X})\lambda_{3,j}^\top \boldsymbol{X} W_{*,2} + \sigma(W_{*,1j}\boldsymbol{X})\lambda_{2,j}^\top \big]$$

$$+ \sum_{j:|\mathcal{V}_j|>1} \exp((W_{*,2}\sigma(W_{*,1j}\boldsymbol{X}))^\top \boldsymbol{X}) \big[ \sigma^{(1)}(W_{*,1j}\boldsymbol{X})\sigma(W_{*,1j}\boldsymbol{X})\boldsymbol{X}^\top \lambda_{1,j} \boldsymbol{X} W_{*,2}$$

$$+ \sigma^2(W_{*,1j}\boldsymbol{X})\lambda_{2,j}^\top \boldsymbol{X} W_{*,2} + \sigma^{(1)}(W_{*,1j}\boldsymbol{X})\lambda_{3,j}^\top \boldsymbol{X} W_{*,2} + \sigma(W_{*,1j}\boldsymbol{X})\lambda_{2,j}^\top$$

$$+ \big([\sigma^{(1)}(W_{*,1j}\boldsymbol{X})]^2 \sigma(W_{*,1j}\boldsymbol{X}) + \sigma^{(2)}(W_{*,1j}\boldsymbol{X})\sigma(W_{*,1j}\boldsymbol{X})\big)\boldsymbol{X}^\top \lambda_{4,j} \boldsymbol{X}\boldsymbol{X}^\top W_{*,2}W_{*,2}$$

$$+ \sigma^2(W_{*,1j}\boldsymbol{X})\sigma(W_{*,1j}\boldsymbol{X})\boldsymbol{X}^\top \lambda_{5,j} \boldsymbol{X} W_{*,2} + \sigma^{(2)}(W_{*,1j}\boldsymbol{X})\boldsymbol{X}^\top \lambda_{4,j} \boldsymbol{X} W_{*,2}$$

$$+ \big(\sigma^{(1)}(W_{*,1j}\boldsymbol{X})[\sigma(W_{*,1j}\boldsymbol{X})]^2 \boldsymbol{X}^\top W_{*,2} + \sigma^{(1)}(W_{*,1j}\boldsymbol{X})\sigma(W_{*,1j}\boldsymbol{X})\big)\boldsymbol{X}^\top \lambda_{6,j} \boldsymbol{X} W_{*,2}$$

$$+ \sigma^{(2)}(W_{*,1j}\boldsymbol{X})\boldsymbol{X}^\top \lambda_{6,j} + [\sigma^{(1)}(W_{*,1j}\boldsymbol{X})]^2 \boldsymbol{X}^\top \lambda_{4,j} \boldsymbol{X}\boldsymbol{X}^\top W_{*,2}W_{*,2}$$

$$+ \sigma^{(1)}(W_{*,1j}\boldsymbol{X})\sigma(W_{*,1j}\boldsymbol{X})\boldsymbol{X}^\top W_{*,2}\boldsymbol{X}^\top \lambda_{6,j} + \sigma^{(1)}(W_{*,1j}\boldsymbol{X})\sigma(W_{*,1j}\boldsymbol{X})\boldsymbol{X}^\top \lambda_{6,j} \boldsymbol{X} W_{*,2}$$

$$+ [\sigma(W_{*,1j}\boldsymbol{X})]^2 \boldsymbol{X}^\top \lambda_{5,j} \big]$$

$$- \sum_{j:|\mathcal{V}_j|=1} \exp((W_{*,2}\sigma(W_{*,1j}\boldsymbol{X}))^\top \boldsymbol{X})\big[\sigma^{(1)}(W_{*,1j}\boldsymbol{X})\boldsymbol{X}^\top \lambda_{1,j}\boldsymbol{X} + \sigma(W_{*,1j}\boldsymbol{X})\lambda_{2,j}^\top \boldsymbol{X}\big]f_{G_n}(\boldsymbol{X})$$

$$- \sum_{j:|\mathcal{V}_j|>1} \exp((W_{*,2}\sigma(W_{*,1j}\boldsymbol{X}))^\top \boldsymbol{X})\big[\sigma^{(1)}(W_{*,1j}\boldsymbol{X})\boldsymbol{X}^\top \lambda_{1,j}\boldsymbol{X} + \sigma(W_{*,1j}\boldsymbol{X})\lambda_{2,j}^\top \boldsymbol{X}$$

$$+ \big([\sigma^{(1)}(W_{*,1j}\boldsymbol{X})]^2 + \sigma^{(2)}(W_{*,1j}\boldsymbol{X}))\boldsymbol{X}^\top \lambda_{4,j}\boldsymbol{X}\boldsymbol{X}^\top W_{*,2} + \sigma^2(W_{*,1j}\boldsymbol{X})\boldsymbol{X}^\top \lambda_{5,j}\boldsymbol{X}$$

$$+ \big(\sigma^{(1)}(W_{*,1j}\boldsymbol{X})\sigma(W_{*,1j}\boldsymbol{X})\boldsymbol{X}^\top W_{*,2} + \sigma^{(1)}(W_{*,1j}\boldsymbol{X}))\boldsymbol{X}^\top \lambda_{6,j}\boldsymbol{X}\big]f_{G_n}(\boldsymbol{X})$$

$$- \sum_{j=1}^{L} \lambda_{0,j} \exp((W_{*,2}\sigma(W_{*,1j}\boldsymbol{X}))^\top \boldsymbol{X})f_{G_n}(\boldsymbol{X})$$

$$+ \sum_{j=1}^{L} \lambda_{0,j} \exp((W_{*,2}\sigma(W_{*,1j}\boldsymbol{X}))^\top \boldsymbol{X})\sigma(W_{*,1j}\boldsymbol{X})W_{*,2} = 0, \tag{26}$$

for almost surely $\boldsymbol{X}$. However, that equation implies that all the coefficients $\{\lambda_{0,j}, \lambda_{1,j}, \lambda_{2,j}, \lambda_{3,j}\}_{j:|\mathcal{V}_j|=1}$, $\{\lambda_{0,j}, \lambda_{1,j}, \lambda_{2,j}, \lambda_{3,j}, \lambda_{4,j}, \lambda_{5,j}, \lambda_{6,j}\}_{j:|\mathcal{V}_j|>1}$ are 0. It is a contradiction to the hypothesis that at least one coefficient among these coefficients is different from 0.

Consequently, we deduce that

$$\lim_{\varepsilon \to 0} \inf_{G \in \mathcal{G}_{L'}(\Theta): \mathcal{D}_1(G,G_*) \leq \varepsilon} \|f_G - f_{G_*}\|_{L_2(\mu)}/\mathcal{D}_1(G, G_*) > 0,$$

which proves the local part of inequality (20).

### A.1.2 GLOBAL PART

From the local part of inequality (20), we can find a positive constant $\varepsilon'$ such that the following inequality holds:

$$\inf_{G \in \mathcal{G}_{L'}(\Theta): \mathcal{D}_1(G,G_*) \leq \varepsilon'} \|f_G - f_{G_*}\|_{L_2(\mu)}/\mathcal{D}_1(G, G_*) > 0.$$

To obtain the conclusion of the theorem, we only need to demonstrate that

$$\inf_{G \in \mathcal{G}_{L'}(\Theta): \mathcal{D}_1(G,G_*) > \varepsilon'} \|f_G - f_{G_*}\|_{L_2(\mu)}/\mathcal{D}_1(G, G_*) > 0.$$

We prove the claim by contradiction. Assuming the claim does not hold implies that there exists a sequence of $G'_n := \sum_{j'=1}^{L'} \exp(b_{n,j'})\delta_{(W_{n,1j'}, W_{n,2})}$ in the set $\mathcal{G}_{L'}(\Theta)$ such that

$$\begin{cases} \mathcal{D}_1(G'_n, G_*) > \varepsilon' \\ \|f_{G'_n} - f_{G_*}\|_{L_2(\mu)}/\mathcal{D}_1(G'_n, G_*) \to 0, \end{cases}$$

as $n$ approaches the infinity. This implies that $\|f_{G'_n} - f_{G_*}\|_{L_2(\mu)} \to 0$ as $n$ approaches infinity.

By hypothesis, the parameter space $\Theta$ is compact. Therefore, there exists a subsequence of $G'_n$'s, that converges to a mixing measure $G'$ where $G'$ lies in the space $\mathcal{G}_{L'}(\Theta)$. From the hypothesis, we have $\mathcal{D}_1(G'_n, G_*) > \varepsilon'$. By taking the limit of both sides as $n \to \infty$, we obtain $\mathcal{D}_1(G', G_*) \geq \varepsilon'$.

An application of Fatou's lemma leads to the following result:

$$0 = \lim_{n \to \infty} \|f_{G'_n} - f_{G_*}\|_{L_2(\mu)} \geq \int \liminf_{n \to \infty} \big\|f_{G'_n}(\boldsymbol{X}) - f_{G_*}(\boldsymbol{X})\big\|^2 d\mu(\boldsymbol{X}).$$

This inequality is only possible if $f_{G'} = f_{G_*}$ for almost surely $\boldsymbol{X}$.

By the identifiability of the function $f_G(\boldsymbol{X})$, this equation only holds when $G' \equiv G_*$. Consequently, $\mathcal{D}_1(G', G_*) = 0$. This contradicts the assumption that $\mathcal{D}_1(G', G_*) \geq \varepsilon' > 0$. Hence, the proof of the global part is complete, thereby establishing the conclusion of the theorem.

**Proof for the Identifiability Property.** The key claim we aim to show is that if the equation $f_G(\boldsymbol{X}) = f_{G_*}(\boldsymbol{X})$ holds for almost surely $\boldsymbol{X}$, then $G \equiv G_*$, namely, that the two mixing measures are identical.

From the hypothesis that $f_G(\boldsymbol{X}) = f_{G_*}(\boldsymbol{X})$ for almost all $\boldsymbol{X}$, it follows that

$$\sum_{j=1}^{N} \frac{\exp(\boldsymbol{X}^\top A_j^0 \boldsymbol{X} + a_j^0))}{D_{f,G}(\boldsymbol{X})} h(\boldsymbol{X}, \eta_j^0) + \sum_{j'=1}^{\tilde{L}} \frac{\exp((BW_2\sigma(W_{1j'}\boldsymbol{X}))^\top \boldsymbol{X} + b_{j'})}{D_{f,G}(\boldsymbol{X})} CW_2\sigma(W_{1j'}\boldsymbol{X})$$

$$= \sum_{j=1}^{N} \frac{\exp(\boldsymbol{X}^\top A_j^0 \boldsymbol{X} + a_j^0))}{D_{f,G_*}(\boldsymbol{X})} h(\boldsymbol{X}, \eta_j^0) + \sum_{j'=1}^{L} \frac{\exp((BW_{*,2}\sigma(W_{*,1j'}\boldsymbol{X}))^\top \boldsymbol{X} + b_{*,j'})}{D_{f,G_*}(\boldsymbol{X})} CW_{*,2}\sigma(W_{*,1j'}\boldsymbol{X}),$$
(27)

where $G = \sum_{j=1}^{\tilde{L}} \exp(b_j)\delta_{(W_{1j}, W_2)}$. Furthermore, we define

$$D_{f,G_*}(\boldsymbol{X}) = \sum_{k=1}^{N} \exp(\boldsymbol{X}^\top A_k^0 \boldsymbol{X} + a_k^0) + \sum_{j'=1}^{L} \exp((BW_{*,2}\sigma(W_{*,1j'}\boldsymbol{X}))^\top \boldsymbol{X} + b_{*,j'}),$$

$$D_{f,G}(\boldsymbol{X}) = \sum_{k=1}^{N} \exp(\boldsymbol{X}^\top A_k^0 \boldsymbol{X} + a_k^0) + \sum_{j'=1}^{\tilde{L}} \exp((BW_2\sigma(W_{1j'}\boldsymbol{X}))^\top \boldsymbol{X} + b_{j'}).$$

This equation implies that the number of atoms of $G$ and $G_*$ should be identical, namely, $L = \tilde{L}$. Therefore, the following equality holds

$$\left\{ \frac{\exp((BW_2\sigma(W_{1j'}\boldsymbol{X}))^\top \boldsymbol{X} + b_{j'})}{D_{f,G}(\boldsymbol{X})} : j' \in [L] \right\} = \left\{ \frac{\exp((BW_{*,2}\sigma(W_{*,1j'}\boldsymbol{X}))^\top \boldsymbol{X} + b_{*,j'})}{D_{f,G_*}(\boldsymbol{X})} : j' \in [L] \right\},$$

for almost surely $\boldsymbol{X}$. By relabelling the indices of these two sets, we can assume without loss of generality that

$$\frac{\exp((BW_2\sigma(W_{1j'}\boldsymbol{X}))^\top \boldsymbol{X} + b_{j'})}{D_{f,G}(\boldsymbol{X})} = \frac{\exp((BW_{*,2}\sigma(W_{*,1j'}\boldsymbol{X}))^\top \boldsymbol{X} + b_{*,j'})}{D_{f,G_*}(\boldsymbol{X})},$$

for any index $j' \in [L]$ and for almost surely $\boldsymbol{X}$. From the translation invariance property of the softmax function, the Equation (27) becomes

$$\sum_{j=1}^{L} \exp(b_j) \exp((BW_2\sigma(W_{1j}\boldsymbol{X}))^\top \boldsymbol{X}) CW_2\sigma(W_{1j}\boldsymbol{X})$$

$$= \sum_{j'=1}^{L} \exp(b_{*,j}) \exp((BW_{*,2}\sigma(W_{*,1j'}\boldsymbol{X}))^\top \boldsymbol{X}) CW_{*,2}\sigma(W_{*,1j'}\boldsymbol{X}), \quad (28)$$

for almost surely $\boldsymbol{X}$. This equation suggests that there exists a partition $K_1, K_2, \ldots, K_m$ of the set $[L]$ for some $m$ such that we have $\exp(b_{j_1}) = \exp(b_{*,j_2})$ for any $j_1, j_2 \in K_i$ and for any $i \in [m]$. Based on this result, the Equation (27) can be rewritten as follows:

$$\sum_{i=1}^{m} \sum_{j_1 \in K_i} \exp(b_{j_1}) \exp\left((BW_2\sigma(W_{1j_1}\boldsymbol{X}))^\top \boldsymbol{X}\right) CW_2\sigma(W_{1j_1}\boldsymbol{X})$$

$$= \sum_{i=1}^{m} \sum_{j_2 \in K_i} \exp(b_{*,j_2}) \exp\left((BW_{*,2}\sigma(W_{*,1j_2}\boldsymbol{X}))^\top \boldsymbol{X}\right) CW_{*,2}\sigma(W_{*,1j_2}\boldsymbol{X}),$$

for almost surely $\boldsymbol{X}$. This equation proves that

$$\{W_2\sigma(W_{1j_1}\boldsymbol{X}) : j_1 \in K_i\} = \{W_{*,2}\sigma(W_{*,1j_2}\boldsymbol{X}) : j_2 \in K_i\},$$

for any $i \in [m]$. Since the activation function $\sigma$ is identifiable, this result indicates that

$$\{(W_{1j_1}, W_2) : j_1 \in K_i\} = \{(W_{*,1j_2}, W_2) : j_2 \in K_i\}.$$

Consequently, we arrive at the following equality:

$$\sum_{i=1}^{m} \sum_{j_1 \in K_i} \exp(b_{j_1})\delta_{(W_{1j_1}, W_2)} = \sum_{i=1}^{m} \sum_{j_2 \in K_i} \exp(b_{*,j_2})\delta_{(W_{*,1j_2}, W_{*,2})}.$$

This is equivalent to $G \equiv G_*$. Thus, we obtain the conclusion of the identifiability property of the function $f_G$.

## A.2 PROOF OF PROPOSITION 1

Recall that $(\boldsymbol{X}_1, Y_1), (\boldsymbol{X}_2, Y_2), \ldots, (\boldsymbol{X}_n, Y_n) \in \mathbb{R}^d \times \mathbb{R}^{d'}$ follow the regression model:

$$Y_i = f_{G_*}(\boldsymbol{X}_i) + \varepsilon_i, \quad i = 1, 2, \ldots, n,$$

where the independent Gaussian noises $\varepsilon_1, \ldots, \varepsilon_n$ satisfy that $\mathbb{E}[\varepsilon_i|\boldsymbol{X}_i] = 0$ and $\mathrm{Var}(\varepsilon_i|\boldsymbol{X}_i) = \sigma^2 I_{d'}$ for all $i \in [n]$. The true regression function takes the following form:

$$f_{G_*}(\boldsymbol{X}) := \sum_{j=1}^{N} \frac{\exp(\boldsymbol{X}^\top A_j^0 \boldsymbol{X} + a_j^0)}{D_f(\boldsymbol{X})} \cdot h(\boldsymbol{X}, \eta_j^0)$$

$$+ \sum_{j'=1}^{L} \frac{\exp((BW_*^{(2)}\sigma(W_{*,j'}^{(1)}\boldsymbol{X}))^\top \boldsymbol{X} + b_{*,j'})}{D_f(\boldsymbol{X})} \cdot CW_*^{(2)}\sigma(W_{*,j'}^{(1)}\boldsymbol{X})$$

where $D_f(\boldsymbol{X}) := \sum_{k=1}^{N} \exp(\boldsymbol{X}^\top A_k^0 \boldsymbol{X} + a_k^0) + \sum_{j'=1}^{L} \exp((BW_*^{(2)}\sigma(W_{*,j'}^{(1)}\boldsymbol{X}))^\top \boldsymbol{X} + b_{*,j'})$. The least-squares estimator $\widehat{G}_n$ is defined as

$$\widehat{G}_n := \arg\min_{G \in \mathcal{G}_{L'}(\Theta)} \sum_{i=1}^{n} \|Y_i - f_G(\boldsymbol{X}_i)\|^2,$$

The assumption that $\varepsilon_i|\boldsymbol{X}_i \sim \mathcal{N}(\boldsymbol{0}_d, \sigma^2 I_{d'})$ indicates that the least-squares estimator $\widehat{G}_n$ is indeed a maximum likelihood estimator, given by:

$$\widehat{G}_n \in \arg\max_{G \in \mathcal{G}_{L'}(\Theta)} \frac{1}{n} \sum_{i=1}^{n} \log(p(Y_i|f_G(\boldsymbol{X}_i), \sigma^2 I_{d'})),$$

where $p(Y_i|f_G(\boldsymbol{X}_i), \sigma^2 I_{d'})$ is the multivariate Gaussian distribution with mean $f_G(\boldsymbol{X})$ and covariance matrix $\sigma^2 I_{d'}$. From the result of Theorem 7.4 in van de Geer (2000), it follows that

$$h(p(Y|f_{\widehat{G}_n}(\boldsymbol{X}), \sigma^2 I_{d'}), p(Y|f_{G_*}(\boldsymbol{X}), \sigma^2 I_{d'})) = \mathcal{O}_P(\sqrt{\log(n)/n}).$$

Given the closed-form expression for the Hellinger distance between two multivariate Gaussian distributions, we have

$$h^2(p(Y|f_{\widehat{G}_n}(\boldsymbol{X}), \sigma^2 I_{d'}), p(Y|f_{G_*}(\boldsymbol{X}), \sigma^2 I_{d'})) = 1 - \exp\left\{ -\frac{1}{8\sigma^2}\|f_{\widehat{G}_n}(\boldsymbol{X}) - f_{G_*}(\boldsymbol{X})\|^2 \right\}.$$

Putting the above results together leads to

$$1 - \exp\left\{ -\frac{1}{8\sigma^2}\|f_{\widehat{G}_n}(\boldsymbol{X}) - f_{G_*}(\boldsymbol{X})\|^2 \right\} = \mathcal{O}_P(\log(n)/n).$$

Hence, for sufficiently large $n$, for some universal constant $C$ the above equality implies

$$\|f_{\widehat{G}_n}(\boldsymbol{X}) - f_{G_*}(\boldsymbol{X})\|^2 \leq 8\sigma^2 \log\left( \frac{1}{1 - C\log(n)/n} \right)$$

$$= 8\sigma^2 \log\left( 1 + \frac{C\log(n)/n}{1 - C\log(n)/n} \right)$$

$$\leq 8\sigma^2 \cdot \frac{C\log(n)/n}{1 - C\log(n)/n}$$

$$\leq 16\sigma^2 C \log(n)/n.$$

This is equivalent to

$$\|f_{\widehat{G}_n}(\boldsymbol{X}) - f_{G_*}(\boldsymbol{X})\| = \mathcal{O}_P(\sqrt{\log(n)/n}).$$

Consequently

$$\|f_{\widehat{G}_n} - f_{G_*}\|_{L_2(\mu)} = \left( \int_{\mathcal{X}} \|f_{\widehat{G}_n}(\boldsymbol{X}) - f_{G_*}(\boldsymbol{X})\|^2 d\mu(\boldsymbol{X}) \right)^{1/2} = \mathcal{O}_P(\sqrt{\log(n)/n}).$$

The proof of the proposition is completed.

# B  ADDITIONAL THEORETICAL RESULTS FOR VISUAL ADAPTIVE PROMPT TUNING

Thus far, in Section 5, we have provided theoretical benefits of visual adaptive prompt tuning when the function $\sigma$ in Equation (12) is nonlinear. In this appendix, we demonstrate that the visual adaptive prompt tuning also has appealing theoretical properties when the function $\sigma$ is linear.

Similar to the setting considered in Section 5, we provide a theoretical guarantee for the linear setting of the visual adaptive prompt tuning via the regression framework. For empirical results, please refer to Appendix E.6.

**Problem Setup.** We assume that the i.i.d. samples of size $n$: $(\boldsymbol{X}_1, Y_1), (\boldsymbol{X}_2, Y_2), \ldots, (\boldsymbol{X}_n, Y_n) \in \mathbb{R}^d \times \mathbb{R}^{d'}$ are generated from the model:

$$Y_i = f_{\widetilde{G}_*}(\boldsymbol{X}_i) + \varepsilon_i, \quad i = 1, 2, \ldots, n, \tag{29}$$

where $\varepsilon_1, \ldots, \varepsilon_n$ are independent Gaussian noise variables such that $\mathbb{E}[\varepsilon_i | \boldsymbol{X}_i] = 0$ and $\mathrm{Var}(\varepsilon_i | \boldsymbol{X}_i) = \nu^2 I_{d'}$ for all $1 \leq i \leq n$. Additionally, we assume that $\boldsymbol{X}_1, \boldsymbol{X}_2, \ldots, \boldsymbol{X}_n$ are i.i.d. samples from some probability distribution $\mu$. The regression function $f_{\widetilde{G}_*}(\cdot)$ in Equation (29) then takes the form

$$f_{\widetilde{G}_*}(\boldsymbol{X}) := \sum_{j=1}^{N} \frac{\exp(\boldsymbol{X}^\top A_j^0 \boldsymbol{X} + a_j^0)}{\tilde{D}_f(\boldsymbol{X})} \cdot h(\boldsymbol{X}, \eta_j^0)$$

$$+ \sum_{j'=1}^{L} \frac{\exp((BW_{*,2}W_{*,1j'}\boldsymbol{X})^\top \boldsymbol{X} + b_{*,j'})}{\tilde{D}_f(\boldsymbol{X})} \cdot CW_{*,2}W_{*,1j'}\boldsymbol{X}, \tag{30}$$

with $N$ pre-trained experts and $L$ unknown experts, where $\tilde{D}_{f,\widetilde{G}_*}(\boldsymbol{X}) := \sum_{k=1}^{N} \exp(\boldsymbol{X}^\top A_k^0 \boldsymbol{X} + a_k^0) + \sum_{j'=1}^{L} \exp((BW_{*,2}W_{*,1j'}\boldsymbol{X})^\top \boldsymbol{X} + b_{*,j'})$, while $\widetilde{G}_* := \sum_{j'=1}^{L} \exp(b_{*,j'})\delta_{W_{*,2}W_{*,1j'}}$ denotes a *mixing measure, i.e.,* a weighted sum of Dirac measures $\delta$, associated with unknown parameters $(b_{*,j'}, W_{*,2}W_{*,1j'})_{j'=1}^{L}$ in the parameter space $\Theta \subset \mathbb{R} \times \mathbb{R}^{d \times d}$. At the same time, the values of the matrix $A_j^0$, the expert parameter $\eta_j^0$, and the bias parameter $a_j^0$ are known for all $1 \leq j \leq N$. Additionally, the matrices $B \in \mathbb{R}^{d \times d}$ and $C \in \mathbb{R}^{d' \times d}$ are given.

**Least-Squares Estimator.** In particular, we consider the estimator

$$\widetilde{G}_n := \arg\min_{G \in \mathcal{G}_{L'}(\Theta)} \sum_{i=1}^{n} \left(Y_i - f_G(\boldsymbol{X}_i)\right)^2, \tag{31}$$

**Voronoi Loss.** The Voronoi loss tailored to the setting in Equation (30) is defined as

$$\mathcal{D}_2(G, \widetilde{G}_*) := \sum_{j'=1}^{L} \left| \sum_{i \in \mathcal{V}_{j'}} \exp(b_i) - \exp(b_{*,j'}) \right| + \sum_{j' \in [L]: |\mathcal{V}_{j'}|=1} \sum_{i \in \mathcal{V}_{j'}} \exp(b_i) \|\Delta W_2 W_{1ij'}\|$$

$$+ \sum_{j' \in [L]: |\mathcal{V}_{j'}|>1} \sum_{i \in \mathcal{V}_{j'}} \exp(b_i) \|\Delta W_2 W_{1ij'}\|^2,$$

where we denote $\Delta W_2 W_{1ij'} := W_2 W_{1i} - W_{*,2}W_{*,1j'}$ for any $i, j'$. Equipped with this loss function, we now provide the convergence rate of prompt estimation for the setting in Equation (30) in Theorem 2.

*Theorem* 2. Given the least-squares estimator $\widetilde{G}_n$ defined in Equation (31), we have that

$$\mathcal{D}_2(\widetilde{G}_n, \widetilde{G}_*) = \mathcal{O}_P([\log(n)/n]^{\frac{1}{2}}).$$

*Proof of Theorem 2.* Based on the convergence rate of regression function estimation presented in Proposition 1, our objective is to establish the following inequality:

$$\inf_{G \in \mathcal{G}_{L'}(\Theta)} \|f_G - f_{G_*}\|_{L_2(\mu)} / \mathcal{D}_2(G, G_*) > 0. \tag{32}$$

We partition the proof of the above inequality into local and global parts.

### B.0.1 Local Part

The local part of the inequality (32) corresponds to the following condition:

$$\lim_{\varepsilon \to 0} \inf_{G \in \mathcal{G}_{L'}(\Theta): \mathcal{D}_2(G, \widetilde{G}_*) \leq \varepsilon} \frac{\|f_G - f_{\widetilde{G}_*}\|_{L_2(\mu)}}{\mathcal{D}_2(G, \widetilde{G}_*)} > 0.$$

We assume that the above inequality does not hold. Then, we can find a sequence of measures $G_n := \sum_{j'=1}^{L'} \exp(b_{n,j'}) \delta_{W_{n,2} W_{n,1j'}}$ in $\mathcal{G}_{L'}(\Theta)$ such that as $n \to \infty$, we have

$$\begin{cases} \mathcal{D}_{2n} := \mathcal{D}_2(G_n, \widetilde{G}_*) \to 0, \\ \|f_{G_n} - f_{\widetilde{G}_*}\|_{L_2(\mu)} / \mathcal{D}_{2n} \to 0. \end{cases}$$

Similar to the proof of Theorem 1, for clarity of presentation, $\mathcal{V}_j^n := \mathcal{V}_j(G_n)$ denote the Voronoi cell of $G_n$ generated by the $j$-th component of the true measure $\widetilde{G}_*$. Since the ensuing arguments are asymptotic, without loss of generality we assume that these Voronoi cells do not depend on the sample size, *i.e.,* we have $\mathcal{V}_j = \mathcal{V}_j^n$ for all $n$ and $1 \leq j \leq L$. Therefore, the Voronoi loss $\mathcal{D}_{2n}$ can be rewritten as follows:

$$\mathcal{D}_{2n} := \sum_{j'=1}^{L} \Big| \sum_{i \in \mathcal{V}_{j'}} \exp(b_{n,i}) - \exp(b_{*,j'}) \Big| + \sum_{j' \in [L]: |\mathcal{V}_{j'}|=1} \sum_{i \in \mathcal{V}_{j'}} \exp(b_{n,i}) \|\Delta W_{n,2} W_{n,1ij'}\|$$
$$+ \sum_{j' \in [L]: |\mathcal{V}_{j'}|>1} \sum_{i \in \mathcal{V}_{j'}} \exp(b_{n,i}) \|\Delta W_{n,2} W_{n,1ij'}\|^2,$$

where we define $\Delta W_{n,2} W_{n,1ij'} := W_{n,2} W_{n,1i} - W_{*,2} W_{*,1j'}$ for all $i \in \mathcal{V}_{j'}$.

From the hypothesis that $\mathcal{D}_{2n} \to 0$ as $n$ approaches $\infty$, we find that $\sum_{i \in \mathcal{V}_j} \exp(b_{n,i}) \to \exp(b_{*,j})$ and $W_{n,2} W_{n,1i} \to W_{*,2} W_{*,1j'}$ for any $i \in \mathcal{V}_j$ and $j \in [L]$. Throughout this proof, we assume WLOG that $B = Id_d$, $C = Id_d$ and $r = 1$ for simplicity; we note that our techniques can be generalized to the general case. We partition the proof for the local part into three steps, as detailed below:

**Step 1 - Taylor expansion.**  We first define the following function:

$$Q_n(\boldsymbol{X}) := \Big[ \sum_{j=1}^{N} \exp(\boldsymbol{X}^\top A_j^0 \boldsymbol{X} + a_j^0) + \sum_{j'=1}^{L} \exp((W_{*,2} W_{*,1j'} \boldsymbol{X})^\top \boldsymbol{X} + b_{*,j'}) \Big] \cdot [f_{G_n}(\boldsymbol{X}) - f_{\widetilde{G}_*}(\boldsymbol{X})].$$

Then, we can decompose the function $Q_n(\boldsymbol{X})$ as follows:

$$Q_n(\boldsymbol{X}) = \sum_{j=1}^{L} \sum_{i \in \mathcal{V}_j} \exp(b_{n,i}) \Big[ \exp((W_{n,2} W_{n,1i} \boldsymbol{X})^\top \boldsymbol{X}) W_{n,2} W_{n,1i} \boldsymbol{X} - \exp((W_{*,2} W_{*,1j} \boldsymbol{X})^\top \boldsymbol{X}) W_{*,2} W_{*,1j} \boldsymbol{X} \Big]$$
$$- \sum_{j=1}^{L} \sum_{i \in \mathcal{V}_j} \exp(b_{n,i}) \Big[ \exp((W_{n,2} W_{n,1i} \boldsymbol{X})^\top \boldsymbol{X}) - \exp((W_{*,2} W_{*,1j} \boldsymbol{X})^\top \boldsymbol{X}) \Big] f_{G_n}(\boldsymbol{X})$$
$$+ \sum_{j=1}^{L} \Big( \sum_{i \in \mathcal{V}_j} \exp(b_{n,i}) - \exp(b_{*,j}) \Big) \exp((W_{*,2} W_{*,1j} \boldsymbol{X})^\top \boldsymbol{X}) \Big[ W_{*,2} W_{*,1j} \boldsymbol{X} - f_{G_n}(\boldsymbol{X}) \Big]$$
$$:= \tilde{A}_n(\boldsymbol{X}) - \tilde{B}_n(\boldsymbol{X}) + \tilde{C}_n(\boldsymbol{X}). \tag{33}$$

**Decomposition of the function $\tilde{A}_n(\boldsymbol{X})$.**  To streamline the argument, we define the following functions: $\tilde{E}(\boldsymbol{X}; W_2 W_1) := \exp((W_2 W_1 \boldsymbol{X})^\top \boldsymbol{X})$ and $\tilde{H}(\boldsymbol{X}; W_2 W_1) = W_2 W_1 \boldsymbol{X}$. The product of the functions $\tilde{E}$ and $\tilde{H}$ is defined as $\tilde{F}(\boldsymbol{X}; W_2 W_1) = \tilde{E}(\boldsymbol{X}; W_2 W_1) \tilde{H}(\boldsymbol{X}; W_2 W_1)$. To account for the differences in the number of components within each Voronoi cells, we further decompose the

function $A_n(\boldsymbol{X})$ as follows:

$$\tilde{A}_n(\boldsymbol{X}) = \sum_{j:|\mathcal{V}_j|=1} \sum_{i \in \mathcal{V}_j} \exp(b_{n,i}) \Big[ \tilde{F}(\boldsymbol{X}; W_{n,2}W_{n,1i}) - \tilde{F}(\boldsymbol{X}; W_{*,2}W_{*,1j}) \Big]$$

$$+ \sum_{j:|\mathcal{V}_j|>1} \sum_{i \in \mathcal{V}_j} \exp(b_{n,i}) \Big[ \tilde{F}(\boldsymbol{X}; W_{n,2}W_{n,1i}) - \tilde{F}(\boldsymbol{X}; W_{*,2}W_{*,1j}) \Big]$$

$$:= \tilde{A}_{n,1}(\boldsymbol{X}) + \tilde{A}_{n,2}(\boldsymbol{X})$$

An application of the first-order Taylor expansion to the functions $\tilde{E}$ and $\tilde{H}$ leads to:

$$\tilde{E}(\boldsymbol{X}; W_{n,2}W_{n,1i}) = \tilde{E}(\boldsymbol{X}; W_{*,2}W_{*,1j}) + \sum_{|\alpha|=1} (\Delta W_{n,2}W_{n,1ij})^\alpha \frac{\partial^{|\alpha|}\tilde{E}}{\partial(W_2W_1)^\alpha}(\boldsymbol{X}; W_{*,2}W_{*,1j}) + \tilde{R}_{ij,1}(\boldsymbol{X}),$$

$$\tilde{H}(\boldsymbol{X}; W_{n,2}W_{n,1i}) = \tilde{H}(\boldsymbol{X}; W_{*,2}W_{*,1j}) + \sum_{|\alpha|=1} (\Delta W_{n,2}W_{n,1ij})^\alpha \frac{\partial^{|\alpha|}\tilde{H}}{\partial(W_2W_1)^\alpha}(\boldsymbol{X}; W_{*,2}W_{*,1j}) + \tilde{R}_{ij,2}(\boldsymbol{X}),$$

Here, the indices $i, j$ in these equations satisfy that $|\mathcal{V}_j| = 1$, i.e., Voronoi cells with exactly one element and $i \in \mathcal{V}_j$. Furthermore, the functions $\tilde{R}_{ij,1}(\boldsymbol{X})$ and $\tilde{R}_{ij,2}(\boldsymbol{X})$ in these equations correspond to Taylor remainders when expanding the functions $\tilde{E}$ and $\tilde{H}$. Putting the above results together leads to:

$$\tilde{A}_{n,1}(\boldsymbol{X}) = \sum_{j:|\mathcal{V}_j|=1} \sum_{i \in \mathcal{V}_j} \frac{\exp(b_{n,i})}{\alpha!} \sum_{|\alpha|=1} \Bigg\{ (\Delta W_{n,2}W_{n,1ij})^\alpha \frac{\partial^{|\alpha|}\tilde{E}}{\partial(W_2W_1)^\alpha}(\boldsymbol{X}; W_{*,2}W_{*,1j})\tilde{H}(\boldsymbol{X}; W_{*,2}W_{*,1j})$$

$$+ (\Delta W_{n,2}W_{n,1ij})^\alpha \frac{\partial^{|\alpha|}\tilde{H}}{\partial(W_2W_1)^\alpha}(\boldsymbol{X}; W_{*,2}W_{*,1j})\tilde{E}(\boldsymbol{X}; W_{*,2}W_{*,1j}) \Bigg\} + \hat{R}_{n,1}(\boldsymbol{X})$$

$$= \sum_{j:|\mathcal{V}_j|=1} \sum_{|\alpha|=1} \Bigg\{ \tilde{M}_{n,j,\alpha} \frac{\partial^{|\alpha|}\tilde{E}}{\partial(W_2W_1)^\alpha}(\boldsymbol{X}; W_{*,2}W_{*,1j})\tilde{H}(\boldsymbol{X}; W_{*,2}W_{*,1j})$$

$$+ \tilde{M}_{n,j,\alpha} \frac{\partial^{|\alpha|}\tilde{H}}{\partial(W_2W_1)^\alpha}(\boldsymbol{X}; W_{*,2}W_{*,1j})\tilde{E}(\boldsymbol{X}; W_{*,2}W_{*,1j}) \Bigg\} + \hat{R}_{n,1}(\boldsymbol{X})$$

where $\alpha = (\alpha_1, \alpha_2)$. Furthermore, due to the uniform smoothness of the functions $\tilde{E}$ and $\tilde{H}$, the remainder term $\hat{R}_{n,1}(\boldsymbol{X})$ from the preceding expansion satisfies $\hat{R}_{n,1}(\boldsymbol{X})/\mathcal{D}_{2n} \to 0$ when $n$ approaches infinity. Finally, for any $|\alpha| = 1$, the terms $\tilde{M}_{n,j,\alpha}$ in the same expansion are given by:

$$\tilde{M}_{n,j,\alpha} = \sum_{i \in \mathcal{V}_j} \frac{\exp(b_{n,i})}{\alpha!}(\Delta W_{n,2}W_{n,1ij})^\alpha.$$

We now turn to the decomposition of $\tilde{A}_{n,2}(\boldsymbol{X})$. Unlike the analysis of $\tilde{A}_{n,1}(\boldsymbol{X})$, which involved a first-order Taylor expansion, to account for more than one element in the Voronoi cells in $\tilde{A}_{n,2}(\boldsymbol{X})$, we resort to the second-order Taylor expansions to the functions $\tilde{E}$ and $\tilde{H}$. Specifically, applying a second-order Taylor expansion yields:

$$\tilde{A}_{n,2}(\boldsymbol{X}) = \sum_{j:|\mathcal{V}_j|>1} \sum_{1 \le |\alpha| \le 2} \Bigg\{ \tilde{M}_{n,j,\alpha} \frac{\partial^{|\alpha|}\tilde{E}}{\partial(W_2W_1)^\alpha}(\boldsymbol{X}; W_{*,2}W_{*,1j})\tilde{H}(\boldsymbol{X}; W_{*,2}W_{*,1j})$$

$$+ \tilde{M}_{n,j,\alpha} \frac{\partial^{|\alpha|}\tilde{H}}{\partial(W_2W_1)^\alpha}(\boldsymbol{X}; W_{*,2}W_{*,1j})\tilde{E}(\boldsymbol{X}; W_{*,2}W_{*,1j}) \Bigg\}$$

$$+ \sum_{|\alpha|=1,|\beta|=1} \tilde{M}_{n,j,\alpha,\beta} \frac{\partial^{|\alpha|}\tilde{E}}{\partial(W_2W_1)^\alpha}(\boldsymbol{X}; W_{*,2}W_{*,1j}) \frac{\partial^{|\beta|}\tilde{H}}{\partial(W_2W_1)^\beta}(\boldsymbol{X}; W_{*,2}W_{*,1j}) + \hat{R}_{n,2}(\boldsymbol{X})$$

where $\alpha = (\alpha_1, \alpha_2)$, $\beta = (\beta_1, \beta_2)$. Due to the uniform smoothness of the functions $\tilde{E}$ and $\tilde{H}$, the term $\hat{R}_{n,2}(\boldsymbol{X})$, representing the combined Taylor remainders from this second-order expansion,

satisfies $\hat{R}_{n,2}(\boldsymbol{X})/\mathcal{D}_{2n} \to 0$ as $n$ goes to infinity. The coefficients $\tilde{M}_{n,j,\alpha}$ and $\tilde{M}_{n,j,\alpha,\beta}$ in the expansion above are defined as follows:

$$\tilde{M}_{n,j,\alpha} = \sum_{i \in \mathcal{V}_j} \frac{\exp(b_{n,i})}{\alpha!}(\Delta W_{n,2} W_{n,1ij})^{\alpha},$$

for any coefficient $\alpha = (\alpha_1, \alpha_2)$ such that $|\alpha| = 2$ and

$$\tilde{M}_{n,j,\alpha,\beta} = \sum_{i \in \mathcal{V}_j} \frac{\exp(b_{n,i})}{\alpha!\beta!}(\Delta W_{n,2} W_{n,1ij})^{\alpha+\beta},$$

for any coefficients $\alpha = (\alpha_1, \alpha_2)$ and $\beta = (\beta_1, \beta_2)$ such that $|\alpha| = |\beta| = 1$. Given the definitions of $\tilde{E}(\boldsymbol{X}; W_1, W_2)$ and $\tilde{H}(\boldsymbol{X}; W_1, W_2)$, their partial derivatives are computed as follows:

$$\frac{\partial \tilde{E}}{\partial (W_2 W_1)^{(u_1 v_1)}}(\boldsymbol{X}; W_2 W_1) = \boldsymbol{X}^{(u_1)} \boldsymbol{X}^{(v_1)} \exp((W_2 W_1 \boldsymbol{X})^{\top} \boldsymbol{X})$$

$$\frac{\partial^2 \tilde{E}}{\partial (W_2 W_1)^{(u_1 v_1)} \partial (W_2 W_1)^{(u_2 v_2)}}(\boldsymbol{X}; W_2 W_1) = \boldsymbol{X}^{(u_1)} \boldsymbol{X}^{(v_1)} \boldsymbol{X}^{(u_2)} \boldsymbol{X}^{(v_2)} \exp((W_2 W_1 \boldsymbol{X})^{\top} \boldsymbol{X}),$$

$$\frac{\partial \tilde{H}}{\partial (W_2 W_1)^{(u_1 v_1)}}(\boldsymbol{X}; W_2 W_1) = \boldsymbol{X}^{(v_1)} e_{u_1},$$

$$\frac{\partial^2 \tilde{H}}{\partial (W_2 W_1)^{(u_1 v_1)} \partial (W_2 W_1)^{(u_2 v_2)}}(\boldsymbol{X}; W_2 W_1) = \boldsymbol{0}_d.$$

Putting the above formulations together, the functions $\tilde{A}_{n,1}(\boldsymbol{X})$ and $\tilde{A}_{n,2}(\boldsymbol{X})$ can be rewritten as follows:

$$\tilde{A}_{n,1}(\boldsymbol{X}) = \sum_{j:|\mathcal{V}_j|=1} \exp((W_{*,2} W_{*,1j} \boldsymbol{X})^{\top} \boldsymbol{X}) \Big[ \sum_{u_1,v_1=1}^{d} M_{n,j,e_{u_1 v_1}} \boldsymbol{X}^{(u_1)} \boldsymbol{X}^{(v_1)} W_{*,2} W_{*,1j} \boldsymbol{X}$$

$$+ \sum_{u_1,v_1=1}^{d} M_{n,j,e_{u_1 v_1}} \boldsymbol{X}^{(v_1)} e_{u_1} \Big] + \hat{R}_{n,1}(\boldsymbol{X}),$$

$$\tilde{A}_{n,2}(\boldsymbol{X}) = \sum_{j:|\mathcal{V}_j|>1} \exp((W_{*,2} W_{*,1j} \boldsymbol{X})^{\top} \boldsymbol{X}) \Big[ \sum_{u_1,v_1=1}^{d} M_{n,j,e_{u_1 v_1}} \boldsymbol{X}^{(u_1)} \boldsymbol{X}^{(v_1)} W_{*,2} W_{*,1j} \boldsymbol{X}$$

$$+ \sum_{u_1,v_1=1}^{d} M_{n,j,e_{u_1 v_1}} \boldsymbol{X}^{(v_1)} e_{u_1} + \sum_{u_1,v_1=1}^{d} \sum_{u_2,v_2=1}^{d} M_{n,j,e_{u_1 v_1}+e_{u_2 v_2}} \boldsymbol{X}^{(u_1)} \boldsymbol{X}^{(v_1)} \boldsymbol{X}^{(u_2)} \boldsymbol{X}^{(v_2)} W_{*,2} W_{*,1j} \boldsymbol{X}$$

$$+ \sum_{u_1,v_1=1}^{d} \sum_{u_2,v_2=1}^{d} M_{n,j,e_{u_1 v_1},e_{u_2 v_2}} \boldsymbol{X}^{(u_1)} \boldsymbol{X}^{(v_1)} \boldsymbol{X}^{(v_2)} e_{u_2} \Big] + \hat{R}_{n,2}(\boldsymbol{X})$$

Here, $e_u$ denotes the standard basis vector in $\mathbb{R}^d$ with 1 in the $u$-th position and 0 in all other positions for any $1 \le u \le d$, while $e_{uv}$ denotes the matrix in $\mathbb{R}^{d \times d}$ whose $uv$-th entry is 1 and all other entries are 0.

**Decomposition of the function $\tilde{B}_n(\boldsymbol{X})$.** Similar to the decomposition of $\tilde{A}_n(\boldsymbol{X})$, we also separate the Voronoi cells with exactly one element and with more than one element in the decomposition of the function $\tilde{B}_n(\boldsymbol{X})$. Therefore, we obtain

$$\tilde{B}_n(\boldsymbol{X}) = \sum_{j:|\mathcal{V}_j|=1} \sum_{i \in \mathcal{V}_j} \exp(b_{n,i}) \Big[ \tilde{E}(\boldsymbol{X}; W_{n,2} W_{n,1i}) - \tilde{E}(\boldsymbol{X}; W_{*,2} W_{*,1j}) \Big] f_{G_n}(\boldsymbol{X})$$

$$+ \sum_{j:|\mathcal{V}_j|>1} \sum_{i \in \mathcal{V}_j} \exp(b_{n,i}) \Big[ \tilde{E}(\boldsymbol{X}; W_{n,2} W_{n,1i}) - \tilde{E}(\boldsymbol{X}; W_{*,2} W_{*,1j}) \Big] f_{G_n}(\boldsymbol{X})$$

$$:= \tilde{B}_{n,1}(\boldsymbol{X}) + \tilde{B}_{n,2}(\boldsymbol{X}).$$

For terms corresponding to Voronoi cells with exactly one component, we use a first-order Taylor expansion. For cells with more than one component, we use a second-order Taylor expansion. This strategy yields the following representations:

$$\tilde{B}_{n,1}(\boldsymbol{X}) = \sum_{j:|\mathcal{V}_j|=1} \sum_{|\alpha|=1} \tilde{M}_{n,j,\alpha} \frac{\partial^{|\alpha|}\tilde{E}}{\partial(W_2 W_1)^{\alpha}}(\boldsymbol{X}; W_{*,2}W_{*,1j}) f_{G_n}(\boldsymbol{X}) + \tilde{R}_{n,3}(\boldsymbol{X}),$$

$$\tilde{B}_{n,2}(\boldsymbol{X}) = \sum_{j:|\mathcal{V}_j|=1} \sum_{1\le|\alpha|\le 2} \tilde{M}_{n,j,\alpha} \frac{\partial^{|\alpha|}\tilde{E}}{\partial(W_2 W_1)^{\alpha}}(\boldsymbol{X}; W_{*,2}W_{*,1j}) f_{G_n}(\boldsymbol{X}) + \tilde{R}_{n,4}(\boldsymbol{X}).$$

In these expressions, the functions $\tilde{R}_{n,3}(\boldsymbol{X})$ and $\tilde{R}_{n,4}(\boldsymbol{X})$ correspond to the Taylor remainders. Due to the uniform smoothness of the function $\tilde{E}$, we obtain $\tilde{R}_{n,3}(\boldsymbol{X})/\mathcal{D}_{2n} \to 0$ and $\tilde{R}_{n,4}(\boldsymbol{X})/\mathcal{D}_{2n} \to 0$ as $n$ approaches infinity. By computing the closed-form expressions for the partial derivatives of the function $\tilde{E}$, both the functions $\tilde{B}_{n,1}(\boldsymbol{X})$ and $\tilde{B}_{n,2}(\boldsymbol{X})$ can be rewritten as follows:

$$\tilde{B}_{n,1}(\boldsymbol{X}) = \sum_{j:|\mathcal{V}_j|=1} \exp((W_{*,2}W_{*,1j}\boldsymbol{X})^\top \boldsymbol{X}) \Big[ \sum_{u_1,v_1=1}^{d} \tilde{M}_{n,j,e_{u_1 v_1}} \boldsymbol{X}^{(u_1)} \boldsymbol{X}^{(v_1)} \Big] f_{G_n}(\boldsymbol{X}) + \tilde{R}_{n,3}(\boldsymbol{X}),$$

$$\tilde{B}_{n,2}(\boldsymbol{X}) = \sum_{j:|\mathcal{V}_j|>1} \exp((W_{*,2}W_{*,1j}\boldsymbol{X})^\top \boldsymbol{X}) \Big[ \sum_{u_1,v_1=1}^{d} \tilde{M}_{n,j,e_{u_1 v_1}} \boldsymbol{X}^{(u_1)} \boldsymbol{X}^{(v_1)}$$

$$+ \sum_{u_1,v_1=1}^{d} \sum_{u_2,v_2=1}^{d} \tilde{M}_{n,j,e_{u_1 v_1}} \boldsymbol{X}^{(u_1)} \boldsymbol{X}^{(v_1)} \boldsymbol{X}^{(u_2)} \boldsymbol{X}^{(v_2)} \Big] f_{G_n}(\boldsymbol{X}) + \tilde{R}_{n,4}(\boldsymbol{X}),$$

Plugging all of these results together, the function $Q_n(\boldsymbol{X})$ can be rewritten as follows:

$$Q_n(\boldsymbol{X}) = \sum_{j:|\mathcal{V}_j|=1} \exp((W_{*,2}W_{*,1j}\boldsymbol{X})^\top \boldsymbol{X}) \Big[ \sum_{u_1,v_1=1}^{d} \tilde{M}_{n,j,e_{u_1 v_1}} \boldsymbol{X}^{(u_1)} \boldsymbol{X}^{(v_1)} W_{*,2}W_{*,1j}\boldsymbol{X} + \sum_{u_1,v_1=1}^{d} \tilde{M}_{n,j,e_{u_1 v_1}} \boldsymbol{X}^{(v_1)} e_{u_1} \Big]$$

$$+ \sum_{j:|\mathcal{V}_j|>1} \exp((W_{*,2}W_{*,1j}\boldsymbol{X})^\top \boldsymbol{X}) \Big[ \sum_{u_1,v_1=1}^{d} \tilde{M}_{n,j,e_{u_1 v_1}} \boldsymbol{X}^{(u_1)} \boldsymbol{X}^{(v_1)} W_{*,2}W_{*,1j}\boldsymbol{X} + \sum_{u_1,v_1=1}^{d} \tilde{M}_{n,j,e_{u_1 v_1}} \boldsymbol{X}^{(v_1)} e_{u_1}$$

$$+ \sum_{u_1,v_1=1}^{d} \sum_{u_2,v_2=1}^{d} \tilde{M}_{n,j,e_{u_1 v_1}+e_{u_2 v_2}} \boldsymbol{X}^{(u_1)} \boldsymbol{X}^{(v_1)} \boldsymbol{X}^{(u_2)} \boldsymbol{X}^{(v_2)} W_{*,2}W_{*,1j}\boldsymbol{X}$$

$$+ \sum_{u_1,v_1=1}^{d} \sum_{u_2,v_2=1}^{d} \tilde{M}_{n,j,e_{u_1 v_1},e_{u_2 v_2}} \boldsymbol{X}^{(u_1)} \boldsymbol{X}^{(v_1)} \boldsymbol{X}^{(v_2)} e_{u_2} \Big]$$

$$- \sum_{j:|\mathcal{V}_j|=1} \exp((W_{*,2}W_{*,1j}\boldsymbol{X})^\top \boldsymbol{X}) \Big[ \sum_{u_1,v_1=1}^{d} \tilde{M}_{n,j,e_{u_1 v_1}} \boldsymbol{X}^{(u_1)} \boldsymbol{X}^{(v_1)} \Big] f_{G_n}(\boldsymbol{X})$$

$$- \sum_{j:|\mathcal{V}_j|>1} \exp((W_{*,2j}W_{*,1j}\boldsymbol{X})^\top \boldsymbol{X}) \Big[ \sum_{u_1,v_1=1}^{d} \tilde{M}_{n,j,e_{u_1 v_1}} \boldsymbol{X}^{(u_1)} \boldsymbol{X}^{(v_1)}$$

$$+ \sum_{u_1,v_1=1}^{d} \sum_{u_2,v_2=1}^{d} \tilde{M}_{n,j,e_{u_1 v_1}} \boldsymbol{X}^{(u_1)} \boldsymbol{X}^{(v_1)} \boldsymbol{X}^{(u_2)} \boldsymbol{X}^{(v_2)} \Big] f_{G_n}(\boldsymbol{X})$$

$$- \sum_{j=1}^{L} \tilde{N}_{n,j} \exp((W_{*,2}W_{*,1j}\boldsymbol{X})^\top \boldsymbol{X}) f_{G_n}(\boldsymbol{X})$$

$$+ \sum_{j=1}^{L} \tilde{N}_{n,j} \exp((W_{*,2}W_{*,1j}\boldsymbol{X})^\top \boldsymbol{X}) W_{*,2}W_{*,1j}\boldsymbol{X}$$

$$+ \hat{R}_{n,1}(\boldsymbol{X}) + \hat{R}_{n,2}(\boldsymbol{X}) - \tilde{R}_{n,3}(\boldsymbol{X}) - \tilde{R}_{n,4}(\boldsymbol{X}) \tag{34}$$

where $\tilde{N}_{n,j} := \sum_{i\in\mathcal{V}_j} \exp(b_{n,i}) - \exp(b_{*,j})$ for any $j \in [L]$.

**Step 2 - Non-vanishing coefficients.** An important insight from Equation (34) is that the ratio $Q_n(\boldsymbol{X})/\mathcal{D}_{1n}$ can be expressed as a linear combination of the following independent functions:

$$E(\boldsymbol{X}; W_{*,2}W_{*,1j})\boldsymbol{X}^{(u_1)}\boldsymbol{X}^{(v_1)}W_{*,2}W_{*,1j}\boldsymbol{X}, \quad E(\boldsymbol{X}; W_{*,2}W_{*,1j})\boldsymbol{X}^{(v_1)}e_{u_1},$$

$$E(\boldsymbol{X}; W_{*,2}W_{*,1j})\boldsymbol{X}^{(u_1)}\boldsymbol{X}^{(v_1)}\boldsymbol{X}^{(u_2)}\boldsymbol{X}^{(v_2)}W_{*,2}W_{*,1j}\boldsymbol{X}, \quad E(\boldsymbol{X}; W_{*,2}W_{*,1j})\boldsymbol{X}^{(u_1)}\boldsymbol{X}^{(v_1)}\boldsymbol{X}^{(v_2)}e_{u_2},$$

$$E(\boldsymbol{X}; W_{*,2}W_{*,1j})\boldsymbol{X}^{(u_1)}\boldsymbol{X}^{(v_1)}f_{G_n}(\boldsymbol{X}), \quad E(\boldsymbol{X}; W_{*,2}W_{*,1j})\boldsymbol{X}^{(u_1)}\boldsymbol{X}^{(v_1)}\boldsymbol{X}^{(u_2)}\boldsymbol{X}^{(v_2)}f_{G_n}(\boldsymbol{X}),$$

$$E(\boldsymbol{X}; W_{*,2}W_{*,1j})f_{G_n}(\boldsymbol{X}), \quad E(\boldsymbol{X}; W_{*,2}W_{*,1j})W_{*,2j}W_{*,1j}\boldsymbol{X},$$

for any $1 \leq j \leq L$, and for $1 \leq u, v, w \leq d$.

We proceed to demonstrate that not all of the coefficients of these linearly independent functions approach 0 as $n$ goes to infinity. Assume, for the sake of contradiction, that all these coefficients approach 0 as $n$ goes to $\infty$. From the representation of the ratio $Q_n(\boldsymbol{X})/\mathcal{D}_{2n}$ in terms of these linearly independent functions, it follows that the ratios $\tilde{M}_{n,j,\alpha}/\mathcal{D}_{2n}$, $\tilde{M}_{n,j,\alpha,\beta}/\mathcal{D}_{2n}$, and $\tilde{N}_{n,j}/\mathcal{D}_{2n}$ approach 0 as $n \to \infty$, for all $\alpha, \beta \in \mathbb{N}^{d \times d}$ such that $1 \leq |\alpha| + |\beta| \leq 2$.

By first considering the condition $\tilde{N}_{n,j}/\mathcal{D}_{2n} \to 0$, it follows that

$$\frac{|\sum_{i \in \mathcal{V}_j} \exp(b_{n,i}) - \exp(b_{*,j})|}{\mathcal{D}_{2n}} = \frac{|\tilde{N}_{n,j}|}{\mathcal{D}_{2n}} \to 0,$$

for any $1 \leq j \leq L$. By varying the index $j$ from 1 to $L$ in these limits and summing them, we find that

$$\frac{\sum_{j=1}^{L} |\sum_{i \in \mathcal{V}_j} \exp(b_{n,i}) - \exp(b_{*,j})|}{\mathcal{D}_{2n}} \to 0. \tag{35}$$

Our strategy now is to consider the limits corresponding to Voronoi cells with exactly one element and more than one element separately. In particular, for Voronoi cells with exactly one element, namely, for indices $j \in [L]$ such that their corresponding Voronoi cells have one element, *i.e.*, $|\mathcal{V}_j| = 1$, as the ratio $\tilde{M}_{n,j,e_{uv}}/\mathcal{D}_{2n} \to 0$, we have

$$\frac{\sum_{i \in \mathcal{V}_j} \exp(b_{n,i})\|\Delta W_{n,2}W_{n,1ij}\|_1}{\mathcal{D}_{2n}} = \frac{\sum_{u,v=1}^{d} |\tilde{M}_{n,j,e_{uv}}|}{\mathcal{D}_{2n}} \to 0.$$

Due to the equivalence between the $\ell_1$ norm and the $\ell_2$ norm, we deduce that

$$\frac{\sum_{j:|\mathcal{V}_j|=1} \sum_{i \in \mathcal{V}_j} \exp(b_{n,i})\|\Delta W_{n,2}W_{n,1ij}\|}{\mathcal{D}_{2n}} \to 0. \tag{36}$$

We now move to the Voronoi cells with more than one element, namely, indices $j \in [L]$ such that $|\mathcal{V}_j| > 1$. Similarly, from the assumption that the ratio $\tilde{M}_{n,j,2e_{uv}}/\mathcal{D}_{2n} \to 0$, we obtain

$$\frac{\sum_{i \in \mathcal{V}_j} \exp(b_{n,i})\|\Delta W_{n,2}W_{n,1ij}\|^2}{\mathcal{D}_{2n}} = \frac{\sum_{u,v=1}^{d} \tilde{M}_{n,j,2e_{uv}}}{\mathcal{D}_{2n}} \to 0. \tag{37}$$

Combining all the results in Equations (35), (36), and (37), we obtain

$$\frac{\mathcal{D}_{2n}}{\mathcal{D}_{2n}} = 1 \to 0, \text{ as } n \to \infty,$$

which is a contradiction. As a consequence, at least one of the coefficients of the linearly independent functions in the expression of the ratio $Q_n(\boldsymbol{X})/\mathcal{D}_{2n}$ does not approach 0 as $n$ approaches infinity.

**Step 3 - Application of Fatou's lemma.** The idea of this step is to divide all of the coefficients of the linearly independent terms in the expression for the ratio $Q_n(\boldsymbol{X})/\mathcal{D}_{2n}$, namely, the terms $\tilde{M}_{n,j,\alpha}/\mathcal{D}_{2n}$, $\tilde{M}_{n,j,\alpha,\beta}/\mathcal{D}_{2n}$, and $\tilde{N}_{n,j}/\mathcal{D}_{2n}$ for all $\alpha, \beta \in \mathbb{N}^{d \times d}$ such that $1 \leq |\alpha| + |\beta| \leq 2$, by the maximum of their absolute values. In particular, we first denote $m_n$ as the maximum of the absolute values of those coefficients. As not all of these coefficients go to 0, it follows that $1/m_n$ does not go to infinity as $n \to \infty$.

From the hypothesis, we have $\|f_{G_n} - f_{\widetilde{G}_*}\|_{L_2(\mu)}/\mathcal{D}_{2n} \to 0$ as $n \to \infty$. Since $1/m_n \not\to \infty$, it follows that $\|f_{G_n} - f_{\widetilde{G}_*}\|_{L_2(\mu)}/(m_n\mathcal{D}_{2n}) \to 0$. An application of Fatou's lemma leads to

$$0 = \lim_{n \to \infty} \frac{\|f_{G_n} - f_{\widetilde{G}_*}\|_{L_2(\mu)}}{m_n\mathcal{D}_{2n}} \geq \int \liminf_{n \to \infty} \frac{\left|f_{G_n}(\boldsymbol{X}) - f_{\widetilde{G}_*}(\boldsymbol{X})\right|}{m_n\mathcal{D}_{2n}} d\mu(\boldsymbol{X}) \geq 0.$$

Combining these results, we obtain $\liminf_{n \to \infty} \dfrac{\left|f_{G_n}(\boldsymbol{X}) - f_{\widetilde{G}_*}(\boldsymbol{X})\right|}{m_n\mathcal{D}_{2n}} = 0$ for almost surely $\boldsymbol{X}$. To simplify the presentation, we define the following limits:

$$\frac{\tilde{M}_{n,j,\alpha}}{m_n\mathcal{D}_{2n}} \to \lambda_{j,\alpha}, \quad \frac{\tilde{M}_{n,j,\alpha,\beta}}{m_n\mathcal{D}_{2n}} \to \xi_{j,\alpha,\beta}, \quad \frac{\tilde{N}_{n,j}}{m_n\mathcal{D}_{2n}} \to \tau_j,$$

for any $1 \leq j \leq L$ and $\alpha, \beta \in \mathbb{N}^{d \times d}$ such that $1 \leq |\alpha| + |\beta| \leq 2$. By the definition of $m_n$, at least one coefficient in the set $\{\lambda_{j,\alpha}, \xi_{j,\alpha,\beta}, \tau_j : j \in [L], \alpha, \beta \in \mathbb{N}^{d \times d} : 1 \leq |\alpha| + |\beta| \leq 2\}$ must be non-zero. Then, the equation $\liminf_{n \to \infty} \dfrac{\left|f_{G_n}(\boldsymbol{X}) - f_{\widetilde{G}_*}(\boldsymbol{X})\right|}{m_n\mathcal{D}_{2n}} = 0$, or equivalently, $\liminf_{n \to \infty} \dfrac{|Q_n(\boldsymbol{X})|}{m_n\mathcal{D}_{2n}} = 0$ leads to

$$\sum_{j:|\mathcal{V}_j|=1} \exp((W_{*,2}W_{*,1j}\boldsymbol{X})^\top \boldsymbol{X})\Big[ \sum_{u_1,v_1=1}^{d} \lambda_{j,e_{u_1v_1}} \boldsymbol{X}^{(u_1)}\boldsymbol{X}^{(v_1)}W_{*,2}W_{*,1j}\boldsymbol{X} + \sum_{u_1,v_1=1}^{d} \lambda_{j,e_{u_1v_1}} \boldsymbol{X}^{(v_1)}e_{u_1} \Big]$$

$$+ \sum_{j:|\mathcal{V}_j|>1} \exp((W_{*,2}W_{*,1j}\boldsymbol{X})^\top \boldsymbol{X})\Big[ \sum_{u_1,v_1=1}^{d} \lambda_{j,e_{u_1v_1}} \boldsymbol{X}^{(u_1)}\boldsymbol{X}^{(v_1)}W_{*,2}W_{*,1j}\boldsymbol{X} + \sum_{u_1,v_1=1}^{d} \lambda_{j,e_{u_1v_1}} \boldsymbol{X}^{(v_1)}e_{u_1}$$

$$+ \sum_{u_1,v_1=1}^{d}\sum_{u_2,v_2=1}^{d} \lambda_{j,e_{u_1v_1}+e_{u_2v_2}} \boldsymbol{X}^{(u_1)}\boldsymbol{X}^{(v_1)}\boldsymbol{X}^{(u_2)}\boldsymbol{X}^{(v_2)}W_{*,2}W_{*,1j}\boldsymbol{X}$$

$$+ \sum_{u_1,v_1=1}^{d}\sum_{u_2,v_2=1}^{d} \xi_{j,e_{u_1v_1},e_{u_2v_2}} \boldsymbol{X}^{(u_1)}\boldsymbol{X}^{(v_1)}\boldsymbol{X}^{(v_2)}e_{u_2} \Big]$$

$$- \sum_{j:|\mathcal{V}_j|=1} \exp((W_{*,2}W_{*,1j}\boldsymbol{X})^\top \boldsymbol{X})\Big[ \sum_{u_1,v_1=1}^{d} \lambda_{j,e_{u_1v_1}} \boldsymbol{X}^{(u_1)}\boldsymbol{X}^{(v_1)} \Big]f_{\widetilde{G}_*}(\boldsymbol{X})$$

$$- \sum_{j:|\mathcal{V}_j|>1} \exp((W_{*,2}W_{*,1j}\boldsymbol{X})^\top \boldsymbol{X})\Big[ \sum_{u_1,v_1=1}^{d} \lambda_{j,e_{u_1v_1}} \boldsymbol{X}^{(u_1)}\boldsymbol{X}^{(v_1)}$$

$$+ \sum_{u_1,v_1=1}^{d}\sum_{u_2,v_2=1}^{d} \xi_{j,e_{u_1v_1},e_{u_2v_2}} \boldsymbol{X}^{(u_1)}\boldsymbol{X}^{(v_1)}\boldsymbol{X}^{(u_2)}\boldsymbol{X}^{(v_2)} \Big]f_{\widetilde{G}_*}(\boldsymbol{X})$$

$$- \sum_{j=1}^{L} \tau_j \exp((W_{*,2}W_{*,1j}\boldsymbol{X})^\top \boldsymbol{X})f_{G_*}(\boldsymbol{X})$$

$$+ \sum_{j=1}^{L} \tau_j \exp((W_{*,2}W_{*,1j}\boldsymbol{X})^\top \boldsymbol{X})W_{*,2}W_{*,1j}\boldsymbol{X} = 0, \tag{38}$$

for almost surely $\boldsymbol{X}$. However, the new equation implies that all the coefficients $\{\lambda_{j,\alpha}, \xi_{j,\alpha,\beta}, \tau_j : j \in [L], \alpha, \beta \in \mathbb{N}^{d \times d} : 1 \leq |\alpha| + |\beta| \leq 2\}$ are 0, which is a contradiction.

As a consequence, we obtain

$$\lim_{\varepsilon \to 0} \inf_{G \in \mathcal{G}_{L'}(\Theta):\mathcal{D}_2(G,\widetilde{G}_*)\leq \varepsilon} \|f_G - f_{\widetilde{G}_*}\|_{L_2(\mu)}/\mathcal{D}_2(G,\widetilde{G}_*) > 0,$$

which proves the conclusion of the local part of the inequality (32).

### B.0.2 Global Part

From the result of the local part of the inequality (32), we can find a positive constant $\varepsilon'$ such that the following inequality holds:

$$\inf_{G \in \mathcal{G}_{L'}(\Theta) : \mathcal{D}_2(G, \widetilde{G}_*) \leq \varepsilon'} \|f_G - f_{\widetilde{G}_*}\|_{L_2(\mu)} / \mathcal{D}_2(G, \widetilde{G}_*) > 0.$$

To obtain the conclusion of the theorem, we only need to demonstrate that

$$\inf_{G \in \mathcal{G}_{L'}(\Theta) : \mathcal{D}_2(G, \widetilde{G}_*) > \varepsilon'} \|f_G - f_{\widetilde{G}_*}\|_{L_2(\mu)} / \mathcal{D}_2(G, \widetilde{G}_*) > 0.$$

We prove the claim by contradiction. Indeed, by assuming the claim does not hold implies that there exists a sequence of $G'_n := \sum_{j'=1}^{L'} \exp(b_{n,j'}) \delta_{W_{n,1j'} W_{n,2}}$ in the set $\mathcal{G}_{L'}(\Theta)$ such that

$$\begin{cases} \mathcal{D}_2(G'_n, \widetilde{G}_*) > \varepsilon' \\ \|f_{G'_n} - f_{\widetilde{G}_*}\|_{L_2(\mu)} / \mathcal{D}_2(G'_n, \widetilde{G}_*) \to 0, \end{cases}$$

as long as $n$ approaches infinity. This implies that $\|f_{G'_n} - f_{\widetilde{G}_*}\|_{L_2(\mu)} \to 0$ as $n$ goes to infinity.

From the hypothesis, the parameter space $\Theta$ is compact. Therefore, a subsequence of $G'_n$'s converges to some mixing measure $G'$ where $G'$ lies in the space $\mathcal{G}_{L'}(\Theta)$. From the hypothesis, we have $\mathcal{D}_2(G'_n, \widetilde{G}_*) > \varepsilon'$. By taking the limit of both sides as $n \to \infty$, we obtain $\mathcal{D}_2(G', G_*) \geq \varepsilon'$.

An application of the Fatou's lemma leads to the following result:

$$0 = \lim_{n \to \infty} \|f_{G'_n} - f_{\widetilde{G}_*}\|_{L_2(\mu)} \geq \int \liminf_{n \to \infty} \left\| f_{G'_n}(\boldsymbol{X}) - f_{\widetilde{G}_*}(\boldsymbol{X}) \right\|^2 d\mu(\boldsymbol{X}).$$

This inequality is only possible if $f_{G'} = f_{\widetilde{G}_*}$ for almost surely $\boldsymbol{X}$.

According to the identifiability of the function $f_G(\boldsymbol{X})$, this equation only holds when $G' \equiv \widetilde{G}_*$. As a consequence, we obtain $\mathcal{D}_2(G', \widetilde{G}_*) = 0$. This contradicts the deduction that $\mathcal{D}_1(G', \widetilde{G}_*) \geq \varepsilon' > 0$. Hence, the proof of the global part is completed. This achieves the conclusion of the theorem.

**Proof for the identifiability property.** The key claim that we aim to show is that if the equation $f_G(\boldsymbol{X}) = f_{\widetilde{G}_*}(\boldsymbol{X})$ for almost every $\boldsymbol{X}$, then $G \equiv \widetilde{G}_*$, namely, the two mixing measures are identical.

From the hypothesis, as $f_G(\boldsymbol{X}) = f_{G_*}(\boldsymbol{X})$ for almost all $\boldsymbol{X}$, we have

$$\sum_{j=1}^{N} \frac{\exp(\boldsymbol{X}^\top A_j^0 \boldsymbol{X} + a_j^0))}{\tilde{D}_{f,G}(\boldsymbol{X})} h(\boldsymbol{X}, \eta_j^0) + \sum_{j'=1}^{\tilde{L}} \frac{\exp((BW_2 W_{1j'} \boldsymbol{X})^\top \boldsymbol{X} + b_{j'})}{\tilde{D}_{f,G}(\boldsymbol{X})} CW_2 W_{1j'} \boldsymbol{X}$$

$$= \sum_{j=1}^{N} \frac{\exp(\boldsymbol{X}^\top A_j^0 \boldsymbol{X} + a_j^0))}{\tilde{D}_{f,\widetilde{G}_*}(\boldsymbol{X})} h(\boldsymbol{X}, \eta_j^0) + \sum_{j'=1}^{L} \frac{\exp((BW_{*,2} W_{*,1j'} \boldsymbol{X})^\top \boldsymbol{X} + b_{*,j'})}{\tilde{D}_{f,\widetilde{G}_*}(\boldsymbol{X})} CW_{*,2} W_{*,1j'} \boldsymbol{X}, \tag{39}$$

where $G = \sum_{j=1}^{\tilde{L}} \exp(b_j) \delta_{W_2 W_{1j}}$. Furthermore, we define

$$\tilde{D}_{f,\widetilde{G}_*}(\boldsymbol{X}) = \sum_{k=1}^{N} \exp(\boldsymbol{X}^\top A_k^0 \boldsymbol{X} + a_k^0) + \sum_{j'=1}^{L} \exp((BW_{*,2} W_{*,1j'} \boldsymbol{X})^\top \boldsymbol{X} + b_{*,j'}),$$

$$\tilde{D}_{f,G}(\boldsymbol{X}) = \sum_{k=1}^{N} \exp(\boldsymbol{X}^\top A_k^0 \boldsymbol{X} + a_k^0) + \sum_{j'=1}^{\tilde{L}} \exp((BW_2 W_{1j'} \boldsymbol{X})^\top \boldsymbol{X} + b_{j'}).$$

That equation implies that the number of atoms of $G$ and $\widetilde{G}_*$ must be identical, namely, we have $L = \tilde{L}$. Therefore, the following result holds

$$\left\{ \frac{\exp((BW_2 W_{1j'} \boldsymbol{X})^\top \boldsymbol{X} + b_{j'})}{\tilde{D}_{f,G}(\boldsymbol{X})} : j' \in [L] \right\} = \left\{ \frac{\exp((BW_{*,2} W_{*,1j'} \boldsymbol{X})^\top \boldsymbol{X} + b_{*,j'})}{\tilde{D}_{f,\widetilde{G}_*}(\boldsymbol{X})} : j' \in [L] \right\},$$

for almost surely $\boldsymbol{X}$. By relabeling the indices of these two sets, we can assume without loss of generality that

$$\frac{\exp((BW_2W_{1j'}\boldsymbol{X})^\top\boldsymbol{X} + b_{j'})}{\tilde{D}_{f,G}(\boldsymbol{X})} = \frac{\exp((BW_{*,2}W_{*,1j'}\boldsymbol{X})^\top\boldsymbol{X} + b_{*,j'})}{\tilde{D}_{f,\widetilde{G}_*}(\boldsymbol{X})},$$

for any index $j' \in [L]$ and for almost surely $\boldsymbol{X}$. From the invariance to translation property of the softmax function, the Equation (39) becomes

$$\sum_{j=1}^{L} \exp\left(b_j\right) \exp((BW_2W_{1j}\boldsymbol{X})^\top\boldsymbol{X})CW_2W_{1j}\boldsymbol{X}$$

$$= \sum_{j'=1}^{L} \exp\left(b_{*,j}\right) \exp((BW_{*,2}W_{*,1j'}\boldsymbol{X})^\top\boldsymbol{X})CW_{*,2}W_{*,1j'}\boldsymbol{X}, \quad (40)$$

for almost surely $\boldsymbol{X}$. This equality suggests that there exists a partition $K_1, K_2, \ldots, K_m$ of the set $[L]$ for some $m$ such that we have $\exp(b_{j_1}) = \exp(b_{*,j_2})$ for any $j_1, j_2 \in K_i$ and for any $i \in [m]$. According to this result, Equation (40) can be rewritten as follows:

$$\sum_{i=1}^{m} \sum_{j_1 \in K_i} \exp\left(b_{j_1}\right) \exp\left((BW_2W_{1j_1}\boldsymbol{X})^\top\boldsymbol{X}\right)CW_2W_{1j_1}\boldsymbol{X}$$

$$= \sum_{i=1}^{m} \sum_{j_2 \in K_i} \exp\left(b_{*,j_2}\right) \exp\left((BW_{*,2}W_{*,1j_2}\boldsymbol{X})^\top\boldsymbol{X}\right)CW_{*,2}W_{*,1j_2}\boldsymbol{X},$$

for almost surely $\boldsymbol{X}$. This equality implies that

$$\{W_2W_{1j_1}\boldsymbol{X} : j_1 \in K_i\} = \{W_{*,2}W_{*,1j_2}\boldsymbol{X} : j_2 \in K_i\},$$

for any $i \in [m]$. This result indicates that

$$\{W_2W_{1j_1} : j_1 \in K_i\} = \{W_{*,2}W_{*,1j_2} : j_2 \in K_i\}.$$

As a consequence, we arrive at the following result:

$$\sum_{i=1}^{m} \sum_{j_1 \in K_i} \exp\left(b_{j_1}\right)\delta_{W_2W_{1j_1}} = \sum_{i=1}^{m} \sum_{j_2 \in K_i} \exp\left(b_{*,j_2}\right)\delta_{W_{*,2}W_{*,1j_2}}.$$

This is equivalent to $G \equiv \widetilde{G}_*$. Consequently, the identifiability property of $f_G$ is established. $\qquad\square$

## C  RELATED WORK

**Mixture of Experts.** Building on classical mixture models with adaptive gating mechanisms (Jacobs et al., 1991; Jordan & Jacobs, 1994; Xu et al., 1994), the MoE model has been extensively studied and refined over the years. Notably, subsequent works (Eigen et al., 2014; Shazeer et al., 2017) introduced the MoE layer as an efficient tool for scaling model capacity. Unlike traditional models that apply uniform parameters to all inputs, the MoE layer applies specific parameter subsets for each input, creating a sparsely activated layer that enables significant capacity growth without proportional increase in computational cost (Fedus et al., 2022; Zhou et al., 2023). This efficiency and scalability have driven its adoption across diverse domains and tasks (Riquelme et al., 2021; Du et al., 2022; Shen et al., 2023).

**Theory of Mixture of Experts.** Although MoEs have been widely used to scale up large models, their theoretical foundations remain under active development. For example, Ho et al. (2022) focused on input-free gating Gaussian MoEs and showed that under maximum likelihood estimation, the experts' convergence rates depend on the algebraic independence of the expert functions. Next, Nguyen et al. (2023; 2024a) established convergence rates for both density and parameter estimation in Softmax-gating Gaussian MoEs, linking these rates to the solvability of polynomial systems under Voronoi-based loss functions. More recently, Nguyen et al. (2024b;c) employed least-squares

estimation to identify conditions under which expert functions are identifiable. Under these conditions, the resulting estimation rates improve substantially.

**Parameter-Efficient Fine-Tuning.** Fine-tuning pre-trained foundational models (Dosovitskiy, 2020; Liu et al., 2021; Kirillov et al., 2023) has become a widely adopted strategy for tackling downstream tasks (Xin et al., 2024). While effective, full fine-tuning is *resource-intensive*, requiring updates to all network parameters and the storage of a separate fine-tuned model for each task (Han et al., 2024). To address these challenges, researchers have increasingly focused on parameter-efficient fine-tuning (PEFT) methods. These methods can be categorized into *partial tuning*, *extra module*, and *prompt tuning*. Partial tuning freezes most of the backbone, fine-tuning only a subset, such as linear heads or a few layers (Mahajan et al., 2018; Chen et al., 2021; He et al., 2022). Extra module methods add trainable parameters to the backbone, such as side structures (Zhang et al., 2020), residual MLP modules in Transformer layers (Houlsby et al., 2019; Cai et al., 2020), or low-rank weight updates (Hu et al., 2021). Despite their promise, partial tuning and extra module methods often face limitations that restrict their applicability. First, they may fail to achieve performance comparable to full fine-tuning (Mahajan et al., 2018; Chen et al., 2021; Jia et al., 2022; He et al., 2023). Second, some methods rely on specific architectural modifications (Rebuffi et al., 2017; Cai et al., 2020; Zhang et al., 2020), limiting their generalizability across different backbone architectures.

**Prompting Methods.** Prompt tuning (Lester et al., 2021), initially proposed for language tasks, offers a simpler yet effective alternative by introducing learnable parameters to the *input sequence* of backbone models, updating only these parameters during fine-tuning (Lester et al., 2021; Jia et al., 2022; Le et al., 2025b;a). Despite its simplicity, prompt tuning has demonstrated significant performance gains. Recently, visual prompt tuning has emerged as a promising paradigm in PEFT techniques for computer vision. Current advancements in visual prompt tuning focus on engineering improvements, such as minimizing parameter usage (Han et al., 2023) and broadening applicability to diverse tasks (Yao et al., 2023; Sohn et al., 2023; Yao et al., 2024). However, the theoretical foundations of prompt-based methods remain underexplored. For instance, He et al. (2021) investigated the relationship between prompt tuning and adapter methods, while Le et al. (2024) analyzed these techniques within the context of MoE models. Furthermore, Petrov et al. (2023) highlighted the limitations of prompting, showing that it cannot alter relative attention patterns and instead biases attention layer outputs in a fixed direction.

**Adaptive Prompting.** A growing body of work investigates adaptive prompting strategies for visual prompt tuning (Zhou et al., 2022). Several approaches employ meta-learning or clustering techniques to tailor prompts to specific downstream datasets. In particular, Huang et al. (2023) and Kim et al. (2023) cluster the downstream data and assign cluster-specific prompts based on the ViT's initial input representation, $X^{(0)}$. Similarly, recent prompt-pool methods (Wang et al., 2022; Kim et al., 2024) first forward the input image through an encoder network to obtain a [CLS] representation, which is then used to select or generate prompts at every attention layer. In all these cases, the prompt experts effectively become functions of the first-layer input, $X^{(0)}$, rather than of the current layer input, $X^{(l)}$.

Our proposed method, VAPT, instead generates prompts at each layer conditioned on the layer-specific representation, $X^{(l)}$. This layer-wise adaptation follows directly from our theoretical framework, which establishes a connection between MoE and VPT and implies that both the experts and their gating functions should depend on $X^{(l)}$. This perspective enables a more principled and analytically grounded examination of model behavior. Importantly, the motivation for VAPT stems from the functional role of prompt experts and their distinction from pre-trained experts, rather than from a purely empirical desire to assign different prompts to different inputs.

Moreover, while some adaptive prompting approaches rely on heuristics or complex image-to-prompt transformations that make their effects difficult to interpret (Huang et al., 2024; Xiao et al., 2025), VAPT adopts a simple yet effective formulation. This design choice leads to the clean expert definitions in Equations (13) and (14), and supports a rigorous theoretical analysis without sacrificing empirical performance—an aspect largely absent in prior work.

Most recently, Zeng et al. (2025) explicitly introduce MoE components into the Transformer architecture, including new routing modules for sparse prompt activation. In contrast, our analysis in Section 3 shows that an implicit MoE structure is already embedded in existing visual prompt tuning frameworks. Consequently, VAPT does not require additional architectural components or routing

mechanisms. Instead, it refines the formulation of prompt experts within this inherent implicit MoE structure. In essence, VAPT enhances the expressive capacity of the underlying prompt experts while leaving the backbone architecture unchanged.

# D  IMPLEMENTATION DETAILS

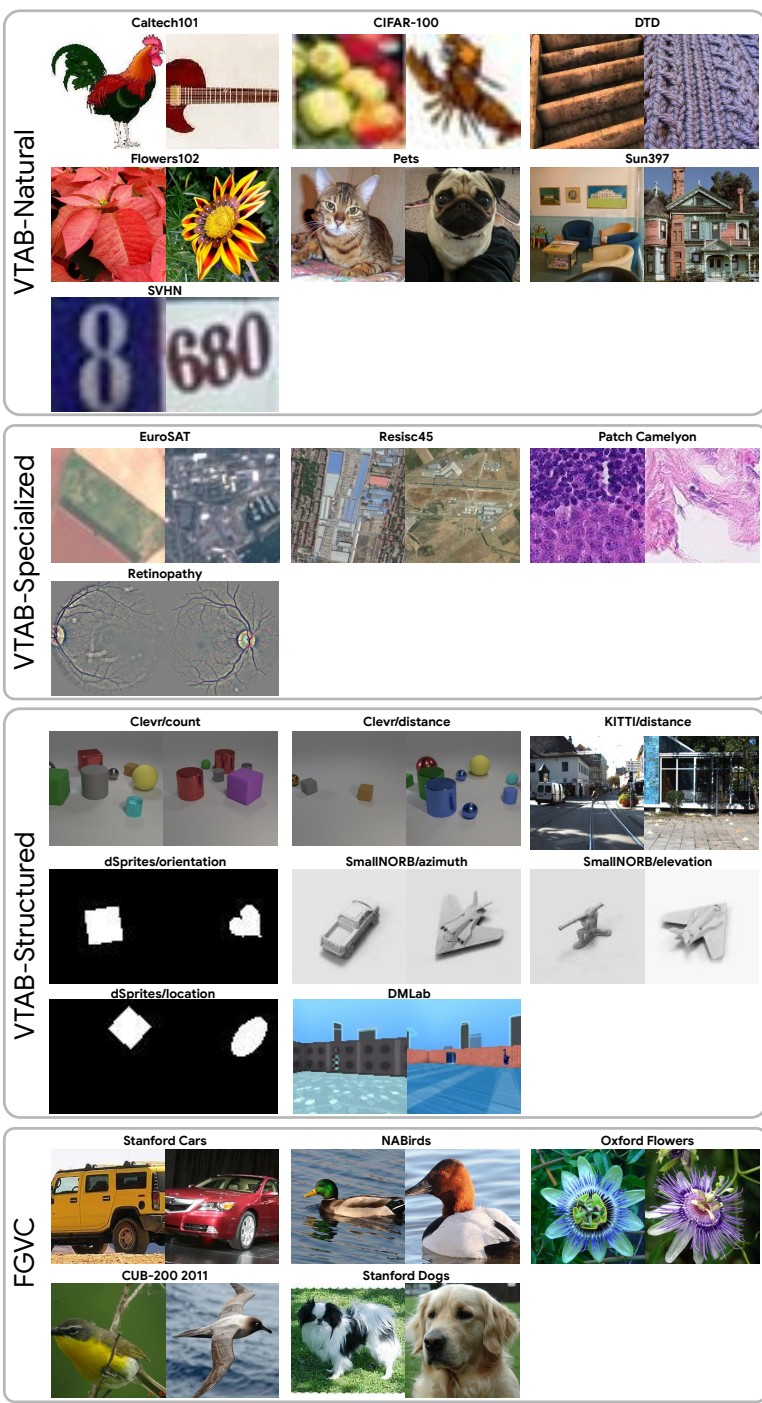

Figure 3: Dataset examples for all classification tasks evaluated

**Dataset Specifications.** Table 4 provides a summary of the statistics and details of the classification datasets evaluated in this paper. Figure 3 presents image examples from all 24 datasets. Following Jia et al. (2022), each FGVC dataset is randomly split into 90% `train` and 10% `val`, with the `val` set

Table 4: Specifications of the datasets used in our experiments, which include 24 datasets from two benchmarks: FGVC and VTAB-1K (Zhai et al., 2019).

| Dataset | Description | # Classes | Train | Validation | Test |
|---|---|---|---|---|---|
| **Fine-grained visual recognition tasks (FGVC)** | | | | | |
| CUB-200-2011 (Wah et al., 2011) | Fine-grained recognition of bird species | 200 | 5,394 | 600 | 5,794 |
| NABirds (Van Horn et al., 2015) | Fine-grained recognition of bird species | 55 | 21,536 | 2,393 | 24,633 |
| Oxford Flowers (Nilsback & Zisserman, 2008) | Fine-grained recognition of flower species | 102 | 1,020 | 1,020 | 6,149 |
| Stanford Dogs (Khosla et al., 2011) | Fine-grained recognition of dog species | 120 | 10,800 | 1,200 | 8,580 |
| Stanford Cars (Gebru et al., 2017) | Fine-grained recognition of car | 196 | 7,329 | 815 | 8,041 |
| **Visual Task Adaptation Benchmark (VTAB-1K)** | | | | | |
| CIFAR-100 (Krizhevsky et al., 2009) | | 100 | | | 10,000 |
| Caltech101 (Fei-Fei et al., 2006) | | 102 | | | 6,084 |
| DTD (Cimpoi et al., 2014) | | 47 | | | 1,880 |
| Flowers102 (Nilsback & Zisserman, 2008) | Natural | 102 | 800/1000 | 200 | 6,149 |
| Pets (Parkhi et al., 2012) | | 37 | | | 3,669 |
| SVHN (Netzer et al., 2011) | | 10 | | | 26,032 |
| Sun397 (Xiao et al., 2010) | | 397 | | | 21,750 |
| Patch Camelyon (Veeling et al., 2018) | | 2 | | | 32,768 |
| EuroSAT (Helber et al., 2019) | Specialized | 10 | 800/1000 | 200 | 5,400 |
| Resisc45 (Cheng et al., 2017) | | 45 | | | 6,300 |
| Retinopathy (Graham, 2015) | | 5 | | | 42,670 |
| Clevr/count (Johnson et al., 2017) | | 8 | | | 15,000 |
| Clevr/distance (Johnson et al., 2017) | | 6 | | | 15,000 |
| DMLab (Beattie et al., 2016) | | 6 | | | 22,735 |
| KITTI/distance (Geiger et al., 2013) | Structured | 4 | 800/1000 | 200 | 711 |
| dSprites/loc (Matthey et al., 2017) | | 16 | | | 73,728 |
| dSprites/ori (Matthey et al., 2017) | | 16 | | | 73,728 |
| SmallNORB/azi (LeCun et al., 2004) | | 18 | | | 12,150 |
| SmallNORB/ele (LeCun et al., 2004) | | 9 | | | 12,150 |

used for hyperparameter tuning. For VTAB-1K, we use an 800-200 split for tuning, and we train on all available data for the final evaluation.

**Pre-trained Backbone Specifications.** This study explores two primary groups of Vision Transformer (ViT) models. The first group comprises conventional ViT architectures (Dosovitskiy, 2020), including ViT-Base, ViT-Large, and ViT-Huge, pre-trained on the large-scale ImageNet-21K (Deng et al., 2009) dataset. The second group focuses on self-supervised models, specifically Masked Autoencoders (MAE) (He et al., 2022) and Momentum Contrast v3 (MoCo v3) (Chen et al., 2021), both pre-trained on ImageNet-1K. Unless otherwise specified, we use ViT-B/16 with supervised pre-training on ImageNet-21K dataset by default.

**Augmentation.** During training, we employ standard image augmentation techniques. For the five FGVC datasets, images are normalized using ImageNet mean and standard deviation, followed by random resized cropping to $224 \times 224$ and random horizontal flipping. For VTAB-1k, images are directly resized to $224 \times 224$.

**Hyperparameters.** We use the `val` set of each dataset to find the optimal prompt length $N_p$, kernel size $K$, and hidden dimension $r$ of the feature projector. Following Jia et al. (2022), the search range for $N_p$ is $\{1, 5, 10, 50, 100, 200\}$. Given the relatively small resolution of our feature map $\boldsymbol{X}_{\mathrm{img}}$ (*e.g.,* $14 \times 14$ for ViT), the kernel size $K$ is selected from $\{2, 3, 4\}$. For the hidden dimension $r$, we explore the range $\{4, 8, 16, 32, 64, 128, 256\}$. To identify the optimal learning rate and weight decay, we perform a grid search, consistent with Mahajan et al. (2018); Jia et al. (2022). The learning rate is searched within $\{50, 25, 10, 5, 2.5, 1, 0.5, 0.25, 0.1, 0.05\}$, while the weight decay is chosen from $\{0.01, 0.001, 0.0001, 0.0\}$. We adopt the batch size settings of Jia et al. (2022), using values of 64 and 128. The models are optimized with SGD for 100 epochs, employing a cosine decay learning rate schedule with 10 warm-up epochs.

**Reproducibility.** Our method, VAPT, is implemented in PyTorch (Paszke et al., 2019). All experimental workflows, including training and evaluation, were conducted on NVIDIA A100-40GB GPUs. The entire implementation will be made openly available to ensure reproducibility and facilitate future research.

Table 5: **VTAB-1K *Natural* per-task results for ViT-B/16 supervised pre-trained on ImageNet-21K.** "Number of Wins" in [·] denotes the number of wins relative to full fine-tuning. "Tuned/Total" represents the percentage of parameters tuned for each task. The highest accuracy among all approaches except Full is highlighted in **bold**.

| ViT-B/16 (Dosovitskiy, 2020) | VTAB-1K (Zhai et al., 2019) *Natural* [7] | | | | | | | Mean |
|---|---|---|---|---|---|---|---|---|
| (85.8M) | CIFAR-100 | Caltech101 | DTD | Flowers102 | Pets | SVHN | Sun397 | |
| Full (Iofinova et al., 2022) | 68.9 | 87.7 | 64.3 | 97.2 | 86.9 | 87.4 | 38.8 | 75.88 |
| Linear (Iofinova et al., 2022) | 63.4 | 85.0 | 63.2 | 97.0 | 86.3 | 36.6 | 51.0 | 68.93 [1] |
| Partial-1 (Yosinski et al., 2014) | 66.8 | 85.9 | 62.5 | 97.3 | 85.5 | 37.6 | 50.6 | 69.44 [2] |
| MLP-2 (Chen et al., 2020) | 63.2 | 84.8 | 60.5 | 97.6 | 85.9 | 34.1 | 47.8 | 67.70 [2] |
| MLP-3 (Chen et al., 2020) | 63.8 | 84.7 | 62.3 | 97.4 | 84.7 | 32.5 | 49.2 | 67.80 [2] |
| MLP-5 (Chen et al., 2020) | 59.3 | 84.4 | 59.9 | 96.1 | 84.4 | 30.9 | 46.8 | 65.98 [1] |
| MLP-9 (Chen et al., 2020) | 53.1 | 80.5 | 53.9 | 95.1 | 82.6 | 24.4 | 43.7 | 61.90 [1] |
| Sidetune (Zhang et al., 2020) | 60.7 | 60.8 | 53.6 | 95.5 | 66.7 | 34.9 | 35.3 | 58.21 [0] |
| Bias (Rebuffi et al., 2017) | 72.8 | 87.0 | 59.2 | 97.5 | 85.3 | 59.9 | 51.4 | 73.30 [3] |
| Adapter-256 (Cai et al., 2020) | 74.1 | 86.1 | 63.2 | 97.7 | 87.0 | 34.6 | 50.8 | 70.50 [4] |
| Adapter-64 (Cai et al., 2020) | 74.2 | 85.8 | 62.7 | 97.6 | 87.2 | 36.3 | 50.9 | 70.65 [4] |
| Adapter-8 (Cai et al., 2020) | 74.2 | 85.7 | 62.7 | 97.8 | 87.2 | 36.4 | 50.7 | 70.67 [4] |
| VPT-Shallow (Jia et al., 2022) | 77.7 | 86.9 | 62.6 | 97.5 | 87.3 | 74.5 | 51.2 | 76.81 [4] |
| - Tuned / Total (%) | 0.18 | 0.10 | 0.04 | 0.27 | 0.08 | 0.19 | 0.36 | 0.17 |
| VPT-Deep (Jia et al., 2022) | 78.8 | 90.8 | 65.8 | 98.0 | 88.3 | 78.1 | 49.6 | 78.48 [6] |
| - Tuned / Total (%) | 0.20 | 0.20 | 0.15 | 0.10 | 0.04 | 0.54 | 0.41 | 0.23 |
| E2VPT (Han et al., 2023) | 78.6 | 89.4 | 67.8 | 98.2 | 88.5 | 85.3 | 52.3 | 80.01 [6] |
| - Tuned / Total (%) | 0.22 | 0.19 | 0.12 | 0.11 | 0.05 | 0.24 | 0.43 | 0.19 |
| VAPT (Ours) | **80.8** ± (0.15) | **91.9** ± (0.46) | **69.7** ± (0.63) | **98.8** ± (0.05) | **89.2** ± (0.05) | **86.7** ± (0.42) | **52.9** ± (0.15) | **81.43** [6] |
| - Tuned / Total (%) | 0.15 | 0.16 | 0.11 | 0.13 | 0.07 | 0.20 | 0.42 | 0.18 |

Table 6: **VTAB-1K *Specialized* per-task results for ViT-B/16 supervised pre-trained on ImageNet-21K.** "Number of Wins" in [·] denotes the number of wins relative to full fine-tuning. "Tuned/Total" represents the percentage of parameters tuned for each task. The highest accuracy among all approaches except Full is highlighted in **bold**.

| ViT-B/16 (Dosovitskiy, 2020) | VTAB-1K (Zhai et al., 2019) *Specialized* [4] | | | | Mean |
|---|---|---|---|---|---|
| (85.8M) | Patch Camelyon | EuroSAT | Resisc45 | Retinopathy | |
| Full (Iofinova et al., 2022) | 79.7 | 95.7 | 84.2 | 73.9 | 83.36 |
| Linear (Iofinova et al., 2022) | 78.5 | 87.5 | 68.6 | 74.0 | 77.16 [1] |
| Partial-1 (Yosinski et al., 2014) | 78.6 | 89.8 | 72.5 | 73.3 | 78.53 [0] |
| MLP-2 (Chen et al., 2020) | 74.3 | 88.8 | 67.1 | 73.2 | 75.86 [0] |
| MLP-3 (Chen et al., 2020) | 77.0 | 88.0 | 70.2 | 56.1 | 72.83 [0] |
| MLP-5 (Chen et al., 2020) | 73.7 | 87.2 | 64.8 | 71.5 | 74.31 [0] |
| MLP-9 (Chen et al., 2020) | 78.5 | 83.0 | 60.2 | 72.3 | 73.49 [0] |
| Sidetune (Zhang et al., 2020) | 58.5 | 87.7 | 65.2 | 61.0 | 68.12 [0] |
| Bias (Rebuffi et al., 2017) | 78.7 | 91.6 | 72.9 | 69.8 | 78.25 [0] |
| Adapter-256 (Cai et al., 2020) | 76.3 | 88.0 | 73.1 | 70.5 | 76.98 [0] |
| Adapter-64 (Cai et al., 2020) | 76.3 | 87.5 | 73.7 | 70.9 | 77.10 [0] |
| Adapter-8 (Cai et al., 2020) | 76.9 | 89.2 | 73.5 | 71.6 | 77.80 [0] |
| VPT-Shallow (Jia et al., 2022) | 78.2 | 92.0 | 75.6 | 72.9 | 79.66 [0] |
| - Tuned / Total (%) | 0.01 | 0.05 | 0.09 | 0.01 | 0.04 |
| VPT-Deep (Jia et al., 2022) | 81.8 | 96.1 | 83.4 | 68.4 | 82.43 [2] |
| - Tuned / Total (%) | 1.06 | 1.07 | 0.15 | 0.02 | 0.57 |
| E2VPT (Han et al., 2023) | 82.5 | **96.8** | 84.8 | 73.6 | 84.43 [3] |
| - Tuned / Total (%) | 0.20 | 0.29 | 0.12 | 0.07 | 0.17 |
| VAPT (Ours) | **84.4** ± (0.72) | 96.5 ± (0.09) | **85.1** ± (0.46) | **74.5** ± (0.32) | **85.13** [4] |
| - Tuned / Total (%) | 0.30 | 0.35 | 0.09 | 0.06 | 0.20 |

# E ADDITIONAL EXPERIMENTS

## E.1 PER-TASK RESULTS FOR VTAB-1K AND FGVC

Table 5, Table 6, Table 7, and Table 8 provide per-task results across 24 classification tasks evaluated in Table 1. All results are averaged of three runs using different initialization seeds, with standard

Table 7: **VTAB-1K *Structured* per-task results for ViT-B/16 supervised pre-trained on ImageNet-21K.** "Number of Wins" in [·] denotes the number of wins relative to full fine-tuning. "Tuned/Total" represents the percentage of parameters tuned for each task. The highest accuracy among all approaches except Full is highlighted in **bold**.

| ViT-B/16 (Dosovitskiy, 2020) (85.8M) | VTAB-1K (Zhai et al., 2019) *Structured* [8] | | | | | | | | |
| --- | --- | --- | --- | --- | --- | --- | --- | --- | --- |
| | Clevr/ count | Clevr/ distance | DMLab | KITTI/ distance | dSprites/ location | dSprites/ orientation | SmallNORB/ azimuth | SmallNORB/ elevation | Mean |
| Full (Iofinova et al., 2022) | 56.3 | 58.6 | 41.7 | 65.5 | 57.5 | 46.7 | 25.7 | 29.1 | 47.64 |
| Linear (Iofinova et al., 2022) | 34.3 | 30.6 | 33.2 | 55.4 | 12.5 | 20.0 | 9.6 | 19.2 | 26.84 [0] |
| Partial-1 (Yosinski et al., 2014) | 41.5 | 34.3 | 33.9 | 61.0 | 31.3 | 32.8 | 16.3 | 22.4 | 34.17 [0] |
| MLP-2 (Chen et al., 2020) | 45.2 | 31.6 | 31.8 | 55.7 | 30.9 | 24.6 | 16.6 | 23.3 | 32.47 [0] |
| MLP-3 (Chen et al., 2020) | 47.8 | 32.8 | 32.3 | 58.1 | 12.9 | 21.2 | 15.2 | 24.8 | 30.62 [0] |
| MLP-5 (Chen et al., 2020) | 50.8 | 32.3 | 31.5 | 56.4 | 7.5 | 20.8 | 14.4 | 20.4 | 29.23 [0] |
| MLP-9 (Chen et al., 2020) | 47.5 | 27.9 | 28.9 | 54.0 | 6.2 | 17.7 | 10.8 | 16.2 | 26.15 [0] |
| Sidetune (Zhang et al., 2020) | 27.6 | 22.6 | 31.3 | 51.7 | 8.2 | 14.4 | 9.8 | 21.8 | 23.41 [0] |
| Bias (Rebuffi et al., 2017) | 61.5 | 55.6 | 32.4 | 55.9 | 66.6 | 40.0 | 15.7 | 25.1 | 44.09 [2] |
| Adapter-256 (Cai et al., 2020) | 45.7 | 37.4 | 31.2 | 53.2 | 30.3 | 25.4 | 13.8 | 22.1 | 32.39 [0] |
| Adapter-64 (Cai et al., 2020) | 42.9 | 39.9 | 30.4 | 54.5 | 31.9 | 25.6 | 13.5 | 21.4 | 32.51 [0] |
| Adapter-8 (Cai et al., 2020) | 45.2 | 41.8 | 31.1 | 56.4 | 30.4 | 24.6 | 13.2 | 22.0 | 33.09 [0] |
| VPT-Shallow (Jia et al., 2022) | 50.5 | 58.6 | 40.5 | 67.1 | 68.7 | 36.1 | 20.2 | 34.1 | 46.98 [4] |
| - Tuned / Total (%) | 0.10 | 0.18 | 0.09 | 0.09 | 0.10 | 0.10 | 0.19 | 0.19 | 0.13 |
| VPT-Deep (Jia et al., 2022) | 68.5 | 60.0 | 46.5 | 72.8 | 73.6 | 47.9 | 32.9 | 37.8 | 54.98 [8] |
| - Tuned / Total (%) | 0.54 | 2.11 | 1.07 | 0.54 | 0.12 | 0.55 | 2.12 | 2.11 | 1.14 |
| E2VPT (Han et al., 2023) | 71.7 | 61.2 | 47.9 | 75.8 | 80.8 | 48.1 | 31.7 | **41.9** | 57.39 [8] |
| - Tuned / Total (%) | 0.34 | 0.65 | 0.44 | 0.36 | 0.10 | 0.38 | 1.14 | 0.66 | 0.51 |
| VAPT (Ours) | **74.8** ± (1.70) | **63.6** ± (0.36) | **50.0** ± (0.74) | **77.2** ± (0.72) | **86.1** ± (0.24) | **48.3** ± (0.89) | **33.8** ± (0.95) | 40.9 ± (2.29) | **59.34** [8] |
| - Tuned / Total (%) | 0.20 | 0.39 | 0.31 | 0.38 | 0.08 | 0.35 | 0.60 | 0.75 | 0.38 |

Table 8: **FGVC per-task results for ViT-B/16 supervised pre-trained on ImageNet-21K.** "Number of Wins" in [·] denotes the number of wins relative to full fine-tuning. "Tuned/Total" represents the percentage of parameters tuned for each task. The highest accuracy among all approaches except Full is highlighted in **bold**.

| ViT-B/16 (Dosovitskiy, 2020) (85.8M) | FGVC [5] | | | | | Mean |
| --- | --- | --- | --- | --- | --- | --- |
| | CUB-200-2011 | NAbirds | Oxford Flowers | Stanford Dogs | Stanford Cars | |
| Full (Iofinova et al., 2022) | 87.3 | 82.7 | 98.8 | 89.4 | 84.5 | 88.54 |
| Linear (Iofinova et al., 2022) | 85.3 | 75.9 | 97.9 | 86.2 | 51.3 | 79.32 [0] |
| Partial-1 (Yosinski et al., 2014) | 85.6 | 77.8 | 98.2 | 85.5 | 66.2 | 82.63 [0] |
| MLP-2 (Chen et al., 2020) | 85.7 | 77.2 | 98.2 | 85.4 | 54.9 | 80.28 [0] |
| MLP-3 (Chen et al., 2020) | 85.1 | 77.3 | 97.9 | 84.9 | 53.8 | 79.80 [0] |
| MLP-5 (Chen et al., 2020) | 84.2 | 76.7 | 97.6 | 84.8 | 50.2 | 78.71 [0] |
| MLP-9 (Chen et al., 2020) | 83.2 | 76.0 | 96.2 | 83.7 | 47.6 | 77.31 [0] |
| Sidetune (Zhang et al., 2020) | 84.7 | 75.8 | 96.9 | 85.8 | 48.6 | 78.35 [0] |
| Bias (Rebuffi et al., 2017) | 88.4 | 84.2 | 98.8 | 91.2 | 79.4 | 88.41 [3] |
| Adapter-256 (Cai et al., 2020) | 87.2 | 84.3 | 98.5 | 89.9 | 68.6 | 85.70 [2] |
| Adapter-64 (Cai et al., 2020) | 87.1 | 84.3 | 98.5 | 89.8 | 68.6 | 85.67 [2] |
| Adapter-8 (Cai et al., 2020) | 87.3 | 84.3 | 98.4 | 88.8 | 68.4 | 85.46 [1] |
| VPT-Shallow (Jia et al., 2022) | 86.7 | 78.8 | 98.4 | 90.7 | 68.7 | 84.62 [1] |
| - Tuned / Total (%) | 0.31 | 0.54 | 0.23 | 0.20 | 0.26 | 0.31 |
| VPT-Deep (Jia et al., 2022) | 88.5 | 84.2 | 99.0 | 90.2 | **83.6** | 89.11 [4] |
| - Tuned / Total (%) | 0.29 | 1.02 | 0.14 | 1.17 | 2.27 | 0.98 |
| E2VPT (Han et al., 2023) | 89.1 | **84.6** | **99.1** | 90.5 | 82.8 | 89.22 [4] |
| - Tuned / Total (%) | 0.32 | 0.65 | 0.15 | 0.88 | 1.27 | 0.65 |
| VAPT (Ours) | **89.7** ± (0.12) | **84.6** ± (0.06) | **99.1** ± (0.04) | **91.7** ± (0.05) | 82.8 ± (0.53) | **89.58** [4] |
| - Tuned / Total (%) | 0.36 | 0.79 | 0.19 | 0.53 | 2.04 | 0.78 |

deviation error bars included. Compared to VPT and other commonly used PEFT methods, VAPT demonstrates consistently superior performance across a variety of downstream tasks while utilizing fewer parameters.

### E.2 PER-TASK RESULTS ON MAE AND MOCO V3

Tables 9, 10, and 11 provide per-task results for MAE (He et al., 2022), as summarized in Table 2. Similarly, Tables 12, 13, and 14 present per-task results for MoCo v3 (Chen et al., 2021), which are also summarized in Table 2.

Table 9: **VTAB-1K *Natural* per-task results for ViT-B/16 pre-trained on MAE (He et al., 2022).** "Number of Wins" in [·] denotes comparisons relative to full fine-tuning. "Tuned/Total" is the percentage of parameters tuned for each task. The highest accuracy among all approaches is highlighted in **bold**.

| ViT-B/16 (Dosovitskiy, 2020) (85.8M) | VTAB-1K (Zhai et al., 2019) *Natural* [7] | | | | | | | Mean |
|---|---|---|---|---|---|---|---|---|
| | CIFAR-100 | Caltech101 | DTD | Flowers102 | Pets | SVHN | Sun397 | |
| Full (Iofinova et al., 2022) | 24.6 | 84.2 | 56.9 | 72.7 | **74.4** | **86.6** | 15.8 | **59.31** |
| VAPT (Ours) | **34.4** | **89.7** | **63.1** | **74.2** | 73.8 | 55.1 | **24.3** | 59.23 [5] |
| - Tuned / Total (%) | 0.17 | 0.16 | 0.10 | 0.13 | 0.07 | 0.12 | 0.40 | 0.16 |

Table 10: **VTAB-1K *Specialized* Per-Task Results for ViT-B/16 Pre-trained on MAE (He et al., 2022).** "Number of Wins" in [·] denotes comparisons relative to full fine-tuning. "Tuned/Total" is the percentage of parameters tuned for each task. The highest accuracy among all approaches is highlighted in **bold**.

| ViT-B/16 (Dosovitskiy, 2020) (85.8M) | VTAB-1K (Zhai et al., 2019) *Specialized* [4] | | | | Mean |
|---|---|---|---|---|---|
| | Patch Camelyon | EuroSAT | Resisc45 | Retinopathy | |
| Full (Iofinova et al., 2022) | **81.8** | **94.0** | 72.3 | 70.6 | 79.68 |
| VAPT (Ours) | 78.9 | 91.6 | **78.7** | **73.7** | **80.73** [2] |
| - Tuned / Total (%) | 0.36 | 0.31 | 0.15 | 0.03 | 0.21 |

Table 11: **VTAB-1K *Structured* Per-Task Results for ViT-B/16 Pre-trained on MAE (He et al., 2022).** "Number of Wins" in [·] denotes comparisons relative to full fine-tuning. "Tuned/Total" is the percentage of parameters tuned for each task. The highest accuracy among all approaches is highlighted in **bold**.

| ViT-Base/16 (Dosovitskiy, 2020) (85.8M) | Clevr/ count | Clevr/ distance | DMLab | KITTI/ distance | dSprites/ location | dSprites/ orientation | SmallNORB/ azimuth | SmallNORB/ elevation | Mean |
|---|---|---|---|---|---|---|---|---|---|
| | VTAB-1K (Zhai et al., 2019) *Structured* [8] | | | | | | | | |
| Full (Iofinova et al., 2022) | **67.0** | **59.8** | **45.2** | **75.3** | 72.5 | **47.5** | **30.2** | 33.0 | **53.82** |
| VAPT (Ours) | 57.9 | 57.1 | 37.3 | 68.2 | **82.6** | 11.5 | 21.6 | **41.7** | 47.24 [2] |
| - Tuned / Total (%) | 0.18 | 0.65 | 0.33 | 0.35 | 0.09 | 0.36 | 0.66 | 0.67 | 0.41 |

Table 12: **VTAB-1K *Natural* Per-Task Results for ViT-B/16 Pre-trained on MoCo v3 (Chen et al., 2021).** "Number of Wins" in [·] denotes comparisons relative to full fine-tuning. "Tuned/Total" is the percentage of parameters tuned for each task. The highest accuracy among all approaches is highlighted in **bold**.

| ViT-B/16 (Dosovitskiy, 2020) (85.8M) | VTAB-1K (Zhai et al., 2019) *Natural* [7] | | | | | | | Mean |
|---|---|---|---|---|---|---|---|---|
| | CIFAR-100 | Caltech101 | DTD | Flowers102 | Pets | SVHN | Sun397 | |
| Full (Iofinova et al., 2022) | 57.6 | 91.0 | 64.6 | 91.6 | 79.9 | **89.8** | 29.1 | 71.95 |
| VAPT (Ours) | **74.5** | **92.0** | **69.5** | **93.2** | **88.2** | 84.6 | **41.8** | **77.69** [6] |
| - Tuned / Total (%) | 0.15 | 0.20 | 0.09 | 0.34 | 0.09 | 0.10 | 0.40 | 0.20 |

### E.3 STATISTICAL SIGNIFICANCE TESTS

To determine whether VAPT consistently surpasses competing PEFT methods across the 19 VTAB tasks, we conducted a one-tailed paired Wilcoxon signed-rank test (Wilcoxon, 1992) for each VAPT-baseline comparison. The null hypothesis $H_0$ for each comparison was that the median difference in performance scores between VAPT and the baseline method is zero. The alternative hypothesis $H_1$ was that VAPT performs better than the baseline method. As reported in Table 15, all resulting $p$-

Table 13: **VTAB-1K *Specialized* Per-Task Results for ViT-B/16 Pre-trained on MoCo v3 (Chen et al., 2021).** "Number of Wins" in [·] denotes comparisons relative to full fine-tuning. "Tuned/Total" is the percentage of parameters tuned for each task. The highest accuracy among all approaches is highlighted in **bold**.

| ViT-B/16 (Dosovitskiy, 2020) (85.8M) | VTAB-1K (Zhai et al., 2019) *Specialized* [4] | | | | Mean |
| --- | --- | --- | --- | --- | --- |
| | Patch Camelyon | EuroSAT | Resisc45 | Retinopathy | |
| Full (Iofinova et al., 2022) | **85.1** | **96.4** | 83.1 | 74.2 | **84.72** |
| VAPT (Ours) | 80.6 | 95.9 | **83.9** | **75.4** | 83.95 [2] |
| - Tuned / Total (%) | 0.36 | 0.34 | 0.11 | 0.03 | 0.21 |

Table 14: **VTAB-1K *Structured* Per-Task Results for ViT-B/16 Pre-trained on MoCo v3 (Chen et al., 2021).** "Number of Wins" in [·] denotes comparisons relative to full fine-tuning. "Tuned/Total" is the percentage of parameters tuned for each task. The highest accuracy among all approaches is highlighted in **bold**.

| ViT-Base/16 (Dosovitskiy, 2020) (85.8M) | VTAB-1K (Zhai et al., 2019) *Structured* [8] | | | | | | | | Mean |
| --- | --- | --- | --- | --- | --- | --- | --- | --- | --- |
| | Clevr/ count | Clevr/ distance | DMLab | KITTI/ distance | dSprites/ location | dSprites/ orientation | SmallNORB/ azimuth | SmallNORB/ elevation | |
| Full (Iofinova et al., 2022) | 55.2 | 56.9 | 44.6 | **77.9** | 63.8 | 49.0 | 31.5 | 36.9 | 51.98 |
| VAPT (Ours) | **74.2** | **65.3** | **48.4** | 73.8 | **88.0** | **51.4** | **32.5** | **52.3** | **60.74** [7] |
| - Tuned / Total (%) | 0.20 | 0.69 | 0.05 | 0.36 | 0.08 | 0.20 | 0.66 | 0.65 | 0.36 |

values were below the 0.05 significance level, indicating that the observed performance improvements of VAPT are statistically significant.

## E.4 DIFFERENT BACKBONE SCALES

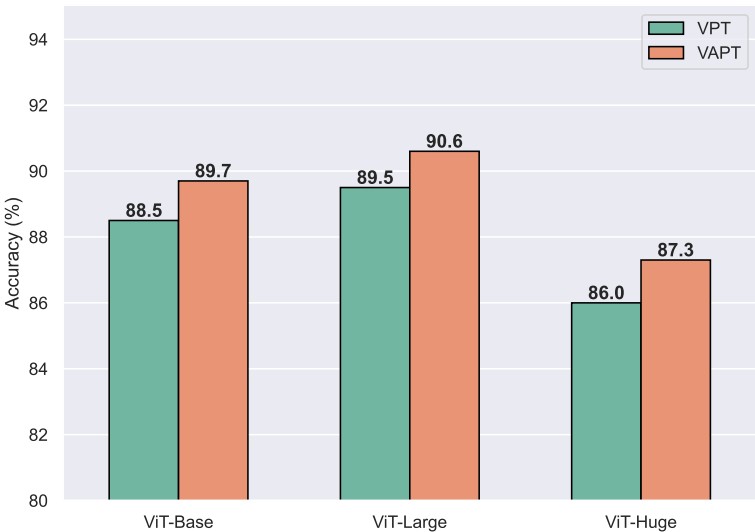

Figure 4: Comparison of VAPT and VPT on CUB-200-2011 across different backbone scales.

In Figure 4, we compare VAPT and VPT on the CUB-200-2011 dataset (Wah et al., 2011) using pre-trained backbones of varying scales (ViT-Base, ViT-Large, and ViT-Huge). The results demonstrate that VAPT consistently surpasses VPT as the model size increases. Notably, with the ViT-Huge backbone, VAPT achieves up to a 1.3% improvement over VPT. These findings highlight the scalability and effectiveness of VAPT compared to VPT as model size grows.

Table 15: **Wilcoxon signed-rank test** evaluating whether VAPT significantly outperforms other methods across 19 VTAB tasks. The results show that VAPT is statistically significantly better than other baselines ($p < 0.05$).

| Method | Full | Bias | Adapter | VPT | E2VPT |
|--------|------|------|---------|-----|-------|
| $p$-value | 5.7e-06 | 1.9e-06 | 1.9e-06 | 1.9e-06 | 0.0001 |

Table 16: **Semantic segmentation results on ADE20K**. All methods are evaluated with SETR (Zheng et al., 2021) using ViT-L. The best mIoU scores among all methods but Full are **bolded**.

| Method | Full | Head | Bias | VPT | VAPT |
|--------|------|------|------|-----|------|
| mIoU | 48.31 | 35.12 | 43.40 | 42.24 | **44.04** |
| # params | 318.31 | 13.18 | 13.46 | 15.60 | 15.29 |

### E.5 SEMANTIC SEGMENTATION

To explore the potential generalizability of VAPT beyond visual classification, we evaluated its performance on the semantic segmentation task. We report mIoU values in Table 16. Our observations indicate that VAPT attains a higher mIoU than VPT while introducing fewer additional parameters. Furthermore, VAPT remains competitive with other PEFT approaches. These findings highlight VAPT's strong generalizability across various computer vision tasks.

### E.6 ABLATION STUDY

Table 17: **Impact of Different Components in VAPT.** Experiments were conducted on the VTAB-1K benchmark. We used ViT-B/16 Supervised Pre-trained on ImageNet-21K as the backbone. "Tuned/Total" is the percentage of parameters tuned for each task. **Bold** highlights the best results.

| | Components | | Tuned | VTAB-1K (Zhai et al., 2019) | | | Mean Total |
|--|--|--|--|--|--|--|--|
| Channel-wise | Feature projector | Sharing Projector | / Total(%) | *Natural* | *Specialized* | *Structured* | |
| ✓ | ✓ | ✓ | 0.27 | **81.43** | **85.13** | **59.34** | **72.91** |
| | ✓ | ✓ | 0.34 | 80.47 | 84.63 | 58.13 | 71.94 |
| ✓ | | | 0.26 | 79.43 | 83.60 | 57.72 | 71.17 |
| ✓ | ✓ | | 0.42 | 80.85 | 85.10 | 59.04 | 72.56 |

**Impact of Different Components.** We conducted ablation studies on the VTAB-1K benchmark (Zhai et al., 2019) to evaluate the individual contributions of each component in VAPT. The results, presented in Table 17, show that incorporating the channel-wise convolution layer increases the overall average performance by 0.97%. This improvement highlights the benefit of explicitly encoding spatial relationships in the feature map before it is processed by the token-wise projectors. Furthermore, the channel-wise convolution layer reduces the overall parameter count by downsampling the feature map from $H \times W$ to $H' \times W'$, where $H' = H - K + 1$ and $W' = W - K + 1$. Consequently, the dimensionality for each token-wise projector is also reduced, leading to fewer parameters. When the feature projector is removed, performance decreases; for instance, in VTAB-1K *Natural*, the performance drops from 81.43% to 79.43%. This decrease highlights the importance of the feature projector. Furthermore, our sharing mechanism for the feature projector not only reduces the number of parameters but also facilitates knowledge transfer between layers, leading to improved performance. Overall, combining all components yielded the best performance, with an average accuracy of 72.91% on VTAB-1K.

**Detailed Analysis of the Channel-wise Convolution Layer.** We evaluated the effectiveness of our channel-wise convolution layer by comparing its performance to that of a standard convolution layer. We also explored alternative strategies for modeling spatial relationships, including an average pooling layer (LeCun et al., 1998). Notably, average pooling can be considered a special case of

Table 18: **Detailed Analysis of Channel-wise Convolution.** Experiments were conducted on the VTAB-1K benchmark. We used ViT-B/16 Supervised Pre-trained on ImageNet-21K as the backbone. "Tuned/Total" is the percentage of parameters tuned for each task. **Bold** highlights the best results.

| Method | Tuned / Total(%) | VTAB-1K (Zhai et al., 2019) | | | Mean Total |
| --- | --- | --- | --- | --- | --- |
| | | *Natural* | *Specialized* | *Structured* | |
| Channel-wise Convolution | 0.27 | **81.43** | **85.13** | **59.34** | **72.91** |
| Standard Convolution | 0.33 | 80.79 | 84.81 | 58.37 | 72.20 |
| Average Pooling | 0.27 | 81.30 | 85.02 | 58.41 | 72.45 |

our channel-wise convolution layer, where the kernel weights are fixed rather than learned. The comparative results are presented in Table 18. Our channel-wise convolution layer not only reduces the number of parameters compared to a standard convolution layer but also mitigates overfitting, thereby improving overall performance. Furthermore, it surpasses the average pooling layer; for instance, on VTAB-1K *Structured*, we achieve a 0.93% performance gain. This improvement is attributed to the greater flexibility of our channel-wise convolution layer compared to that of average pooling.

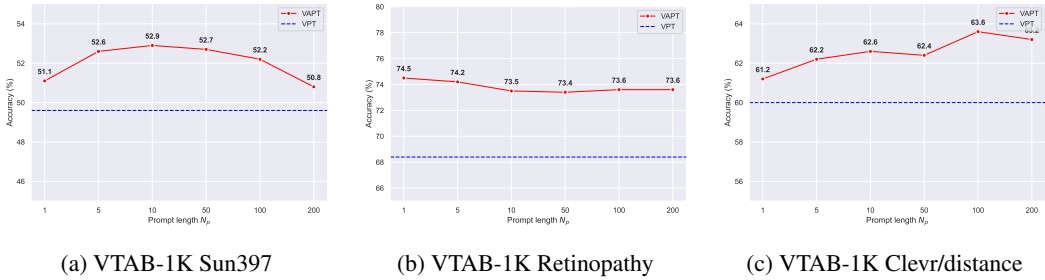

(a) VTAB-1K Sun397          (b) VTAB-1K Retinopathy          (c) VTAB-1K Clevr/distance

Figure 5: Ablation on prompt length $N_p$. Results are reported on 3 datasets, each corresponding to a distinct VTAB subgroup. The dashed line indicates the best results of VPT.

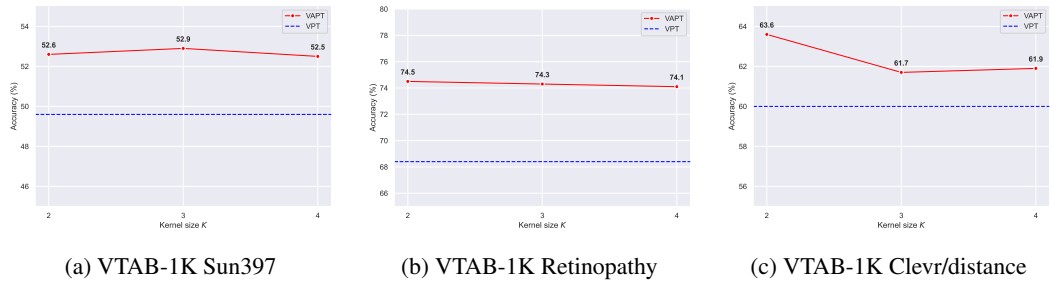

(a) VTAB-1K Sun397          (b) VTAB-1K Retinopathy          (c) VTAB-1K Clevr/distance

Figure 6: Ablation on kernel size $K$. Results are reported on 3 datasets, each corresponding to a distinct VTAB subgroup. The dashed line indicates the best results of VPT.

**Robustness to Different Hyperparameters.** We systematically evaluated the influence of key hyperparameters, including prompt length $N_p$, kernel size $K$, and hidden dimension $r$, by conducting experiments on three representative VTAB-1K tasks: Sun397 (*Natural*), Retinopathy (*Specialized*), and Clevr/distance (*Structured*). The results are shown in Figures 5, 6, and 7. As depicted, the optimal hyperparameters vary across tasks. Nevertheless, VAPT consistently achieves higher performance than VPT across a range of these hyperparameters. Notably, even with only a single prompt, VAPT maintains its advantage over VPT.

**Linear Activation in the Feature Projector.** As shown in Appendix B, VAPT preserves its optimal sample efficiency even when the activation function $\sigma$ in the feature projector (see Equation (11)) is

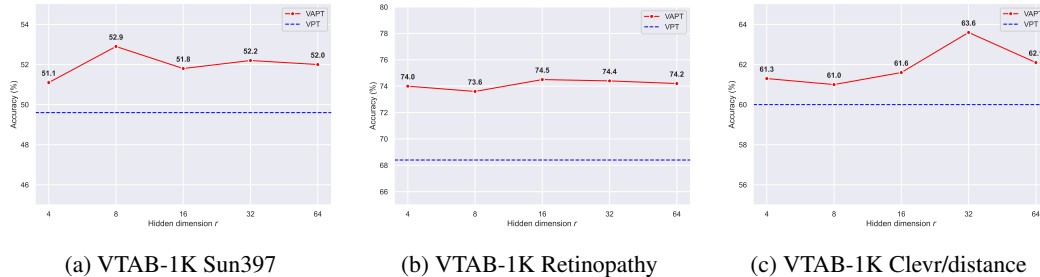

| (a) VTAB-1K Sun397 | (b) VTAB-1K Retinopathy | (c) VTAB-1K Clevr/distance |

Figure 7: Ablation on hidden dimension $r$. Results are reported on 3 datasets, each corresponding to a distinct VTAB subgroup. The dashed line indicates the best results of VPT.

Table 19: **Linear activation in the feature projector.** Experiments were conducted on the VTAB-1K benchmark. We used ViT-B/16 Supervised Pre-trained on ImageNet-21K as the backbone. "Tuned/Total" is the percentage of parameters tuned for each task. **Bold** highlights the best results.

| Method | Tuned / Total(%) | VTAB-1K (Zhai et al., 2019) | | | Mean Total |
| | | *Natural* | *Specialized* | *Structured* | |
|---|---|---|---|---|---|
| VPT | 0.69 | 78.48 | 82.43 | 54.98 | 69.43 |
| VAPT$_{\text{linear}}$ | 0.27 | 80.89 | 84.93 | 59.28 | 72.64 |
| VAPT$_{\text{non-linear}}$ | 0.27 | **81.43** | **85.13** | **59.34** | **72.91** |

replaced by a linear identity function. To substantiate this theoretical result, we report the corresponding performance in Table 19, where $\sigma$ is removed. The results demonstrate that the linear version of VAPT remains competitive with the non-linear variant. Notably, it still outperforms VPT by a substantial margin (e.g., 4.30% on the VTAB-1K *Structured* task), confirming both the theoretical and empirical robustness of our approach.

### E.7 COMPUTATIONAL COST

Table 20: **Comparison of FLOPs and MACs for VAPT and VPT.** The experiments were conducted on FGVC benchmark. We used ViT-B/16 Supervised Pre-trained on ImageNet-21K as the backbone.

| Metric | Method | Stanford Cars | CUB-200-2011 | Oxford Flowers | NABirds | Stanford Dogs |
|---|---|---|---|---|---|---|
| FLOPs (GFLOPS) | VAPT | 73.67 (↑ 0.18%) | 37.04 (↑ 0.11%) | 36.10 (↑ 0.08%) | 44.61 (↑ 0.31%) | 54.30 (↑ 0.59%) |
| | VPT | 73.54 | 37.00 | 36.07 | 44.47 | 53.98 |
| MACs (GMACs) | VAPT | 36.80 (↑ 0.16%) | 18.51 (↑ 0.11%) | 18.03 (↑ 0.06%) | 22.29 (↑ 0.32%) | 27.13 (↑ 0.59%) |
| | VPT | 36.74 | 18.49 | 18.02 | 22.22 | 26.97 |

One primary concern when designing prompts that adapt to the input is the associated computational overhead. Although VAPT outperforms VPT, one notable advantage of VPT is its simplicity, as its prompts remain fixed regardless of the input. To examine these trade-offs, we compare both the performance and computational costs of the two methods. Table 20 reports these costs, measured in FLOPs (GFLOPs) and MACs (GMACs) for VAPT and VPT across five FGVC datasets. VAPT incurs only a slight increase in computational cost, with FLOPs increasing by 0.18% for Stanford Cars and ranging between 0.06% and 0.59% across other datasets. A similar pattern is observed for MACs. These modest increases are outweighed by the significant performance gains. Notably, VAPT also employs fewer parameters than VPT. Overall, these findings highlight the efficiency of VAPT, delivering robust performance improvements at a minimal additional computational cost.

### E.8 ADVERSARIAL ROBUSTNESS

To evaluate the adversarial robustness of VAPT, we apply the Projected Gradient Descent (PGD) attack (Madry et al., 2017) to VTAB-1K CIFAR-100 test samples. As illustrated in Figure 8, model

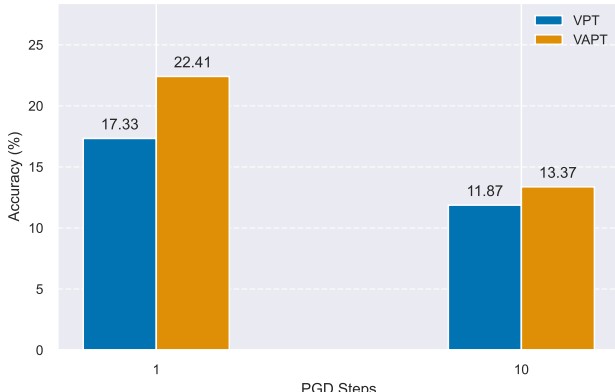

Figure 8: Comparison of VAPT and VPT performance on VTAB-1K CIFAR-100 under PGD adversarial attack.

Table 21: **Image-Caption Retrieval results on ImageNet.** We compare VAPT against VPT using a CLIP ViT-B/32 backbone in a 16-shot setting. The models are evaluated using Recall@100 on the validation set. **Bold** highlights the best results.

| Method | Recall@100 (%) | | |
|---|---|---|---|
| | Image $\rightarrow$ Text | Text $\rightarrow$ Image | Average |
| VPT | 79.05 | 74.80 | 76.93 |
| **VAPT (Ours)** | **82.66** | **77.72** | **80.19** |
| *Improvement* | *+3.61* | *+2.92* | *+3.26* |

performance degrades significantly under attack, which is expected given the limited training data and the absence of adversarial-specific training methods (*e.g.,* adversarial fine-tuning). Despite this, VAPT consistently outperforms VPT under adversarial conditions, suggesting that it may confer some robustness benefits. However, these findings are preliminary, and further experiments are required to substantiate the observed trends.

### E.9    INTERPRETIVE VISUALIZATIONS

To facilitate a deeper understanding of our method, we provide visualizations to illustrate the advantages of VAPT. Specifically, we use GradCAM (Selvaraju et al., 2017) to generate attention maps by computing the gradients of a target concept with respect to the model's final layer. Figure 9 presents examples from five VTAB-1K datasets, namely Sun397, SVHN, Resisc45, Clevr/count, and KITTI/distance, comparing heatmaps generated by VAPT and VPT. These visualizations enable us to examine the regions of the input data to which each technique directs the model's focus. We observe that VAPT can localize relevant image regions more accurately than VPT. For instance, in the Sun397 dataset, while VPT struggles to capture the complete structure of an object, VAPT succeeds in identifying and highlighting its key features. This enhanced localization indicates VAPT's stronger ability to capture salient visual patterns. Consequently, VAPT not only improves performance over VPT but also enhances the interpretability of the model by providing more coherent and precise visual explanations.

### E.10    MULTI-MODAL EXPERIMENTS

To investigate the generalizability of Visual Adaptive Prompt Tuning beyond uni-modal visual recognition, we extend our evaluation to a multi-modal setting, specifically image-caption retrieval. This experiment assesses whether the enhanced functional expressiveness of adaptive prompt experts can facilitate better alignment between visual representations and textual semantics within a contrastive learning framework.

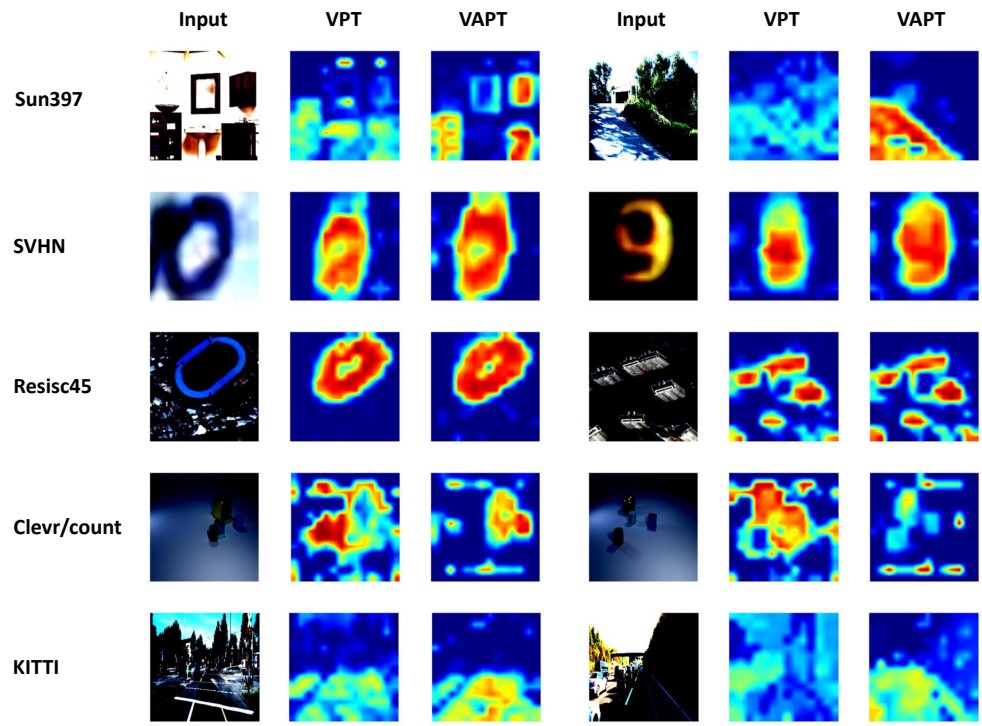

Figure 9: GradCAM visualization of VPT and VAPT on five VTAB-1K datasets. Red regions indicate areas of higher class activation. From left to right: the input image after standard data augmentation, the GradCAM output from VPT, and the GradCAM output from VAPT.

**Experimental Setup.** We perform image-text retrieval on the ImageNet dataset, enriched with rich natural language descriptions as proposed by Fang et al. (2022). We employ the pre-trained CLIP ViT-B/32 model (Radford et al., 2021) as the backbone architecture. Consistent with our main experiments, VAPT is applied exclusively to the visual encoder, while the text encoder remains frozen. We adopt a few-shot learning protocol, utilizing 16 samples per class, which amounts to 16,000 total training samples across the 1,000 ImageNet classes. Both the baseline VPT and our VAPT method utilize a prompt length of $N_p = 10$, resulting in a marginal parameter increase of only 0.19% relative to the backbone. The models are optimized using SGD with a learning rate of 0.01 and a global batch size of 64. The training duration is set to 2 epochs, utilizing the first epoch for warmup, and the objective function is the standard symmetric cross-entropy loss used in CLIP pre-training. Evaluation is conducted on 10,000 samples from the ImageNet validation set, reporting Recall@100 (R@100) for both Image-to-Text (I→T) and Text-to-Image (T→I) retrieval directions.

**Results.** The empirical results, summarized in Table 21, demonstrate that VAPT significantly outperforms VPT in the few-shot cross-modal retrieval task. VAPT achieves an average R@100 of **80.19%**, surpassing the VPT baseline by **3.26%**. This performance gain is consistent across both retrieval directions, with improvements of **3.61%** for Image-to-Text and **2.92%** for Text-to-Image retrieval. These findings suggest that the input-conditional nature of VAPT prompt experts allows the visual encoder to dynamically emphasize features that correlate more effectively with natural language descriptions. Consequently, VAPT achieves superior alignment in the multi-modal embedding space compared to static prompting strategies, without necessitating extensive parameter updates or modification of the text encoder.

# F   USE OF LARGE LANGUAGE MODELS

Large language models were employed solely for editorial purposes, including grammar correction and spelling refinement. They were not used for content generation, data analysis, or the design of experiments.

