# OpenReview forum: "Revisit Visual Prompt Tuning: The Expressiveness of Prompt Experts"
_ICLR.cc/2026/Conference — ICLR 2026 Poster_

### Official Review · Reviewer_sjhP · 2025-10-17

**Soundness:** 3
**Presentation:** 3
**Contribution:** 1
**Rating:** 4
**Confidence:** 4

**Summary:**

The paper reinterprets VPT as the introduction of new prompt experts into these MoE structures, solving the current limitation in existing VPT frameworks: the restricted functional expressiveness of prompt experts.

**Strengths:**

1. The experiment settings are sound with sufficient numbers of ablation studies in the Appendix.
2. The paper is easy to follow, and the motivation for introducing MoE to prompt tuning is reasonable.

**Weaknesses:**

1. The baselines provided in this paper are not new. More recent prompt tuning and other PEFT methods [1-4] should be included for completeness.

2. A critical problem of this paper is its novelty; [5] has proposed MoE prompt tuning as a manifold mapper, indicating that MoE design on prompt tuning can bring stronger expressivity. This work is highly related to the proposed research, although it has not been discussed.

3. Inconsistent experimental report. Table 1 includes E2VPT, while Table 2 does not. Similar for GateVPT.

4. The claim on the low-data regime in the introduction does not have further showcases. Also, the noticeable performance gap might be an outline for VPT. The reason is, as shown in [6], VPT generally brings good few-shot performance, when the training repeats for several times (to avoid bad samples for the training).

[1] Visual Fourier Prompt Tuning

[2] Visual instance-aware prompt tuning

[3] Apla: A simple adaptation method for vision transformers

[4] DS2VP: Dynamically-Selected Spatially Visual Prompting

[5] MEPT: Mixture of Expert Prompt Tuning as a Manifold Mapper

[6] Facing the Elephant in the Room: Visual Prompt Tuning or Full Finetuning?

**Questions:**

The major concern is the novelty and some claims in this paper. The most relevant paper on MoE prompt tuning is not discussed in this paper. Also, the paper sounds like the direct integration of MoE. Even without a similar approach, the novelty itself is questionable. The authors do not clearly separate their method from traditional MoE attempts.

Another problem is that some claims might be misleading, though the authors formulate the MoE and the proposed method's training objective; the core idea is intuitive and simple. I think some equations are unnecessary and further complex the understanding of the basic concept of this paper.

---

> ### Author Response · Authors · 2025-11-20
> **Response to Reviewer sjhP (Part 1)**
>
> We thank the reviewer for the constructive feedback and for recognizing the soundness of our experimental settings and the clarity of our motivation. We have carefully addressed the concerns regarding baselines, novelty, and theoretical presentation below.
>
>
> **Q1: Comparison with Recent Baselines**
>
> Thank you for highlighting this important line of work. In response, we have incorporated the relevant comparative results into the experimental section of the manuscript (highlighted in blue). Our analysis shows that the proposed method, VAPT, achieves **competitive performance relative to state-of-the-art methods** across multiple benchmarks. Notably, VAPT attains this performance while **using approximately 50% fewer trainable parameters** (0.36% compared to 0.66%), underscoring the **efficiency** of our method.
>
> Furthermore, when using an MAE pre-trained ViT, the aggregated results on the VTAB-1K benchmark, covering 19 diverse datasets, indicate that VAPT surpasses existing methods on several subsets. The table below summarizes these findings and highlights the **effectiveness** of our proposed approach.
>
> |Method|VTAB-Natural|VTAB-Specialized| VTAB-Structured|
> |-|:-:|:-:|:-:|
> |VPT-Deep|36.02|60.61|26.57|
> |ViaPT|54.26|78.01|37.52|
> |VFPT|53.59|77.75|36.15|
> |**VAPT**|**59.23**|**80.73**|**47.24**|
>
>
> It is important to emphasize that **the primary objective of this research is not necessarily to establish a new state-of-the-art across all datasets**. Rather, as outlined in the abstract and introduction, our goal centers on scholarly value rather than solely methodological advancement, specifically, **to provide the research community with insights into the benefits of enhancing the functional expressiveness of prompt experts within visual prompt tuning frameworks**. VAPT is presented as a concrete design that fulfills this objective. Its adaptive formulation demonstrates **the potential of adaptive prompt experts not only to improve performance but also to increase model efficiency**.
>
> A key distinction of VAPT lies in its functional formulation. While **prior work has often employed very complex functions** to map inputs to prompts, the complexity of these mappings typically precludes formal theoretical analysis. In contrast, **VAPT’s straightforward formulation** is a significant advantage, as it **enables a rigorous theoretical examination of its effectiveness, an aspect largely absent from previous research in this domain.**
>
> Accordingly, the most relevant comparison is between VAPT’s adaptive prompt-expert formulation and VPT, where prompt experts are constant functions. Our results consistently show that **VAPT outperforms the baseline VPT on the majority of benchmarks, while requiring a more compact set of trainable parameters**. Moreover, our empirical findings detail the statistical advantages of VAPT over VPT, a conclusion rigorously supported by our theoretical analysis in Section 5 and Appendix A. We are confident that the revised experimental section successfully fulfills its intended purpose.
>
>
> **Q2: Discussion of Relevant Work on MoE Prompt Tuning**
>
> Thank you for the feedback. We have incorporated a comparison with [5] in Appendix C (highlighted in blue). In [5], the authors **introduce explicit additional MoE components** into the Transformer architecture, specifically, new routing modules that sparsely activate or select prompts based on the input.
>
> In contrast, we emphasize (as analyzed in Section 3) that **an MoE structure is already implicitly present in existing visual prompt tuning frameworks**: each prompt can be interpreted as containing parameters corresponding to multiple prompt experts. Therefore, our method does not introduce any new MoE modules or routing components, unlike [5]. Instead, our work aims to investigate the benefits of improving the formulation of prompt experts **within this already existing implicit MoE structure**. In other words, our approach enhances the expressiveness of the built-in prompt experts without adding any architectural components to the backbone. Additionally, the straightforward formulation of VAPT is a noteworthy advantage, as it enables **a formal theoretical analysis of its effectiveness, an aspect that has been largely absent in prior work in this area.**

---

> > ### Author Response · Authors · 2025-11-20
> > **Response to Reviewer sjhP (Part 2)**
> >
> > **Q3: Comparison with E2VPT and GateVPT**
> >
> > Thank you for your question. GateVPT [a] is specifically designed for enhancing VPT with **self-supervised vision transformers**. For this reason, we include it in Table 2 only when comparing methods that rely on pre-trained weights from MAE or MoCo v3. This choice aligns with the comparative methodology adopted in prior work [b], ensuring a consistent and fair evaluation.
> >
> > Following the reviewer’s suggestion, we have added E2VPT to Table 2 (highlighted in blue). Below, we summarize the results on VTAB-1K using a ViT-B/16 backbone with MAE pre-training. As shown, VAPT achieves performance comparable to E2VPT on VTAB-Natural, while outperforming it on both VTAB-Specialized and VTAB-Structured. VAPT also maintains competitive performance when using MoCo v3 pre-trained weights.
> >
> > |Method|VTAB-Natural|VTAB-Specialized| VTAB-Structured|
> > |-|-|-|-|
> > |E2VPT|**59.52**|77.80|44.65|
> > |VAPT|59.23|**80.73**|**47.24**|
> >
> >
> > Although these results are encouraging, we would like to reiterate that the primary objective of this research is not to establish a new state-of-the-art across all datasets. Rather, as stated in the abstract and introduction, our goal is to provide conceptual and empirical insights into the benefits of increasing the functional expressiveness of prompt experts within visual prompt tuning frameworks, thereby contributing to the scholarly understanding of this design space.
> >
> > [a] Improving Visual Prompt Tuning for Self-supervised Vision Transformers. ICML 2023
> >
> > [b] Visual Fourier Prompt Tuning. NeurIPS 2024
> >
> >
> > **Q4: Low-Data Regime**
> >
> > We thank the reviewer for the insightful comments. We agree that the performance gap between VAPT and VPT in the low-data regime is a key aspect of our findings. We clarify our contributions as follows:
> >
> > Our work does not assert that VPT is an ineffective method in a low-data regime. Rather, we aim to demonstrate that VAPT's use of more expressive prompt experts provides a significant advantage over VPT's constant prompt expert formulation, particularly in scenarios of high data scarcity. Our results across various data fractions (1%, 10%, 30%, 50%, and 100%) confirm VAPT's superior performance, with the gap being most substantial at the 1% level.
> >
> > While [6] demonstrates VPT's effectiveness in few-shot learning, their experiments were conducted with significantly larger data fractions. Their experiments with varying fractions of datasets were primarily conducted on VTAB-1K, with the smallest number of samples for an experiment being 400, which constitutes at least 40% of the dataset. **Our findings, therefore, do not contradict but rather complement this research by providing a specific analysis in a more extreme low-data regime.**
> >
> > Crucially, **this empirical advantage is supported by our theoretical analysis**. As detailed in Section 5 and Appendix A, our theoretical analysis demonstrates that the VAPT formulation attains optimal sample efficiency, **providing a robust explanation** for its strong performance when training data is limited.
> >
> >
> > **Q5: The paper sounds like the direct integration of MoE. The authors do not clearly separate their method from traditional MoE attempts.**
> >
> > Thank you for your comment. We would like to clarify that **our work does not introduce any new MoE components to the base model**. As analyzed in Section 3, **an implicit MoE structure is already extant in standard visual prompt tuning frameworks**. In this view, each prompt inherently contains parameters that can be interpreted as multiple prompt experts. VAPT is designed to operate within this existing implicit structure, obviating the need for any new architectural modules or routing components. Our contribution lies in **enhancing the expressiveness of these inherent prompt experts**.
> >
> > Conventional MoE-based methods, such as the one proposed in, directly integrate new explicit MoE components into the Transformer architecture. This typically involves introducing new routing modules that sparsely select prompts or experts based on the input, thereby creating a new MoE-infused architecture.
> >
> > In contrast, our approach not only **avoids significant modifications to the backbone architecture but also yields a formulation that is amenable to formal theoretical analysis**. The ability to theoretically prove the effectiveness of VAPT, a characteristic largely absent from prior work in this domain, is a noteworthy advantage of our methodology.

---

> > > ### Author Response · Authors · 2025-11-20
> > > **Response to Reviewer sjhP (Part 3)**
> > >
> > > **Q6: The authors formulate the MoE and the proposed method's training objective.**
> > >
> > >
> > > Thank you for your comment. We would like to clarify that the primary contribution of our work is the enhancement of the expressiveness of visual prompt experts within the existing VPT framework. As such, **we do not introduce a novel training objective**.
> > >
> > > The training of our model adheres to the original VPT methodology, employing only a cross-entropy loss function. It is important to note that any loss functions discussed in Section 5 or Appendix A serve as a statistical tool for the theoretical analysis of prompt expert behavior. These are not empirical losses utilized during the experimental phase. This analytical methodology is consistent with prior research that has adopted a similar framework [c].
> > >
> > > [c] Mixture of Experts Meets Prompt-Based Continual Learning, NeurIPS 2024
> > >
> > >
> > > **Q7: Complexity of Equations**
> > >
> > > Thank you for your comment. While the underlying concept of VAPT is indeed intuitive, a primary objective of our paper is to **extend beyond intuition by providing a rigorous mathematical foundation for adaptive prompting**. We believe that the paper is clear and easy to follow (Reviewer YSjb, Reviewer DQQA). Each equation included is essential to the formal development of prompt experts within the VAPT framework and plays a important role in demonstrating the optimal sample efficiency of VAPT, a key contribution that differentiates our work in the field.
> > >
> > >
> > > **Q8: The Novelty of Our Work**
> > >
> > > We appreciate the opportunity to clarify the novelty and contributions of our study. Our work delivers distinct and substantive advances in three key aspects:
> > >
> > > - **Foundational Motivation**: As noted by Reviewer DQQA, our work stems from a strong motivation: We connect VPT to MoE, identifying a key limitation  in the expressiveness of prompt experts. While pre-trained experts $f_j(X) = {W_m^V}^\top E_jX$ adapt to the input, standard prompt experts in VPT $f_{N + j'}(X) = {W^V_m}^\top p_{j'}$ function as static, constant experts. This discrepancy naturally raises the question of whether more expressive, input-dependent prompt experts could offer substantial benefits. **Uniquely, we leverage the existing MoE structure within attention heads rather than artificially imposing an MoE architecture as seen in prior work**. This perspective allows us to design VAPT as **a straightforward implementation while using MoE theory to rigorously analyze its behavior**.
> > >
> > > - **Efficiency without Compromise**: **A common assumption is that increasing functional expressiveness inevitably reduces efficiency**. Our design challenges this assumption by demonstrating that both objectives can be achieved simultaneously. VAPT effectively enhances the expressiveness of prompt experts by introducing adaptive prompt experts while maintaining parameter efficiency (as highlighted by Reviewer YSjb). Moreover, our results demonstrate that the efficiency gains extend beyond parameter count: VAPT also exhibits strong data efficiency, **supported by both experimental evidence and theoretical justification**. This dual improvement emphasizes that expressiveness and efficiency need not be mutually exclusive.
> > >
> > > - **Novel Theoretical Insights and Analytical Tractability**: A major contribution of our work is the demonstration that VAPT achieves an optimal sample efficiency rate. **Prior works often rely on complex designs utilizing heuristic strategies** to vary prompts across inputs. While potentially effective, **the functional complexity of these architectures frequently precludes meaningful theoretical analysis**. In contrast, **we deliberately designed VAPT to maintain a simple yet expressive functional form**. This facilitates the clean expert formulations in Equations (13) and (14), which **enable thorough theoretical analysis without compromising empirical effectiveness, an aspect largely absent in prior literature. The resulting balance between empirical performance and theoretical guarantees addresses a significant gap in the field.** VAPT not only performs well in practice but also offers provable robustness and generalization behavior, adding substantive scholarly value.

---

> > > > ### Comment · Reviewer_sjhP · 2025-11-20
> > > >
> > > > I appreciate the authors' constructive feedback. I thus have no further questions and would like to raise my rating accordingly.
> > > >
> > > > Please ensure that you incorporate the discussed content into the revised paper.

---

> > > > > ### Author Response · Authors · 2025-11-20
> > > > > **Thank You**
> > > > >
> > > > > Dear Reviewer sjhP,
> > > > >
> > > > > We're glad our rebuttal addresses most of your concerns and appreciate that you increase your rating to 6.
> > > > >
> > > > > We will incorporate your suggestions into the revision of our paper as discussed. Please feel free to let us know if you have any further concerns.
> > > > >
> > > > > Best regards,
> > > > >
> > > > > Authors

---

### Official Review · Reviewer_YSjb · 2025-10-31

**Soundness:** 3
**Presentation:** 3
**Contribution:** 2
**Rating:** 6
**Confidence:** 4

**Summary:**

This paper revisits Visual Prompt Tuning through the lens of Mixture of Experts and identifies a fundamental limitation: conventional prompt tokens are static and input-invariant, thus lacking expressive power. To address this, the authors propose Visual Adaptive Prompt Tuning, which generates input-dependent adaptive prompt experts using lightweight token-wise projectors, channel-wise convolutions, and a shared feature projector. This design enhances the functional expressiveness of prompts while maintaining parameter efficiency. Theoretically, VAPT achieves optimal sample efficiency under the MoE framework, and empirically, it outperforms both fully fine-tuned and VPT baselines across VTAB-1K and FGVC benchmarks.

**Strengths:**

1. The paper offers a clear theoretical reinterpretation of VPT through the lens of MoE, providing both conceptual insight and mathematical grounding for understanding prompt tuning behavior.

2. The proposed Visual Adaptive Prompt Tuning (VAPT) effectively enhances the expressiveness of prompt experts by introducing input-dependent adaptive prompts while maintaining parameter efficiency.

3. Overall writing is clear and easy to follow,

**Weaknesses:**

1. Conceptually, VAPT’s “input-adaptive prompt experts” is similar to prompt-pool-based approaches [R1. R2]. These methods also condition prompt selection or generation on input features. Especially, [R2] generates tokens based on visual prompts based on the input. If authors could provide comparison between proposed method and existing  prompt-pool-based approaches, it strengthens the novelty of  works.

[R1] Wang, Zifeng, et al. "Learning to prompt for continual learning." CVPR 2022.

[R2] Kim, Youngeun, et al. "Open-world dynamic prompt and continual visual representation learning." ECCV 2024.

2. The proposed approach introduces multiple small components (channel-wise conv, token-wise projector, shared MLP). While lightweight in current ViT-B/16 settings, their scalability to larger backbones (ViT-L/14, ViT-H/14) or higher-resolution inputs is not discussed. The added modules could potentially become computational bottlenecks or require additional tuning.

3. Although the paper argues that VAPT enhances functional expressiveness, there are no visual analyses (e.g., attention maps, learned prompt diversity, or feature attribution) to substantiate this claim. Qualitative results could help illustrate how adaptive prompts differ in behavior from static ones.

4. (optional) All experiments are performed on classification and segmentation benchmarks. It would strengthen the contribution to show that VAPT generalizes to non-classification visual tasks, such as detection or vision-language retrieval, especially since prompt-tuning is often used in multimodal settings.

**Questions:**

Please address questions in Weakness section. Thank you.

---

> ### Author Response · Authors · 2025-11-20
> **Response to Reviewer YSjb (Part 1)**
>
> Thank you for your constructive feedback and insightful comments, as well as for the positive overall assessment (score 6). Below, we provide a point-by-point response and summarize the corresponding revisions that will be incorporated into the final version.
>
>
> **Q1: Comparison between proposed method and existing prompt-pool-based approaches**
>
> Thank you for your valuable suggestion and for highlighting this important line of work. In response, we have expanded our discussion in Appendix C to explicitly include these references (highlighted in blue).
>
> Recent prompt-pool-based approaches [R1, R2] first forward the input image through an encoder to obtain a $\mathrm{[CLS]}$ representation, which we denote as an unprompted feature $q(X)$. This feature is then used to select or generate prompts that are applied uniformly across **all attention layers**. Consequently, in these methods the effective prompt experts are functions of the first-layer input $X^{(0)}$. Moreover, this design **incurs non-trivial computational overhead, as the model must process each input twice during both training and inference**: once to compute $q(X)$ for prompt selection or generation, and a second time to perform the actual forward pass with the chosen prompts.
>
> VAPT differs in two key aspects. **First, at the functional level,** VAPT generates prompts at each layer $l$, explicitly **conditioned on the layer-specific representation $X^{(l)}$, rather than on $X^{(0)}$**. This layer-wise conditioning follows directly from our theoretical framework linking MoE and VPT, which shows that both the experts and their score functions should depend on $X^{(l)}$. Consequently, VAPT treats prompt experts as functions of the current hidden representation, not merely of the raw input. This design also enables VAPT to generate prompt tokens within a **single** forward pass, since prompts are conditioned on the input to the current self-attention layer. As a result, **VAPT introduces only minimal computational overhead compared to standard VPT** (see Appendix E.7).
>
>
> **Second, at the design level**, many prior adaptive prompting methods rely on complex, heuristic image-to-prompt transformations. Although these approaches can be empirically effective, their functional complexity often makes rigorous theoretical analysis difficult. In contrast, **VAPT is deliberately constructed to maintain a simple yet expressive functional form**: its expert and score functions admit clean closed-form formulations (e.g., Equations (13) and (14) in our paper). **This structure supports a thorough theoretical analysis without sacrificing empirical performance, an aspect that is largely missing in prior literature.**
>
>
> **Q2: Scalability to larger backbones**
>
> Thank you for raising this important point. We would like to clarify that we have included an analysis of backbone scaling in Appendix E.4. The results show that, as the model size increases, VAPT consistently outperforms VPT, supporting the scalability and effectiveness of VAPT as backbone capacity grows. Moreover, these empirical observations are aligned with our theoretical guarantees, which further support the robustness of the proposed method.

---

> ### Author Response · Authors · 2025-11-20
> **Response to Reviewer YSjb (Part 2)**
>
> **Q3: Qualitative visual analyses to support functional expressiveness**
>
>
> Thank you for this thoughtful comment. We would like to clarify what we mean by *functional expressiveness* in the context of VAPT. In our setting, pre-trained experts are of the form $f_j(X) = {W_m^V}^\top E_jX$, which explicitly depend on the input features $X$. By contrast, standard prompt experts in VPT are given by $f_{N + j'}(X) = {W^V_m}^\top p_{j'}$, where $p_{j'}$ is a fixed prompt vector. These experts are therefore static and independent of the input. Our VAPT formulation enhances the expressiveness of prompt experts by making them adaptive to the input, as shown in Equation (13) of the main paper: $f_{N + j'}(X) = {W_m^V}^\top P_{j'}(X)$. Here $P_{j'}(X)$ is an explicit function of $X$, so the prompt experts vary across inputs. This analytic distinction between input-independent and input-dependent experts is the primary sense in which we argue that VAPT is more expressive, and it is precisely what enables our rigorous theoretical analysis without compromising empirical performance.
>
>
> Regarding qualitative evidence, we agree that visual analyses can provide additional intuition. To this end, we have already included a visual comparison in Appendix E.9, where we analyze attention maps for VAPT and VPT. We observe that VAPT tends to localize relevant image regions more accurately than VPT. For example, on the SUN397 dataset, VPT sometimes fails to capture the complete structure of the target object, whereas VAPT more reliably identifies and highlights its key components. This improved localization suggests that VAPT captures salient visual patterns more effectively. Consequently, VAPT not only achieves stronger quantitative performance than VPT, but also yields more coherent and precise visual explanations, thereby enhancing interpretability.
>
>
> **Q4: Multimodal Settings**
>
> Thank you for your valuable suggestion. In response, we conduct an image–caption retrieval experiment on ImageNet following [1], using rich natural language descriptions in Appendix E.10 (highlighted in blue). Unlike standard ImageNet classification, which relies on simple template-based labels (e.g., “a photo of a [class]”), we use descriptive captions derived from Flickr metadata. Each caption is formed by concatenating the image title, tags, and description, thereby providing more diverse and informative text supervision.
>
> We evaluate on 10,000 samples from the ImageNet validation set using Recall@100 for both image-to-text (I→T) and text-to-image (T→I) retrieval. The results are summarized below:
>
> | Method | I→T R@100 | T→I R@100 | Avg. R@100 |
> |--------|-----------|-----------|------------|
> | VPT    | 79.05%    | 74.80%    | 76.93%     |
> | *VAPT* | *82.66%* | *77.72%* | *80.19%* |
> | Improvement | *+3.61%* | *+2.92%* | *+3.27%* |
>
>
> These results show that VAPT surpasses VPT in cross-modal retrieval, with consistent gains in both retrieval directions. We attribute this improvement to the input-conditional nature of VAPT’s prompt experts, which enables the visual encoder to dynamically emphasize features that align more closely with natural language descriptions.
>
> [1] Data Determines Distributional Robustness in Contrastive Language Image Pre-training (CLIP). ICML 2022

---

### Official Review · Reviewer_DQQA · 2025-11-01

**Soundness:** 3
**Presentation:** 3
**Contribution:** 2
**Rating:** 4
**Confidence:** 4

**Summary:**

This paper reinterprets Visual Prompt Tuning (VPT) through the lens of Mixture of Experts (MoE) and identifies a key limitation: conventional VPT uses static, input-invariant prompt tokens, which restricts expressiveness. Building on this observation, the authors propose Visual Adaptive Prompt Tuning (VAPT)—a parameter-efficient extension that generates input-dependent prompts. The authors empirically demonstrate consistent gains over VPT and other PEFT methods across VTAB-1K and FGVC benchmarks.

**Strengths:**

- The main motivation of this paper can provide a mathematically grounded analysis

- The paper is easy to follow.

- The authors provide a variety of experiments, with results on FGVC, VTAB-1K, and supervised and self-supervised pretrained backbones, showing the robustness of the proposed method. In addition, ablation studies in the Appendix are very helpful to understand the proposed method.

**Weaknesses:**

- My major concern is the novelty.
For adaptive visual prompt tuning, there are many visual prompt tuning works (e.g., CVPT, CoCoOp, ViaPT, V2APT) already exploring visually adaptive or instance-aware prompts. Hence, the contribution in adaptivity itself is incremental rather than fundamentally new. In addition, MoE Interpretation is also heavily motivated by Le et al, who already framed attention and prompting under MoE theory.


[CVPT] CVPT: Cross-Attention help Visual Prompt Tuning adapt visual task, NeurIPS 2024

[CoCoOp] Conditional Prompt Learning for Vision-Language Models, CVPR 2022

[ViaPT] Visual Instance-aware Prompt Tuning, MM 2025

[V2APT] Visual Variational Autoencoder Prompt Tuning, arXiv 2025


- The paper lacks sufficient comparison with other recent variants of Visual Prompt Tuning. There has been a surge of follow-up works for VPT, yet these are not adequately discussed or compared. Especially, input-dependent or adaptive prompting methods should be carefully compared and discussed.

- There are some fairness issues, where the Tuned/Total ratio differs substantially across methods. For instance, Gated VPT (in Table 2) uses only about 0.05 % of trainable parameters, while VAPT tunes 0.27 % – 0.28 % of the model. Such discrepancy can partly explain the performance gap, making the comparison less fair. A more rigorous evaluation should control for the number of trainable parameters.

**Questions:**

- Can the proposed visual adaptive prompt method be applied on top of recent VPT variants?

---

> ### Author Response · Authors · 2025-11-20
> **Response to Reviewer DQQA (Part 1)**
>
> Thank you for your constructive feedback and insightful comments. Below, we provide a point-by-point response and summarize the corresponding revisions in the final version.
>
>
> **Q1: The Novelty of Our Work**
>
> We appreciate the opportunity to clarify the novelty and contributions of our study. Our work delivers distinct and substantive advances in three key aspects:
>
> - **Foundational Motivation**: Our work is driven by a strong motivation: we connect VPT to MoE and identify a key limitation in the expressiveness of prompt experts. While pre-trained experts $f_j(X) = {W_m^V}^\top E_jX$ adapt to the input, standard prompt experts in VPT $f_{N + j'}(X) = {W^V_m}^\top p_{j'}$ function as static, input-independent experts. This discrepancy naturally raises the question of whether more expressive, input-dependent prompt experts could offer substantial benefits. **Crucially, our approach leverages the existing MoE structure within attention heads rather than imposing an external MoE architecture, as is common in prior work.** This perspective allows us to design VAPT as **a simple and practical implementation while still using MoE theory to rigorously analyze its behavior.**
>
> - **Expressiveness and Efficiency, Without Trade-off**: **A common assumption is that increasing functional expressiveness necessarily comes at the cost of efficiency**. Our design challenges this assumption by demonstrating that both objectives can be achieved simultaneously. VAPT enhances the expressiveness of prompt experts through adaptive, input-dependent prompts while maintaining parameter efficiency (as highlighted by Reviewer YSjb). Moreover, our results show that the benefits extend beyond parameter count: VAPT also exhibits strong data efficiency, **supported by both experimental evidence and theoretical justification**. This dual improvement underscores that expressiveness and efficiency need not be mutually exclusive.
>
> - **Novel Theoretical Insights and Analytical Tractability**: A central contribution of our work is to show that VAPT achieves an optimal sample efficiency rate. **Prior works often rely on more complex architectures and heuristic mechanisms to vary prompts across inputs.**  While these designs can be empirically effective, **their functional complexity often precludes meaningful theoretical analysis**. In contrast, **we deliberately designed VAPT to retain a simple yet expressive functional form**, which leads to the clean expert formulations in Equations (13) and (14). These formulations **enable a thorough theoretical analysis without sacrificing empirical performance, an aspect largely missing in prior literature. The resulting balance between empirical effectiveness and theoretical guarantees addresses a significant gap in the field**: VAPT not only performs well in practice but also offers provable robustness and generalization behavior, thereby adding substantive scholarly value.
>
>
> **Q2: In addition, MoE Interpretation is also heavily motivated by Le et al, who already framed attention and prompting under MoE theory.**
>
> Thank you for this insightful comment. We explicitly acknowledge that the MoE interpretation was originally proposed by Le et al., and we do not claim this conceptual framework as our own contribution. Instead, **we build upon their interpretation to uncover a critical limitation in the expressiveness of prompt experts**. In particular, while pre-trained experts $f_j(X) = {W_m^V}^\top E_jX$ depend on the input features $X$, standard prompt experts in VPT, $f_{N + j'}(X) = {W^V_m}^\top p_{j'}$ act as static, input-independent experts.
>
> Our work is precisely aimed at addressing this discrepancy. We use the MoE perspective as an analytical tool to design and study VAPT, which in turn enables a rigorous theoretical analysis. This framework allows us to **demonstrate the statistical advantages of employing more expressive, input-dependent experts, an aspect that, to the best of our knowledge, has been largely missing from prior work.**

---

> > ### Author Response · Authors · 2025-11-20
> > **Response to Reviewer DQQA (Part 2)**
> >
> > **Q3: Comparison with other recent variants of Visual Prompt Tuning**
> >
> > Thank you for your valuable suggestion and for pointing out this important line of work. Accordingly, we have expanded our discussion in Appendix C to include these references (highlighted in blue).
> >
> > **Prior works [CVPT, CoCoOp, ViaPT, V2APT] often rely on complex designs utilizing heuristic strategies** to adapt prompts across inputs. While these methods can be empirically effective, **their functional complexity often makes rigorous theoretical analysis intractable**. By contrast, **VAPT is deliberately designed to maintain a simple yet expressive functional form**. This **simplicity** leads to the clean expert formulations in Equations (13) and (14), which in turn **enable thorough theoretical guarantees without sacrificing empirical performance, an aspect largely missing in prior literature.**
> >
> > In addition, in our response to Reviewer sjhP, we have incorporated the relevant comparative results into the experimental section of the manuscript (highlighted in blue). Our analysis shows that VAPT achieves **competitive performance relative to state-of-the-art methods** across multiple benchmarks. Notably, VAPT attains this performance while **using approximately 50% fewer trainable parameters** (0.36% compared to 0.66%), underscoring the **efficiency** of our method.
> >
> > Furthermore, when using an MAE pre-trained ViT, the aggregated results on the VTAB-1K benchmark, covering 19 diverse datasets, indicate that VAPT surpasses existing methods on several subsets. The table below summarizes these findings and highlights the **effectiveness** of our proposed approach.
> >
> > |Method|VTAB-Natural|VTAB-Specialized| VTAB-Structured|
> > |-|:-:|:-:|:-:|
> > |VPT-Deep|36.02|60.61|26.57|
> > |ViaPT|54.26|78.01|37.52|
> > |VFPT|53.59|77.75|36.15|
> > |**VAPT**|**59.23**|**80.73**|**47.24**|
> >
> > We would like to emphasize that **the primary objective of this research is not necessarily to establish a new state-of-the-art across all datasets**. Rather, as stated in the abstract and introduction, our goal centers on scholarly value rather than solely methodological advancement, specifically, **to provide the research community with insights into the benefits of enhancing the functional expressiveness of prompt experts within visual prompt tuning frameworks**. VAPT serves as a concrete instantiation of this idea. Its adaptive formulation illustrates **the potential of adaptive prompt experts to improve both performance and efficiency**.
> >
> > Consequently, the most meaningful comparison is between VAPT’s adaptive prompt-expert formulation and VPT, in which prompt experts are constant functions. Across the majority of benchmarks, **VAPT consistently outperforms the VPT baseline while requiring a more compact set of trainable parameters**. Moreover, our empirical findings detail the statistical advantages of VAPT over VPT, a conclusion rigorously supported by our theoretical analysis in Section 5 and Appendix A. We are confident that the revised experimental section successfully achieves its intended purpose.
> >
> >
> > **Q4: Trainable Parameters Control**
> >
> > Thank you for this comment. For Gated VPT (Table 2), **we strictly followed the hyperparameter settings and architectural configurations recommended in the original paper**. This ensures that our comparison is faithful to the authors’ intended design and **reflects the method’s best-reported performance**. Since both methods operate in a highly parameter-efficient regime (fewer than 0.3% trainable parameters), we believe it is more informative to compare their respective *optimal* configurations rather than artificially normalizing for parameter count. In our view, this yields the most practical and meaningful assessment of their relative performance.

---

> ### Author Response · Authors · 2025-11-20
> **Response to Reviewer DQQA (Part 3)**
>
> **Q5: Can the proposed visual adaptive prompt method be applied on top of recent VPT variants?**
>
> Thank you for this insightful question. Owing to its simple and modular design (see Figure 1), VAPT is architecturally compatible with a range of recent VPT frameworks. While a comprehensive integration study across all variants is beyond the scope of this work, **which prioritizes analyzing the theoretical expressiveness of prompt experts over purely chasing SOTA metrics**, we agree that such extensions are promising directions for future research. To illustrate feasibility, we briefly outline how VAPT can be integrated with Visual Fourier Prompt Tuning (VFPT) [1], a recent state-of-the-art method.
>
> In VFPT, let the learnable visual prompt parameters in the $l-th$ attention block be denoted by $P^{(l)} = [  p_1^{(l)},\dots,p_N^{(l)} ]$. Before concatenation with the layer input $X^{(l)}$, VFPT applies a 2D Fast Fourier Transform to the prompt embeddings across both the sequence dimension (i.e., $F_\text{seq}$) and hidden dimension (i.e., $F_\text{h}$). The prompts appended to the input are given by $F_{seq} (F_h ([  p_1^{(l)},\dots,p_N^{(l)}] ))$
>
> VAPT can be directly integrated into this framework by replacing the static parameters $p_j^{(l)}$ with the adaptive tokens $P_j^{(l)}(X^{(l)})$ generated from VAPT block, as defined in Equation (12) of the paper. This exemplifies that VAPT’s simple functional form not only enables rigorous theoretical analysis without sacrificing empirical performance, but also facilitates seamless compatibility with advanced VPT variants.
>
> [1] Visual Fourier Prompt Tuning. NeurIPS 2024

---

> > ### Author Response · Authors · 2025-11-26
> > **Looking forward to your response**
> >
> > Dear Reviewer DQQA,
> >
> > We would like to thank you very much for insightful review, and we hope that our response addresses your previous concerns regarding our paper. However, as the discussion period is expected to end in a few days, please feel free to let us know if you have any further comments on our work. We would be willing to address any additional concerns from you. Otherwise, we hope that you will consider increasing your rating.
> >
> > Thank you again for spending time on the paper, we really appreciate it!
> >
> > Best regards,
> >
> > Authors

---

### Author Response · Authors · 2025-11-20
**Response to Reviewers' Comments**

Dear Area Chairs and Reviewers,

We would like to express our sincere gratitude for the time, effort, and constructive feedback you have invested in evaluating our submission. Your insightful comments have significantly helped us improve the clarity and completeness of the paper. We are especially encouraged by the positive assessments, which highlight the following aspects of our work:

- **Contributions**:
    - Strong motivation (Reviewer DQQA).
    - A clear theoretical reinterpretation of VPT through the lens of MoE, providing both conceptual insight and mathematical grounding for understanding prompt tuning behavior (Reviewer YSjb).
    - The proposed Visual Adaptive Prompt Tuning (VAPT) effectively enhances the expressiveness of prompt experts by introducing input-dependent adaptive prompts while **maintaining parameter efficiency** (Reviewer YSjb).
- **Soundness**: The paper presents a variety of experiments, with results on FGVC, VTAB-1K, and both supervised and self-supervised pretrained backbones, demonstrating the robustness of the proposed method. In addition, the ablation studies in the appendix are very helpful for understanding the proposed method (Reviewer DQQA, Reviewer sjhP).
- **Presentation**: The paper is easy to follow (Reviewer DQQA, Reviewer YSjb, Reviewer sjhP).

In what follows, we would like to clarify the novelty of our contribution and explain why we believe the paper is substantively original along the following dimensions:

- **Foundational Motivation**: As noted by Reviewer DQQA, our work is driven by a strong motivation: we connect VPT to MoE and identify a key limitation in the expressiveness of prompt experts. While pre-trained experts $f_j(X) = {W_m^V}^\top E_jX$ adapt to the input, standard prompt experts in VPT $f_{N + j'}(X) = {W^V_m}^\top p_{j'}$ function as static, input-independent experts. This discrepancy naturally raises the question of whether more expressive, input-dependent prompt experts could offer substantial benefits. **Crucially, our approach leverages the existing MoE structure within attention heads rather than imposing an external MoE architecture, as is common in prior work.** This perspective allows us to design VAPT as **a simple and practical implementation while still using MoE theory to rigorously analyze its behavior.**

- **Expressiveness and Efficiency, Without Trade-off**: **A common assumption is that increasing functional expressiveness necessarily comes at the cost of efficiency**. Our design challenges this assumption by demonstrating that both objectives can be achieved simultaneously. VAPT enhances the expressiveness of prompt experts through adaptive, input-dependent prompts while maintaining parameter efficiency (as highlighted by Reviewer YSjb). Moreover, our results show that the benefits extend beyond parameter count: VAPT also exhibits strong data efficiency, **supported by both experimental evidence and theoretical justification**. This dual improvement underscores that expressiveness and efficiency need not be mutually exclusive.

- **Novel Theoretical Insights and Analytical Tractability**: A central contribution of our work is to show that VAPT achieves an optimal sample efficiency rate. **Prior works often rely on more complex architectures and heuristic mechanisms to vary prompts across inputs.**  While these designs can be empirically effective, **their functional complexity often precludes meaningful theoretical analysis**. In contrast, **we deliberately designed VAPT to retain a simple yet expressive functional form**, which leads to the clean expert formulations in Equations (13) and (14). These formulations **enable a thorough theoretical analysis without sacrificing empirical performance, an aspect largely missing in prior literature. The resulting balance between empirical effectiveness and theoretical guarantees addresses a significant gap in the field**: VAPT not only performs well in practice but also offers provable robustness and generalization behavior, thereby adding substantive scholarly value.

We deeply appreciate the reviewers' positive feedback and thoughtful suggestions. We will address each reviewer's specific concerns in our detailed responses below and describe the corresponding revisions made based on their valuable recommendations.

Best regards,

Authors

---

### Author Response · Authors · 2025-12-02
**Discussion Overview for New Area Chair**

We thank the Area Chairs for overseeing the discussion and the reviewers for their thoughtful engagement throughout the rebuttal process. Below, we provide a comprehensive overview of the main contributions of our paper, together with a summary of the key concerns raised by reviewers and our corresponding responses. We hope this concise summary provides a clear view of how each concern has been addressed and how reviewers acknowledged these resolutions in their final assessments.

## **Summary of main contributions**

- **Identifying a key limitation in the formulation of VPT**: As highlighted by Reviewer DQQA, our work is driven by a strong motivation: we connect VPT to MoE and identify a key limitation in the expressiveness of prompt experts. In each attention head, the pre-trained experts $f_j(X) = {W_m^V}^\top E_jX$ are input-dependent, whereas standard prompt experts in VPT, $f_{N + j'}(X) = {W^V_m}^\top p_{j'}$ are static and input-independent. This discrepancy naturally raises the question of whether more expressive, input-dependent prompt experts can yield substantial gains. **Crucially, our approach leverages the existing MoE structure within attention heads rather than imposing an external MoE architecture, as is common in prior work. This perspective allows us to design VAPT as a simple and practical mechanism while still exploiting MoE theory to rigorously analyze its behavior.**

- **Proposing VAPT, a novel formulation that injects input-adaptive prompt experts with VPT-level parameter efficiency**: A common assumption is that increasing functional expressiveness necessarily comes at the cost of computational or parameter efficiency. **Our formulation challenges this assumption**. VAPT enhances the expressiveness of prompt experts via adaptive, input-dependent prompts while maintaining parameter efficiency (as noted by Reviewer YSjb). Beyond parameter efficiency, our experiments show that VAPT also delivers strong data efficiency, and we support these findings with theoretical analysis. Taken together, these results demonstrate that expressiveness and efficiency can be achieved simultaneously, rather than being mutually exclusive.

- **Novel theoretical insights and analytical tractability**: A central contribution of our work is to show that VAPT achieves an optimal sample efficiency rate. **Prior works often rely on more complex architectures and heuristic mechanisms to vary prompts across inputs**. While these designs can be empirically effective, **their functional complexity often precludes meaningful theoretical analysis**. In contrast, we deliberately designed VAPT to retain a simple yet expressive functional form, leading to the clean expert formulations in Equations (13) and (14). These formulations **enable a thorough theoretical analysis without sacrificing empirical performance, an aspect largely missing in prior literature.**


## **Overview of Per-Reviewer Discussion**

| **Reviewer** | **Initial Score** | **Post-rebuttal Score** | **Key Concerns** | **Response**                                                                                                                                            |
|-------------|-------------------|-------------------------|------------------|---------------------------------------------------------------------------------------------------------------------------------------------------------|
| **sjhP**    | 4                 | 6                       | P1, P2, P3                | Confirmed that all concerns were satisfactorily addressed during the rebuttal and subsequently raised the score.                                       |
| **YSjb**    | 6                 | 6                       | P2, P4               | Did not participate in the discussion but maintained a positive assessment until the discussion ended. Nonetheless, all concerns were comprehensively addressed in the revised paper. |
| **DQQA**    | 4                 | 4                       | P1, P2, P3                | **Did not engage in the post-rebuttal discussion** despite the AC reminder. Nonetheless, all concerns were comprehensively addressed in the revised paper. |

---

> ### Author Response · Authors · 2025-12-02
>
> Here is the detailed summary of each primary concern raised and our corresponding response.
>
> ## **Key Concerns & Responses**
>
> **P1: Comparison with recent baseline**
>
> In the revised manuscript (Section 6; changes highlighted in blue), we have incorporated the requested recent baselines. Our analysis shows that VAPT achieves competitive performance relative to state-of-the-art methods across multiple benchmarks. Notably, VAPT attains this performance while using roughly **50%** fewer trainable parameters (0.36% vs. 0.66%), underscoring the **parameter efficiency** of our approach.
>
> When using an MAE-pretrained ViT, the aggregated results on VTAB-1K (19 diverse datasets) further indicate that VAPT outperforms existing methods on several subsets, demonstrating the robustness of the proposed method. **These empirical findings are consistent with, and supported by, our theoretical analysis.**
>
> We have also clarified our overarching goal in Section 6 of the revised manuscript. The primary objective of this work is not to establish a new state of the art on every dataset. As stated in the abstract and introduction, our focus is on scholarly value rather than purely incremental gains: we aim to provide clear insight into the benefits of increasing the functional expressiveness of prompt experts within visual prompt tuning frameworks. VAPT serves as a concrete instantiation of this idea. Its adaptive formulation illustrates how input-dependent prompt experts can deliver **improved performance**, **stronger parameter efficiency**, and **enhanced sample efficiency**, both empirically and theoretically.
>
> Finally, we note that Reviewer sjhP, the only reviewer who participated in the post-rebuttal discussion, explicitly acknowledged that this concern had been satisfactorily addressed.
>
>
> **P2: Discussion of relevant work**
>
> In the revised manuscript, we have expanded the discussion of related work in Appendix C (changes highlighted in blue). The key distinctions between our approach and prior methods are as follows:
>
> - **Use of existing MoE structure**. We explicitly leverage the MoE structure already present within each self-attention head, rather than introducing an additional external MoE architecture. This leads to a simple and straightforward implementation of VAPT.
>
> - **Analytical tractability versus architectural complexity.** Prior works that vary prompts across inputs typically rely on more complex architectures or heuristic mechanisms. While often empirically effective, their functional complexity generally precludes rigorous theoretical analysis. In contrast, our design yields a clean expert formulation, which enables a thorough theoretical study of its effectiveness, an aspect largely missing in the existing literature.
>
> Again, Reviewer sjhP, the only reviewer who engaged in the post-rebuttal discussion, explicitly acknowledged that this concern had been satisfactorily addressed.
>
>
> **P3: The novelty of our work**
>
> We have further highlighted the novelty of our contributions in Section 4.2 of the revised manuscript (changes highlighted in blue), emphasizing how VAPT uniquely combines input-adaptive prompt experts, parameter efficiency, and analytical tractability within a unified framework.
>
> Reviewer sjhP, the only reviewer who participated in the post-rebuttal discussion, also explicitly acknowledged that this concern had been satisfactorily addressed.
>
> **P4: Experiments in multimodal settings**
>
> We have added experiments on text–image retrieval task in Appendix E.10. The results again demonstrate that VAPT surpasses VPT by a considerable margin, highlighting the robustness of our method across different modalities and settings.

---

### Meta-Review · Area_Chair_3WrR · 2025-12-28

**Summary:**

This work provides a MoE-based reinterpretation of visual prompt tuning and identifies a concrete limitation of standard VPT: prompt “experts” are static (input-independent), unlike the input-dependent experts already present in attention. It proposes VAPT to make prompt experts input-adaptive while staying parameter-efficient, and supports the approach with both theory (sample-efficiency analysis) and extensive empirical results. Reviewers generally agreed the paper is clear and the experiments are solid, while concerns were mainly about novelty/positioning and missing comparisons; after rebuttal, one reviewer raised the score from 4→6, while another stayed at 4.

**Reviewer Concerns:**

Addressed by the rebuttal / revision:
- Baseline coverage: Added/expanded comparisons with recent VPT variants and PEFT baselines, and clarified the goal of the work (insight + expressiveness rather than only SOTA chasing).
- Related work / novelty positioning: Expanded discussion distinguishing VAPT from prior adaptive prompting/prompt-pool methods and from explicit MoE prompt-tuning approaches; clarified what is inherited from prior MoE interpretations vs what is new here.
- Qualitative evidence: Added qualitative analysis (e.g., attention-map style comparisons) to better illustrate how adaptive prompts behave differently from static prompts.
- Beyond standard classification: Added an image–caption retrieval experiment showing VAPT improves over VPT in a multimodal setting.
- Scalability discussion: Added/pointed to results on larger backbones and argued the added modules remain lightweight.

Still outstanding / partially addressed:
- Novelty: Even with stronger positioning, “input-adaptive prompts” overlaps with prior work; the core remaining question is whether the contribution is sufficiently distinct beyond the MoE framing + a simpler formulation.
- Strict parameter-controlled fairness: Comparisons are improved, but a fully controlled “same trainable-parameter budget” sweep would make the empirical story cleaner.

**Reviewer Scores:**

sjhP: 4 → 6 (already updated after discussion; likely stays 6).

YSjb: 6 → 6 (did not discuss; likely remains 6).

DQQA: 4 → 4 (did not discuss; could plausibly move to ~5 if fully considering the added comparisons, but uncertainty remains).

---

### Decision · Program_Chairs · 2026-01-26

Accept (Poster)